# Ybx1 fine-tunes PRC2 activities to control embryonic brain development

Myron K. Evans [1], Yurika Matsui[1,4], Beisi Xu [2,4], Catherine Willis[1], Jennifer Loome [1], Luis Milburn[1], Yiping Fan[2], Vishwajeeth Pagala[3] & Jamy C. Peng [1✉]

Chromatin modifiers affect spatiotemporal gene expression programs that underlie organismal development. The Polycomb repressive complex 2 (PRC2) is a crucial chromatin modifier in executing neurodevelopmental programs. Here, we find that PRC2 interacts with the nucleic acid–binding protein Ybx1. In the mouse embryo in vivo, Ybx1 is required for forebrain specification and restricting mid-hindbrain growth. In neural progenitor cells (NPCs), Ybx1 controls self-renewal and neuronal differentiation. Mechanistically, Ybx1 highly overlaps PRC2 binding genome-wide, controls PRC2 distribution, and inhibits H3K27me3 levels. These functions are consistent with Ybx1-mediated promotion of genes involved in forebrain specification, cell proliferation, or neuronal differentiation. In *Ybx1*-knockout NPCs, H3K27me3 reduction by PRC2 enzymatic inhibitor or genetic depletion partially rescues gene expression and NPC functions. Our findings suggest that Ybx1 fine-tunes PRC2 activities to regulate spatiotemporal gene expression in embryonic neural development and uncover a crucial epigenetic mechanism balancing forebrain–hindbrain lineages and self-renewal–differentiation choices in NPCs.

[1] Department of Developmental Neurobiology, St. Jude Children's Research Hospital, 262 Danny Thomas Place, Memphis, TN 38105, USA. [2] Center for Applied Bioinformatics, St. Jude Children's Research Hospital, 262 Danny Thomas Place, Memphis, TN 38105, USA. [3] Center for Proteomics and Metabolomics, St. Jude Children's Research Hospital, 262 Danny Thomas Place, Memphis, TN 38105, USA. [4] These authors contributed equally: Yurika Matsui, Beisi Xu. ✉email: jamy.peng@stjude.org

Embryonic NPCs massively amplify, specify to different brain regions and the spinal cord, and subsequently differentiate into neurons and glial cells. NPC amplification, specification, and differentiation/maturation require an intricate coordination between extracellular signals and cell-intrinsic mechanisms[1]. This coordination is achieved by epigenetic machineries that integrate instructions from signaling pathways and transcription factor networks to output gene expression programs. In developing invertebrate and vertebrate brains, Polycomb group proteins integrate spatial and temporal signals to execute gene expression programs underlying neural diversity[2–4].

PRC2 is a Polycomb group ribonucleoprotein complex that methylates lysine 27 of histone H3 (H3K27me) and forms a closed chromatin structure to suppress transcription by RNA polymerase II[5,6]. In humans, mutations in PRC2 core subunits have been causally linked to Weaver syndrome, a congenital multisystemic syndrome characterized by craniofacial defects, intellectual disabilities, and often macrocephaly, bone and joint malformations, and an increased predisposition to cancer[7,8]. In mice, depletion of the PRC2 core subunits Ezh2, Suz12, and Eed results in gastrulation arrest[9–11]; their knockout (KO) in embryonic stem cells (ESCs) result in failure in neural ectoderm specification[12–14]. Further loss of function studies of PRC2 have demonstrated that PRC2 is required for brain region specification, region-specific proliferation of NPCs, and growth control of brain regions and the spinal cord[11,15–17]. These influences suggest that PRC2 function and regulation vary in different neurodevelopmental contexts.

PRC2 exists as two variant complexes that are defined by their accessory subunits: PRC2.1 associates with subunit EPOP or PALI1[18–20], whereas PRC2.2 associates with JARID2 and AEBP2[21]. These proteins and others likely function distinctively in the PRC2 complex's binding to chromatin, methylating H3K27, and/or suppressing genes. To shed light on PRC2 function and regulation during neural development, we used a proteomics-based approach to identify new interactors of JARID2. JARID2 critically regulates PRC2.2 and neural development: it is required for PRC2.2 to bind chromatin, influences H3K27me levels, and neural tube development. Our effort identified Y-box binding protein 1 (Ybx1) as a previously uncharacterized PRC2-interacting protein.

Ybx1 is a nucleic acid-binding protein known to affect transcriptional activation, DNA repair and replication, RNA processing and stability, and protein translation[22,23]. Through its diverse molecular roles, Ybx1 regulates proliferation, apoptosis, cell differentiation, and cell stress response. For example, Ybx1 binds the consensus sequence, 5′-CTGATTGG-3′, to mediate transcriptional activation of genes involved in epithelial-to-mesenchymal transition and drug resistance, likely thereby promoting cancer progression[24]. Its overexpression promotes glioblastoma and medulloblastoma cancer cell proliferation[25,26]. Ybx1 depletion results in hematopoiesis failure and exencephaly in the mouse embryo[27–29]. Although Ybx1-KO animals show defects in neural tube closure and growth, the cellular and molecular bases have remained unclear. Here, we show that Ybx1 regulates the self-renewal and neuronal differentiation of NPCs in part through regulating PRC2. PRC2 regulation by Ybx1 fine-tunes the spatiotemporal expression of neurodevelopmental genes.

## Results

**YBX1 is a candidate PRC2-interacting factor**. We used mass spectrometry to detect endogenous proteins that co-immunoprecipitated with JARID2 from the human ESC (hESC) nuclear extract. We used a validated, specific anti-JARID2 antibody (Supplementary Fig. S1a, b), which did not co-immunoprecipitate G9A in the hESC nuclear extract (Supplementary Fig. S1c). In addition to JARID2, we detected the known PRC2 subunits EZH2, SUZ12, EED, AEBP2, and RBBP4 (Supplementary Fig. S1d, e). Importantly, we detected YBX1 (Supplementary Fig. S1d, e), which was also uncovered by Ezh2 immunoprecipitation-mass spectrometry from NT2 cells[30]. These provided the rationale for our study to examine Ybx1's function and interaction with PRC2.

**Nuclear Ybx1 modulates the self-renewal of NPCs**. Ybx1-KO mice are known to display a strong, highly penetrant exencephaly phenotype[27,29]. We analyzed Ybx1-KO mice in a back-crossed C57BL/6 background and found that ~84% displayed exencephaly, confirming previous findings (Supplementary Fig. S2a, b). Exencephaly can be caused by deregulation of NPCs, so we examined how Ybx1 affects NPC properties. First, we examined NPC proliferation in Ybx1-KO embryos and sibling controls by immunofluorescence (IF) and fluorescence-activated cell sorting (FACS). To identify proliferating cells in vivo, we injected BrdU into pregnant dams and/or detected the mitotic chromatin marker phosphorylated-serine 10 in histone H3 (PH3; Fig. 1a). Mouse NPCs were identified by their expression of the neural stem cell marker Sox2. IF of cryosections of embryonic day 13.5 (E13.5) mouse neural tubes showed a significant increase in the number and frequency of proliferating BrdU-positive NPCs in Ybx1-KO embryos compared to controls ($P < 0.0001$; Fig. 1b, c). As the number of Sox2-positive NPCs in Ybx1-KO embryos was significantly higher than that in control embryos, we normalized BrdU counts to the total number of Sox2-positive cells that remained significant when compared to the control ($P < 0.05$; Fig. 1d).

FACS analysis showed that, on average, the proportion of BrdU-positive NPCs was ~14% in control and ~26% in Ybx1-KO neural tubes at E13.5, which confirmed the IF results (Supplementary Fig. S2c, d). Quantification of matched littermate embryos revealed that there were significantly more BrdU-positive ($P < 0.01$; Supplementary Fig. S2d) and PH3-positive ($P < 0.05$; Supplementary Fig. S2d) NPCs in Ybx1-KO than in sibling control embryos. In contrast, Ybx1-KO mouse embryonic fibroblasts (MEFs) displayed decreased proliferation compared to controls (Supplementary Fig. S2e–g), consistent with previous reports[27]. These data suggest that Ybx1 has different cellular influences in different developmental contexts: promoting the proliferation of fibroblasts and suppressing the proliferation of NPCs.

We used the neurosphere assay to examine the effect of Ybx1 over self-renewal. Mouse NPCs were purified by Neurofluor™ CDr3 FACS (which targets NPC marker FABP7[31]) and allowed to form neurospheres that were serially passaged (Fig. 1e). We quantified neurosphere number and area and found that, by the third passage, both were higher for Ybx1-KO compared to controls (Fig. 1f–h). A separate clonal assay confirmed that Ybx1-KO NPCs had increased self-renewal compared to controls (Supplementary Fig. S2i, j).

To investigate the nuclear role of Ybx1 in NPC proliferation, we transduced Ybx1-KO NPCs with viruses expressing wild-type YBX1 or a mutant without the nuclear localization signal (delNLS from deletion of amino acids 183-205; Fig. 1i). In transduced Ybx1-KO NPCs, wild-type YBX1 proteins were expressed in both the nucleus and cytoplasm, whereas the delNLS–YBX1 proteins were expressed only in the cytoplasm (Fig. 1j and Supplementary Fig. S2k). While expression of PRC2 subunits were unaffected, wild-type (but not delNLS) YBX1 restored the expression of some forebrain lineage genes (Supplementary Fig. S2l). We found that

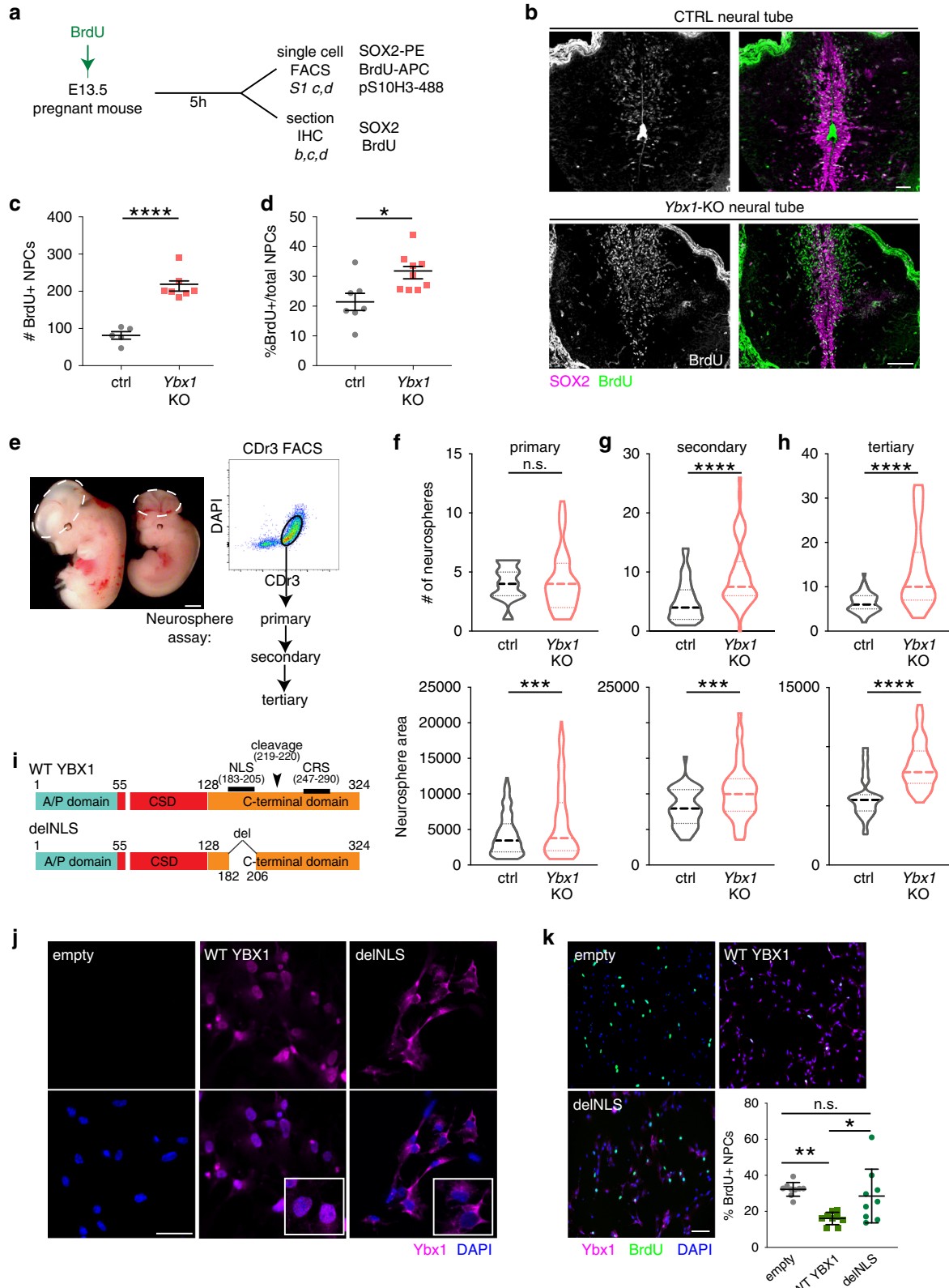

wild-type YBX1 resulted in reduced proliferation of *Ybx1*-KO NPCs, whereas delNLS–YBX1 did not significantly affect the proliferation of *Ybx1*-KO NPCs (Fig. 1k). These data suggest that nuclear Ybx1 modulates the self-renewal of NPCs.

**Ybx1 is required for neurodevelopmental gene expression**. To examine how Ybx1 affects gene expression in NPCs, we performed RNA-sequencing (RNA-seq) of Sox2-positive NPCs sorted from E13.5 *Ybx1*-KO and sibling controls (Supplementary Fig. S3a). We analyzed expressed genes as those having FPKM values > 1 in either control or *Ybx1*-KO NPCs. Using the criteria of adjusted $P < 0.05$ and fold-change > 1.5 to compare three datasets each from control and *Ybx1*-KO NPCs, we identified 604 upregulated genes and 366 downregulated genes in *Ybx1*-KO

**Fig. 1 *Ybx1*-KO NPCs had increased self-renewal. a** Schematic for BrdU labeling and analyses of mouse embryos. **b** IF of Sox2 and BrdU in cryosections from *Ybx1*-KO or sibling control embryos at E13.5. Bar, 100 μm. Quantification of IF in cryosections to determine the **c** number of BrdU-positive NPCs per cryosection (*n* = 7/genotype) or **d** percentage of BrdU-positive NPCs (*n* = 7 control or 9 *Ybx1*-KO biologically independent embryos) from *Ybx1*-KO and sibling control embryos. **e** Diagram of NPC isolation by CDr3 FACS and subsequent neurosphere formation and serial passages. Bar, 10 μm. Quantification of the number and area of **f** primary, **g** secondary, and **h** tertiary neurospheres formed by NPCs from *Ybx1*-KO and sibling control embryos. **i** Diagrams of WT full-length (FL) and mutant delNLS-YBX1 cDNAs. **j** Ybx1 IF in *Ybx1*-KO NPCs transduced with lentiviruses from empty vector, WT YBX1, or delNLS–YBX1. Inserts contain zoom-in images of representative cells. Bar, 50 μm. **k** YBX1 and BrdU IF in *Ybx1*-KO NPCs transduced with lentiviruses from empty vector, WT YBX1, or delNLS–YBX1. *n* = 9 images from one experiment, which was repeated twice. Bar, 50 μm. Quantification of NPCs transduced with different lentiviruses. Data are presented as mean ± SEM for **c**, **d**, and **k** and as violin plot of frequency distribution (lines at median and quartiles) in **f**–**h**. *P* values by two-tailed unpaired *t* test are indicated. n.s. indicate not significant. *, **, ***, and **** indicate *P* < 0.05, 0.01, 0.001, and 0.0001, respectively. Source data are provided in Source Data file. *P* values: 1**c**-<0.0001; 1**d**-0.01; 1**f**(bottom)-0.0005; 1**g**(top)-<0.0001; 1**g**(bottom)-0.002; 1**h**(top)-<0.0001, 1**h** (bottom)-<0.0001, 1**k**(empty-WT)-0.003, 1**k**(WT-delNLS)-0.02.

(Fig. 2a, Supplementary Fig. S3b). Many of the differentially expressed genes were involved in nervous system development or cell fate (Fig. 2b). Gene ontology and gene set enrichment analysis (GSEA) revealed that downregulated genes in *Ybx1*-KO NPCs were enriched in functions related to central nervous system, neuronal differentiation, and morphogenesis (Fig. 2c, d). Upregulated genes in *Ybx1*-KO NPCs were enriched in functions related to embryo development, transcriptional regulation, cell proliferation, and known YBX1 targets[32] (Fig. 2e, f). These data suggest that Ybx1 suppresses genes involved in cell proliferation and promotes genes involved in neuronal differentiation.

As many differentially expressed genes in *Ybx1*-KO NPCs were related to brain development, we wanted to determine whether the observed changes in gene expression occurred in NPCs from the brain, the spinal cord, or both. Thus, we separately purified NPCs (by Neurofluor™ CDr3 FACS) from these 2 regions and performed RT-qPCR. Neurodevelopmental genes that we analyzed were significantly downregulated in brain NPCs but not similarly affected in spinal cord NPCs (Supplementary Fig. S3c). Therefore, we focused on Ybx1 in the developing brain.

Our RNA-seq analysis showed that several crucial forebrain lineage genes were downregulated in *Ybx1*-KO NPCs. These genes included *Fgf8*[33], *Six3*[34,35], *Emx2*[36], *Arx*[37], and *Dkk1*[38], which are required for the patterning and formation of the forebrain. Other downregulated genes included *Hes5*, which is required for NPC proliferation and differentiation to neurons[39,40], and *Fezf2*, which is required for neuronal differentiation in the forebrain[41]. TaqMan™ assays confirmed the downregulation of forebrain lineage genes (Fig. 2g) and the upregulation of midbrain and hindbrain lineage genes (Supplementary Fig. S3d) in *Ybx1*-KO NPCs. IF analysis of E13.5 sagittal cryosections confirmed the decreased expression and coverage of the forebrain marker Foxg1 (Fig. 2h), whereas midbrain and hindbrain regions positive for the marker Gbx2 were expanded (Supplementary Fig. S3e) in *Ybx1*-KO compared to controls. We infer that reduced forebrain specification in *Ybx1*-KO embryos led to the overgrowth of the midbrain and hindbrain. We concluded that Ybx1 regulates the expression of genes that are crucial for brain patterning and formation, NPC proliferation, and neuronal differentiation.

**Ybx1 is required for the differentiation of NPCs to neurons.** Our RNA-seq results suggest that in NPCs, Ybx1 is required to promote the expression of neuronal differentiation genes (Fig. 2c, d). Therefore, we used a 14-day in vitro differentiation assay (Fig. 3a) to test whether Ybx1 affects the differentiation of NPCs (purified by Neurofluor™ CDr3 FACS) to neurons (Tuj1 marker) or glia (Gfap marker). This assay enabled us to focus on the cell-intrinsic role of Ybx1 in NPCs. We observed reduced expression of neuronal markers in differentiating cells from *Ybx1*-KO versus control (Fig. 3b, c). Quantification of IF showed significantly

fewer Tuj1-positive neurons and shorter and less-arborized neurites in the sparse Tuj1-positive neurons of *Ybx1*-KO than in sibling controls (Fig. 3d–g). Although we detected a modest and significant increase in glial gene expression in *Ybx1*-KO differentiating cells, the number of Gfap-positive cells were not significantly different between *Ybx1*-KO and control (Fig. 3h). These findings are consistent with a previous report that sustained Ybx1 expression correlates with neuronal differentiation whereas Ybx1 loss correlates with gliogenesis[25]. We conclude that Ybx1 is required for an NPC-intrinsic mechanism to promote gene expression during neuronal differentiation.

**Ybx1 regulates the expression of neurodevelopmental genes.** We profiled the genome-wide distribution of Ybx1 by CUT&RUN-seq[42] in *Ybx1*-KO and sibling control NPCs (by Sox2-GFP FACS; Fig. 4a). ChIP-seq and CUT&RUN-seq were performed with spike-in method: *Drosophila* S2 chromatin for histone profiling and H2Av CUT&RUN of S2 cells for Ybx1 CUT&RUN-seq. The heat map in Fig. 4a shows Ybx1 distribution in *Ybx1*-KO and control NPCs. Using a $10^{-3}$ false discovery rate to analyze seq read counts per million, we identified 23,070 Ybx1-bound regions in NPCs and only 493 (445 overlapping those in control) regions in *Ybx1*-KO NPCs; showing the high specificity of Ybx1 CUT&RUN-seq. Approximately 46% of Ybx1-bound peaks were within 10 bp of the transcription start site (TSS; Supplementary Fig. S4a) and 58.8% were within promoters (within 2 kb of transcription start sites; Supplementary Fig. S4b). Of Ybx1-bound peaks, most located in protein coding genes (Supplementary Fig. S4b). Only 15.3% of Ybx1-bound peaks were located distal to a gene.

We found that 382 of the 604 (63%) upregulated genes and 263 of the 366 (68%) downregulated genes in *Ybx1*-KO NPCs were bound by Ybx1 in NPCs (Fig. 4b). The increased frequency of Ybx1 binding in downregulated genes versus upregulated genes was significant ($P = 0.0062$ by two-tailed Fisher's exact test; Fig. 4b). The downregulated Ybx1-bound genes were enriched in functions related to nervous system development, neuron differentiation, synapse transmission, axon guidance, glutamate signaling, and BDNF signaling (Supplementary Fig. S4c), suggesting that Ybx1 directly promotes the expression of genes related to brain development, neurogenesis, and neuronal function. Upregulated Ybx1-bound genes were enriched in functions related to neural crest differentiation, transcriptional regulation, cell cycle, dopaminergic neurogenesis, transcriptional misregulation in cancer, and *Hox* genes in hindbrain development (Supplementary Fig. S4d), suggesting that Ybx1 directly suppresses genes involved in these functional categories. Unsupervised clustering of differentially expressed genes in these categories effectively separated control and *Ybx1*-KO NPCs (Supplementary Fig. S4e), further supporting that Ybx1 directly regulates the expression of neurodevelopmentally important genes.

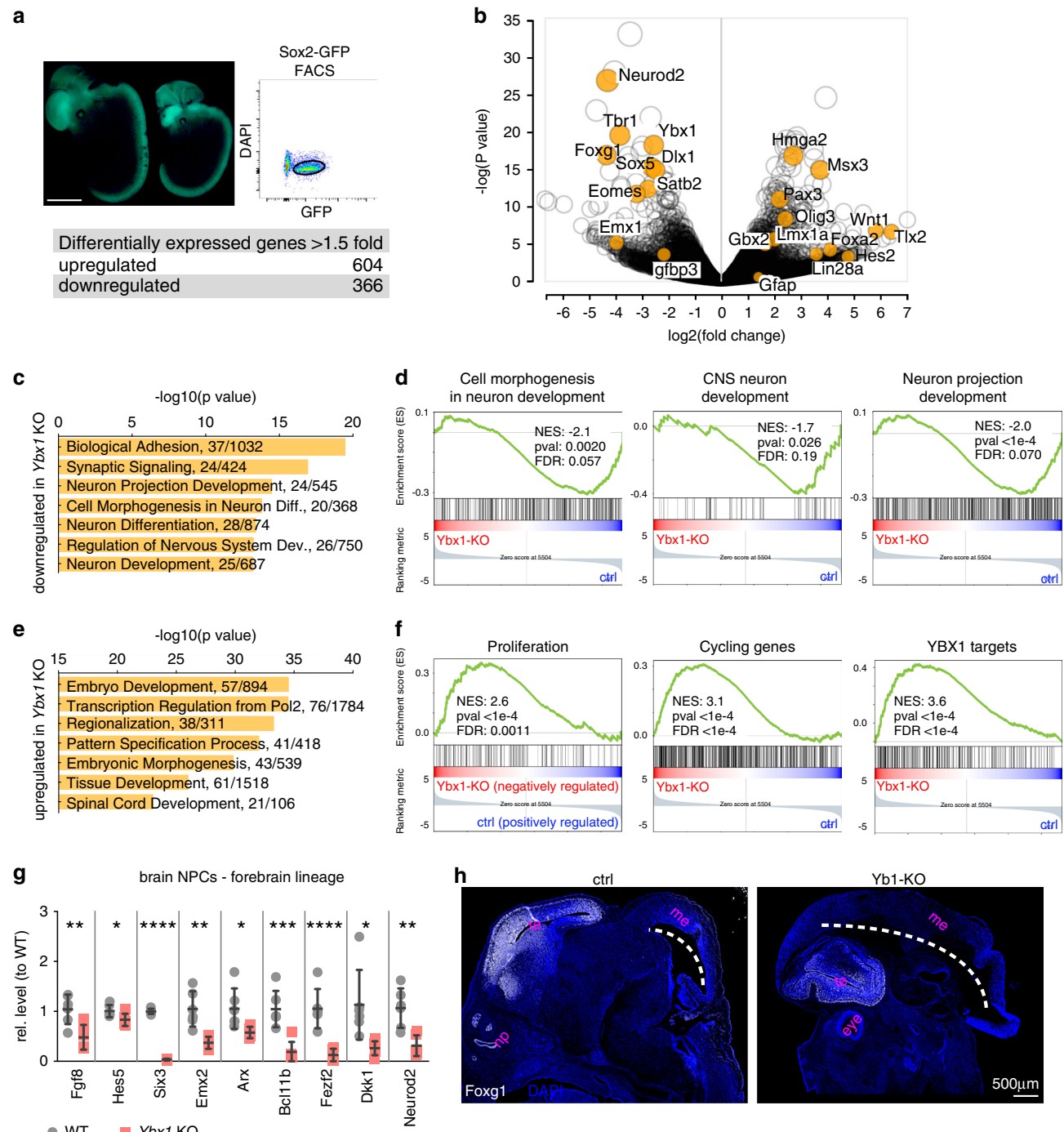

**Fig. 2 Differential gene expression in control and *Ybx1*-KO NPCs purified from embryos. a** Image of *Ybx1*-KO and sibling control embryos expressing Sox2-GFP. Bar, 2 mm. Representative plot of Sox2-GFP FACS. Summary of differentially expressed genes in Sox2-GFP–positive NPCs from Ybx1-KO. **b** Volcano plot comparing transcript profiles of four samples of control NPCs and three samples of *Ybx1*-KO NPCs. *p*-values by two-tailed unpaired *t* test in limma package. Gene ontology analysis identified ontology terms of **c** downregulated or **e** upregulated genes in *Ybx1*-KO vs. control NPCs. Terms were ranked by *P* value, calculated by two-tailed Fisher's exact test, with the number of enriched genes indicated. GSEA identified enrichment gene sets in **d** downregulated or **f** upregulated genes in *Ybx1*-KO NPCs. *P* values were calculated by one-tailed Kolmogorov–Smirnov statistic test. **g** RT-qPCR with TaqMan assays of forebrain lineage markers in NPCs purified from brains of *Ybx1*-KO or sibling control embryos at E13.5. $n = 6$. Data are presented as mean ± SEM for **g**. *P* values by two-tailed unpaired *t* test are indicated. n.s., not significant. *, **, ***, and **** indicate $P < 0.05$, 0.01, 0.001, and 0.0001, respectively, by one-sided Student's *t* test. **h** Foxg1 IF in cryosection from *Ybx1*-KO or sibling control embryos at E13.5. Bar, 500 μm. Te telencephalon, me mesencephalon, np nasal plate. Source data are provided in Source Data file. *P* values: 2**g**(Fgf8)-0.005, (Hes5)-0.03, (Six3)-<0.0001, (Emx2)-0.001, (Arx)-0.03, (Bcl11b)-0.0005, (Fezf2)-0.0003, (Dkk1)-0.01 (NeuroD2)-0.002.

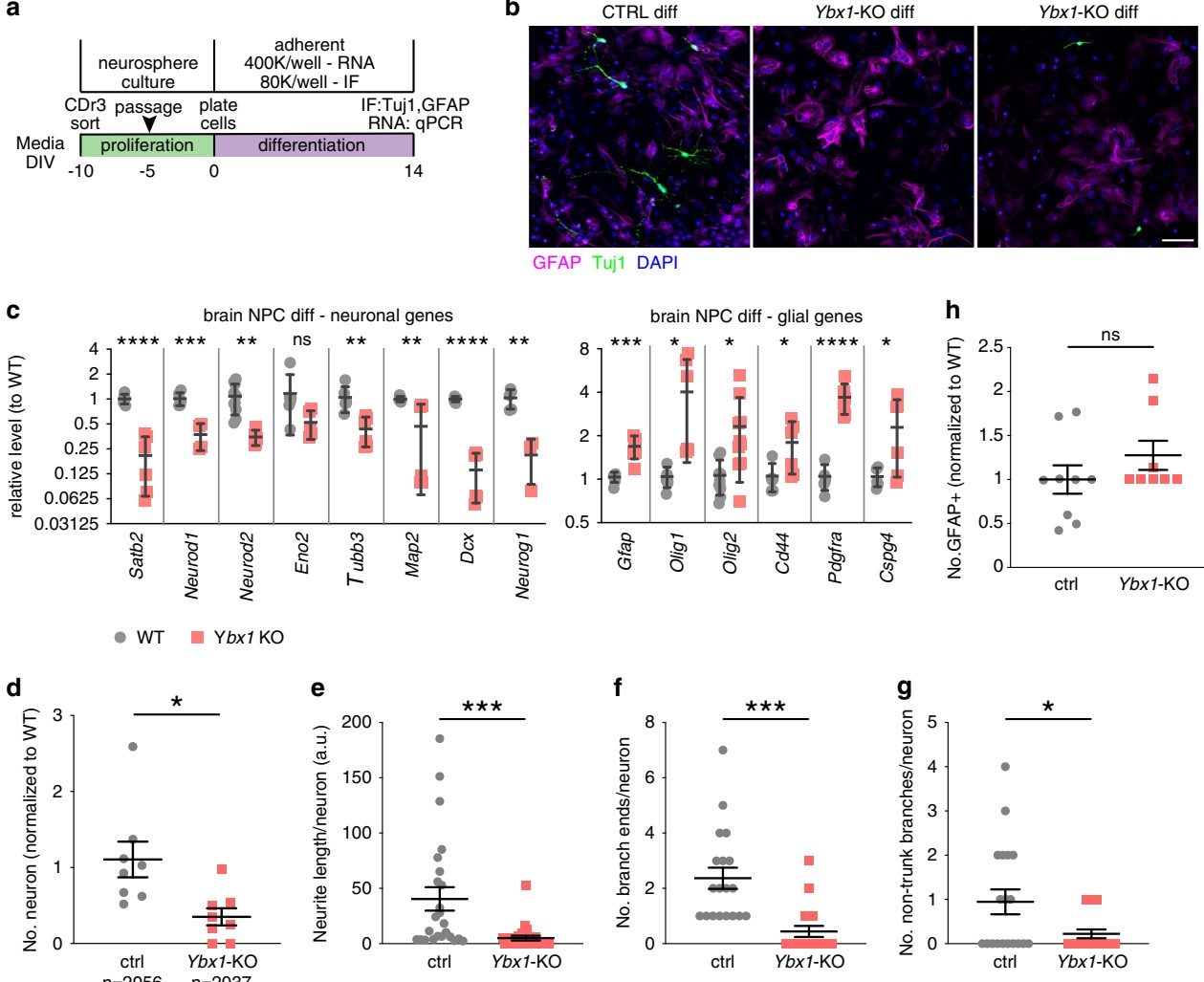

**Fig. 3 Ybx1 is required for the differentiation of NPCs to neurons. a** Diagram illustrating the workflow of in vitro differentiation of NPCs. **b** IF of Tuj1 and Gfap in *Ybx1*-KO and sibling control differentiating cells. **c** RT-qPCR with TaqMan assays of neuronal and glial genes in control and *Ybx1*-KO differentiating cells at day 14. $n = 6$/genotype. $n = 10$ for *Neurod2* in WT, and $n = 10$ for *Olig2* (see source data). Quantification of **d** relative number of Tuj1-positive neurons, **e** neurite length, **f** number of branch points in axons, **g** number of non-trunk branches (not connected to axons or dendrites; $n = 8$ images and cell numbers indicated), and **h** number of Gfap-positive (glial marker; $n = 9$ control images and 8 *Ybx1*-KO images) cells in *Ybx1*-KO and sibling control differentiating cells. Data are presented as mean ± SEM for **c–h**. *P* values by two-tailed unpaired *t* test are indicated. n.s. indicates not significant. *, **, ***, and **** indicate $P < 0.05$, 0.01, 0.001, and 0.0001, respectively. Source data are provided in Source Data file. *P* values: 3**c**(Satb2)-<0.0001, (NeuroD1)-0.0003, (NeuroD2)-0.003, (Tubb3)-0.004, (Map2)-0.009, (Dcx)-<0.0001, (NeuroG1)-0.003, (Gfap)-0.0005, (Olig1)-0.02, (Olig2)-0.01, (Cd44)-0.04, (Pdgfra)<0.0001, (Cspg4)-0.03; 3**d**-0.01; 3**e**-0.002; 3**f**-0.0001; 3**g**-0.02.

**Ybx1 suppresses H3K27me3 levels in brain NPCs.** Given that we detected an interaction between Ybx1 and PRC2, we investigated whether Ybx1 affects the genomic distribution of H3K27me3, which is deposited by PRC2, and H3K4me3 by low-input ChIP-seq[43] and CUT&RUN-seq[42] in *Ybx1*-KO and sibling control NPCs (Fig. 4a). H3K27me3 and H4me3 levels were similar at *Actb* locus in control and *Ybx1*-KO NPCs (Supplementary Fig. S4f). H3K27me3 levels were markedly higher at select forebrain lineage genes including *Foxg1*, *Neurod2*, and *Satb2* in *Ybx1*-KO versus control NPCs, whereas H3K4me3 levels were unchanged (Fig. 4c). These were downregulated in *Ybx1*-KO NPCs and bound by Ybx1 in control NPCs. *Hox* C genes were bound by Ybx1 and displayed higher levels of H3K27me3 in *Ybx1*-KO vs. control NPCs (Supplementary Fig. S4g), but this did not strongly correlate with gene expression changes. H3K27me3 levels were unchanged and H3K4me3 levels increased at genes including *Hmga2* and *Lin28a* in *Ybx1*-KO vs. control NPCs; these

genes were bound by Ybx1 and upregulated in *Ybx1*-KO NPCs (Supplementary Fig. S4h).

Genome wide, H3K27me3 ChIP-seq signals were higher in *Ybx1*-KO than those in control NPCs (Fig. 4a, d). The observed increases in H3K27me3 ChIP-seq signals in *Ybx1*-KO NPCs were validated by H3K27me3 CUT&RUN-seq (Supplementary Fig. S5a). However, the distribution of H3K27me3-occupied regions was largely the same in control and *Ybx1*-KO NPCs (Supplementary Fig. S5b, c), suggesting that few H3K27me3-occupied regions occurred de novo in *Ybx1*-KO NPCs. There was a strong correlation between downregulated gene expression and increased levels of H3K27me3 (nearly passing a more stringent FDR criterion; Fig. 4d, e, Supplementary Fig. S5d), suggesting that Ybx1 reduces H3K27me3 levels to promote gene expression. We concluded that Ybx1 attenuates the deposition of H3K27me3 to promote the expression of genes involved in forebrain specification and neuronal differentiation.

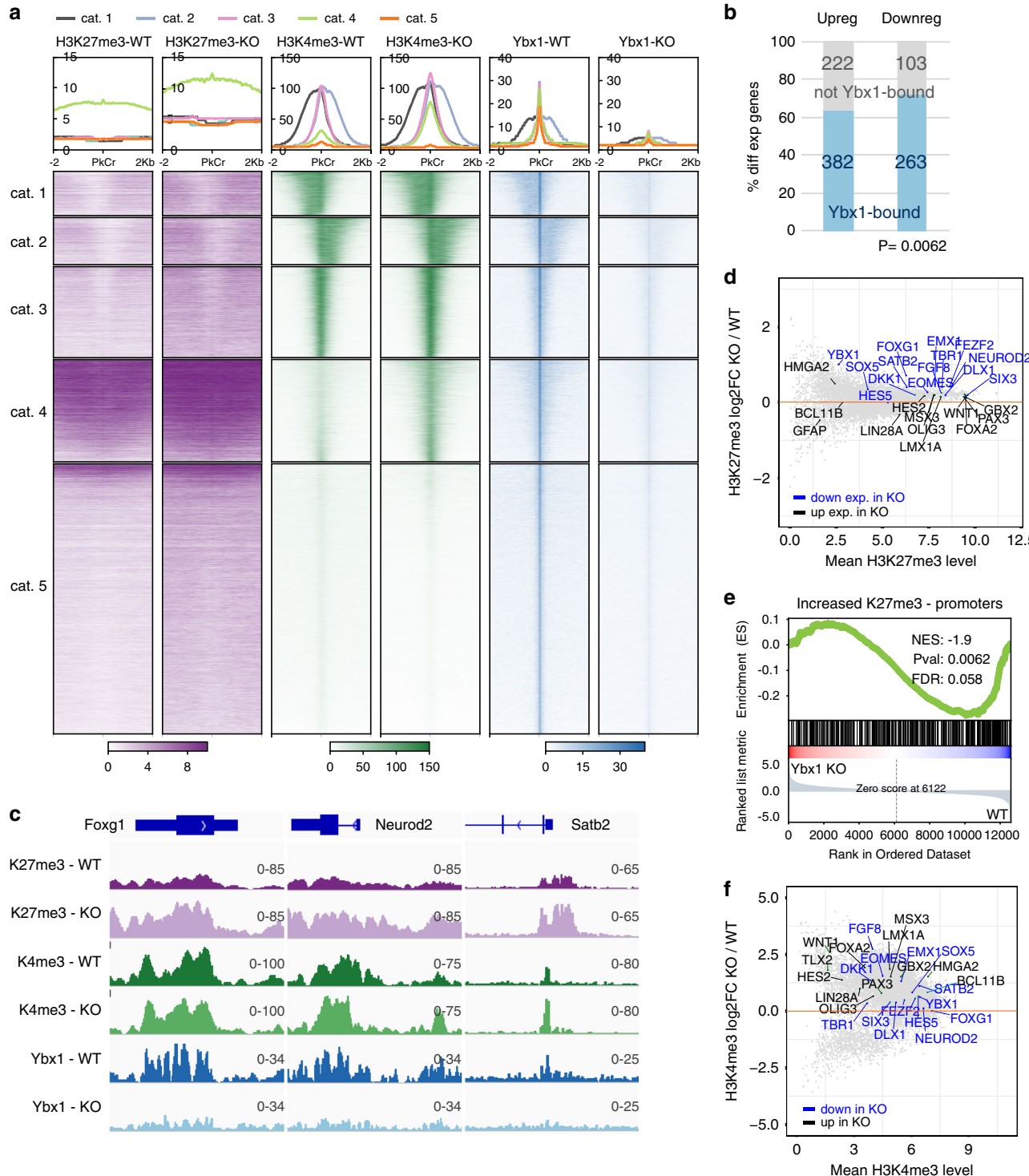

**Fig. 4 Ybx1 suppresses H3K27me3 levels. a** Heat maps indicate the binding intensity of H3K27me3, H3K4me3, and Ybx1, with low (white) or high (dark color) intensity, within 2 kb of all Ybx1 CUT&RUN-seq peaks. Average binding profiles from five different categories (cat.) are presented. In all five categories, H3K27me3 levels increased in *Ybx1*-KO NPCs. Cat. 1, H3K4me3 peaks to the left of Ybx1 peaks; 2, H3K4me3 peaks to the right of Ybx1 peaks; 3, H3K4me3 peaks at Ybx1-bound peak center; 4, high H3K4me3 increase in *Ybx1*-KO; 5, no H3K4me3. **b** Proportion of differentially expressed genes in *Ybx1*-KO vs. WT NPCs bound by Ybx1 protein in control NPCs. *P* values were calculated by two-tailed Fisher's exact test. **c** H3K27me3 and H3K4me3 ChIP-seq and Ybx1 CUT&RUN-seq tracks at *Foxg1*, *Neurod2*, and *Satb2* gene loci in *Ybx1*-KO and sibling control NPCs. **d** MA plot of LOG2(H3K27me3 fold-changes of *Ybx1*-KO/WT NPCs) versus mean H3K27me3 levels at individual genes. Blue indicates downregulated genes and black indicates upregulated genes in *Ybx1*-KO NPCs. **e** GSEA of genes containing significantly increased H3K27me3 levels at promoters with differentially expressed genes in *Ybx1*-KO versus control NPCs. *P* values were calculated by one-tailed Kolmogorov–Smirnov statistic test. **f** MA plot is similar to **d**, but shows H3K4me3 levels.

We separated Ybx1-bound regions into five categories by intensity and location of H3K27me3 and H3K4me3 distribution (relative to Ybx1 peaks) in *Ybx1*-KO NPCs, as shown by different average profiles at the top of the heat map in Fig. 4a. (To facilitate explanation, profiles were separated by categories in Supplementary Fig. S5e). For example, H3K27me3 and H3K4me3 levels increased at Ybx1-bound regions in category 4 in *Ybx1*-KO NPCs. H3K4me3 ChIP-seq signals genome-wide were higher in *Ybx1*-KO than in control NPCs (Fig. 4f), but the numbers of H3K4me3-occupied regions at promoters 3′ of TSS, exons, introns, termination sites, distal regions (within 50 kb of a coding gene), and intergenic regions all decreased in *Ybx1*-KO NPCs (Supplementary Fig. S5f, g). Although genes (regardless of Ybx1 binding status) with higher H3K4me3 levels were enriched in upregulated genes in *Ybx1*-KO NPCs (Supplementary Fig. S5h), we did not observe Ybx1 protein binding with H3K4 methyltransferases, which are represented by the core subunit Rbbp5 (Supplementary Fig. S5i). These data suggest that Ybx1 does not directly influence H3K4me3.

We next analyzed H3K27me3 and H3K4me3 distributions at TSSs of Ybx1-bound genes that were unchanged, downregulated, or upregulated in *Ybx1*-KO NPCs (Supplementary Fig. S5j). As an additional comparison, we generated the same profiles at genes that were not bound by Ybx1; they had little change in *Ybx1*-KO NPCs (Supplementary Fig. S5k). H3K27me3 levels increased by approximately 2 fold at all Ybx1-bound TSSs. WB analysis of neural tube cells showed that H3K27me3 levels were higher in *Ybx1*-KO vs. control (Supplementary Fig. 5l). We concluded that Ybx1 suppresses H3K27me3 distribution genome-wide.

**Ybx1 preferentially binds PRC2-bound genomic features**. To investigate possible mechanistic activities of Ybx1, we examined genomic features PRC2 preferentially binds. Nearly all Ybx1-bound regions (99.6%) and TSSs (99.8%) in categories 1–4 overlapped CpG islands (Fig. 5a, Supplementary Fig. S6a). However, 44.9% of Ybx1-bound regions and 30.5% of Ybx1-bound TSSs in category 5 overlapped CpG islands (Fig. 5a, Supplementary Fig. S6a). This finding suggests that Ybx1 binds CpG islands.

We investigated whether Ybx1 binds to bivalent regions, which are co-occupied by H3K27me3 and H3K4me3. Using the 1-nt overlap criterion, we determined that Ybx1 bound to 4344/7953 (54.6%) of bivalent regions in NPCs (Fig. 5b). This strong enrichment prompted us to determine whether *Ybx1* KO led to differential expression of bivalent genes. Compared with sibling control NPCs, 179 bivalent genes were downregulated and 212 bivalent genes were upregulated in *Ybx1*-KO NPCs (Supplementary Fig. S6b), suggesting Ybx1 affects the expression of bivalent genes. As transcription at bivalent genes is generally low and not expected to be downregulated, we examined RNA-seq profiles of some of the 179 bivalent genes. At many of these genes, RNA-seq reads were restricted to the TSSs in control NPCs and reduced further in *Ybx1*-KO NPCs.

We next examined a potential association of Ybx1 with enhancer elements, including active enhancers co-occupied by H3K4me1 and H3K27ac and poised enhancers co-occupied by H3K4me1 and H3K27me3. We found that 18.4% (4235/23070) and 14.5% (3352/23070) of Ybx1-bound regions overlapped active and poised enhancers, respectively. About 12.1% (4095/33934) of active enhancers were bound by Ybx1. We found that 29.1% (3782/9229) of poised enhancers were bound by Ybx1; this higher enrichment suggests a functional association. As our data suggest that Ybx1 suppresses H3K27me3 to promote H3K4me3 and gene expression, we focused on the 271 Ybx1-bound active enhancers (little to no H3K27me3) in control NPCs that became

downregulated in *Ybx1*-KO NPCs (Supplementary Fig. S6c). Our data suggest that Ybx1 binds enhancers to promote the expression of genes for nervous system development, synapses, axonogenesis, axon guidance, and dendrite morphogenesis.

**Ybx1 promotes PRC2 binding to chromatin but inhibits H3K27me3**. We identified a consensus DNA motif for Ybx1 with $p < 1e − 6$ (Supplementary Fig. S6d) in promoters of genes that were upregulated in *Ybx1*-KO NPCs, consistent with our observation that known Ybx1 target genes were upregulated in *Ybx1*-KO NPCs (Fig. 2e). On the other hand, of the 382 Ybx1-bound genes that were upregulated in *Ybx1*-KO NPCs, 132 are known to be bound by PRC2 ($p = 2.3e − 18$ in Suz12 ChIP-seq datasets using ChEA of Enrichr[44]), supporting that Ybx1 binds and regulates PRC2-bound genes.

We next examined how Ybx1 affects the expression and chromatin binding of PRC2. The expression of PRC2 subunits did not differ between control and *Ybx1*-KO NPCs (Supplementary Fig. S6e, f). We performed CUT&RUN-seq of the enzymatic subunits of PRC2, Ezh2/1, in sibling control and *Ybx1*-KO NPCs (by Sox2-GFP FACS). Ezh2/1 distribution at the *Hox* A cluster paralleled H3K27me3 distribution and remained unchanged in *Ybx1*-KO NPCs (Supplementary Fig. S6g). Moreover, Ezh2/1 occupancy was significantly lower at the forebrain lineage genes *Foxg1*, *Neurod2*, and *Satb2* (Fig. 5c) and cell proliferation genes *Lin28a* and *Hmga2* (Supplementary Fig. S6h), whereas H3K27me3 levels were increased. Also, Ezh2/1 localized to an ectopic (only in *Ybx1*-KO but not in control NPCs) PRC2-bound region that showed increased H3K27me3 in *Ybx1*-KO NPCs (Supplementary Fig. S6i). We concluded that at these genes, PRC2 binding decreased but H3K27me3 levels increased in *Ybx1*-KO NPCs.

Extending our analysis genome-wide with criteria of FDR-corrected $p < 0.05$, we identified 43,469 Ezh2/1-bound regions in control NPCs and 10,902 Ezh2/1-bound regions in *Ybx1*-KO NPCs. Using this criterion, 15,036/43,469 (34.5%) Ezh2/1-bound regions were found to overlap with Ybx1-bound regions in control NPCs. With a more relaxed criterion of $p < 0.05$, 29,068/43,469 (66.9%) Ezh2/1-bound regions were found to overlap Ybx1-bound regions. We generated heat maps to compare Ybx1 and Ezh2/1 distribution at all Ybx1-bound regions (Fig. 5d) or all Ezh2/1-bound regions (Fig. 5e) and showed Ezh2/1 and Ybx1 binding patterns were highly similar and likely overlapped by more than 66.9% in wild-type NPCs.

Ezh2/1 binding at most regions genome-wide was reduced in *Ybx1*-KO NPCs compared to control NPCs (Fig. 5e). Ezh2/1 gained binding at ectopic regions (specific to *Ybx1*-KO in Fig. 5e). We could not attribute functional significance to ectopic Ezh2/1-bound regions as they were not associated with differentially expressed genes in *Ybx1*-KO NPCs. H3K27me3 levels increased at all regions with reduced Ybx1 and Ezh2/1 binding in *Ybx1*-KO NPCs (H3K4me3 remained unchanged; Fig. 5e). At the ectopic Ezh2/1-bound regions in *Ybx1*-KO, H3K27me3 levels increased and H3K4me3 levels remained undetectable (Fig. 5e). Collectively, our data suggest that Ybx1 binding promotes PRC2 binding at many sites while restraining its activity including H3K27 methylation. This balancing/fine-tuning of PRC2 binding and histone modifications likely facilitates precise spatiotemporal gene regulation required for neural development.

**PRC2 complex physically binds YBX1**. We confirmed the JARID2–YBX1 binding with new IP-mass spectrometry datasets using a different JARID2 antibody (R&D Systems AF6090) in mouse neural stem cells NE-4C (Supplementary Fig. S7a, b). We validated the specificity of the antibody (Supplementary

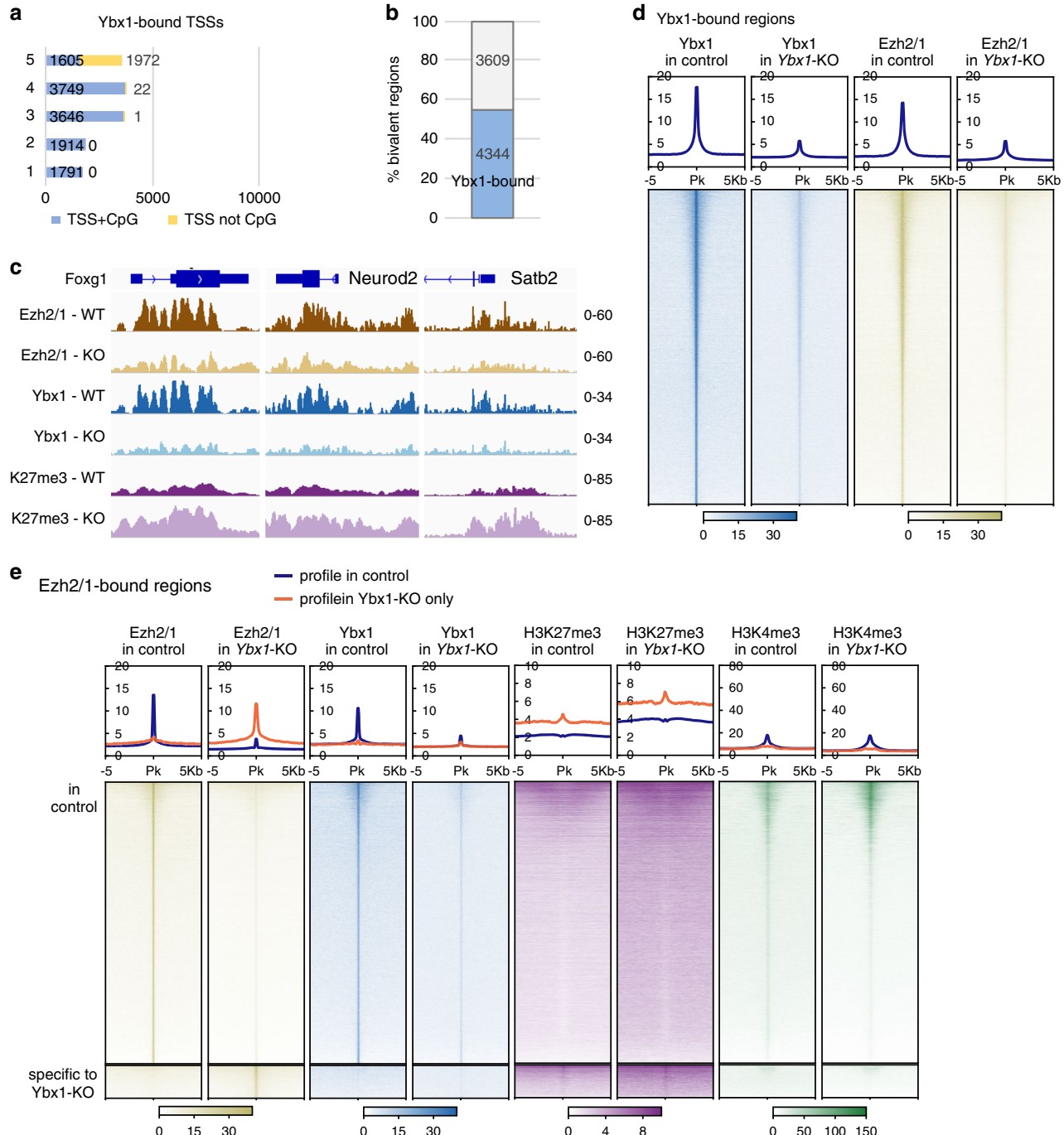

**Fig. 5 Ybx1 is required for appropriate levels of PRC2 binding to chromatin. a** Proportions of Ybx1-bound TSSs (categories 1–5 based on H3K27me3 and H3K4me3 distribution in Fig. 4) that overlapped CpG islands. **b** Percent of bivalent (co-occupied by H3K27me3 and H3K4me3) regions that overlap Ybx1-bound regions. **c** Ezh2/1 and Ybx1 CUT&RUN-seq tracks and H3K27me3 ChIP-seq at *Foxg1*, *Neurod2*, and *Satb2* gene loci in *Ybx1*-KO and sibling control NPCs. **d** Heat maps of Ybx1 and Ezh2/1 distribution within 5 kb of Ybx1 CUT&RUN-seq peaks. **e** Heat maps of Ezh2/1, Ybx1, H3K27me3, and H3K4me3 within 5 kb of Ezh2/1 CUT&RUN-seq peaks.

Figs. S2a, S7c, d). Using the hESC nuclear extract, we then performed additional co-IP experiments using benzonase treatment (eliminates nucleic acids for possibly mediating the JARID2–YBX1 interaction), high-stringency washes, and 2 other JARID2 antibodies (Fig. 6a and Supplementary Fig. S7e–g). The depletion of Ybx1[27,29] and PRC2 subunits[11,15–17] results in neurodevelopmental defects, so we examined their interactions in NPCs. We showed that that JARID2, SUZ12, and YBX1 interact in hNPCs (Fig. 6c) and

further showed that Jarid2 and Ybx1 interact with each other and PRC2 in mouse neural tubes at E13.5 (Fig. 6d). The enrichment of JARID2 in YBX1 co-IP and YBX1 in SUZ12 co-IP were low (Fig. 6c, d), suggesting substoichiometric inter-action. Subcellular fractionation showed that Ybx1 protein localizes to the cytoplasm, nucleoplasm, and on chromatin (Supplementary Fig. S7h). Pull-down showed that JARID2 protein binds to YBX1 protein (Fig. 6e) and that amino acids 1-104 of YBX1 were required for binding to JARID2 (Fig. 6f).

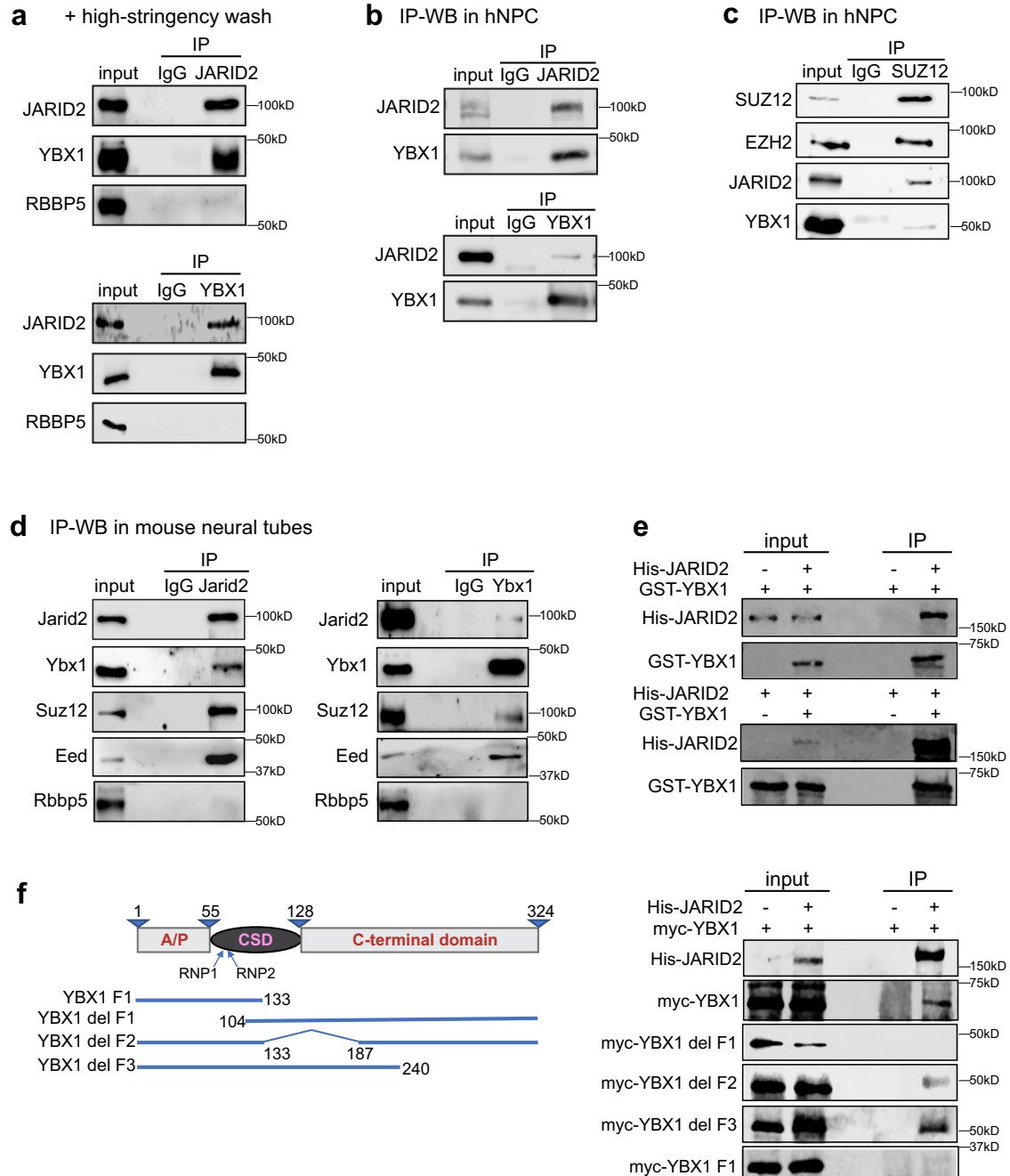

**Fig. 6 PRC2 complex binds to Ybx1 in human stem cells and mouse neural tubes.** WB analysis of IgG, JARID2, and YBX1 co-IP of **a** the hESC nuclear extract with high-stringency washes and **b** the hNPC nuclear extract. **c** WB analysis of IgG and SUZ12 co-IP of the hNPC nuclear extract. **d** WB analysis of IgG, Jarid2, and Ybx1 co-IP of the mouse neural tube nuclear extract. WB analysis of His-JARID2 and **e** GST-YBX1 or **f** myc-tagged YBX1 protein fragments. Source data are provided in Source Data file.

**Ybx1-mediated regulation of NPCs involves inhibition of H3K27me3**. If Ybx1 inhibits H3K27me3 to affect gene expression and, and in turn, NPC self-renewal and differentiation, we predict that H3K27me3 reduction in *Ybx1*-KO cells should rescue their defects. To test this, we treated *Ybx1*-KO NPCs with a specific chemical inhibitor of PRC2, GSK126[45], and examined gene expression, proliferation, neurosphere formation, and differentiation (Fig. 7a). Treatment with 100 or 500 nM of GSK126 reduced the global levels of H3K27me3 in *Ybx1*-KO and sibling control NPCs, as expected (Fig. 7b). Compared with dimethyl sulfoxide (DMSO) treatment, GSK126 treatment of *Ybx1*-KO NPCs significantly

increased the expression of assayed forebrain lineage genes (Fig. 7c and Supplementary Fig. S8a) and decreased the expression of most of the assayed midbrain and hindbrain lineage genes (Supplementary Fig. S8b). GSK126 treatment of control NPCs led to decreased expression of some forebrain lineage genes and increased expression of some midbrain and hindbrain lineage genes (Supplementary Fig. S8c, d). These data suggest that PRC2 inhibition in *Ybx1*-KO NPCs leads to the reactivation of forebrain lineage genes and suppression of midbrain and hindbrain lineage genes.

We also examined how GSK126 treatment of *Ybx1*-KO NPCs affects the expression of *Cdkn1a* and *Reln*, which are known targets

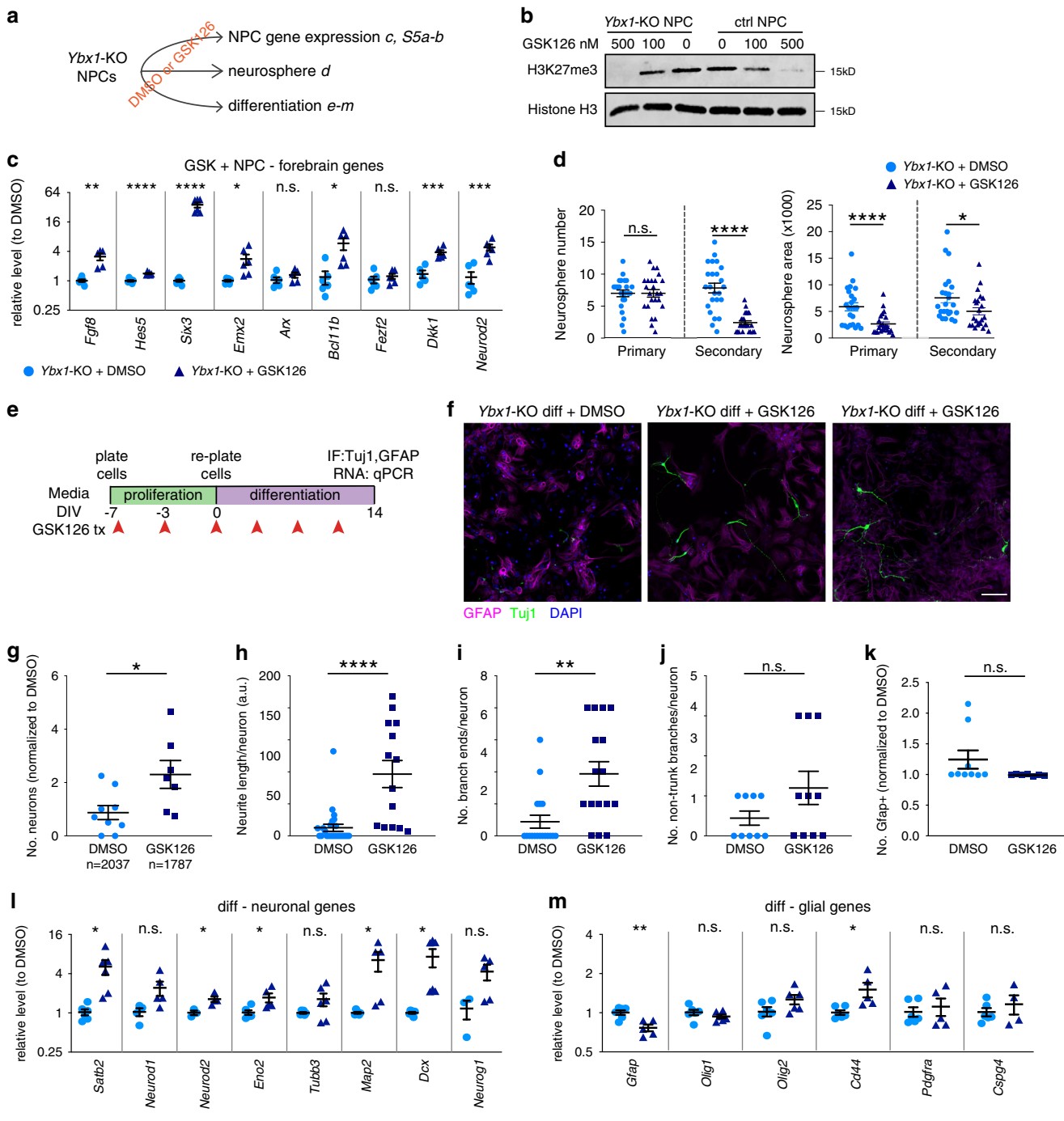

**Fig. 7 PRC2 inhibition partially restores the appropriate gene expression, self-renewal, and neuron differentiation in *Ybx1*-KO NPCs. a** Diagram summarizing assays and the corresponding data panels. **b** WB analysis of acid-extracted histones from *Ybx1*-KO and control NPCs treated with DMSO or GSK126 after 4 days. **c** RT-qPCR with TaqMan assays of *Ybx1*-KO NPCs treated with DMSO or 500 nM GSK126 after 4 days. *n* = 6. **d** Number and area of primary and secondary neurospheres formed by *Ybx1*-KO NPCs pretreated with DMSO or 500 nM GSK126 for 2 days. *n* = 24 images/genotype. **e** Diagram detailing in vitro differentiation of *Ybx1*-KO NPCs treated with DMSO or 500 nM GSK126. Red arrowheads indicate the days at which DMSO or 500 nM GSK126 was replenished. **f** IF analysis of *Ybx1*-KO cells treated with DMSO or 500 nM GSK126 at day 14 of differentiation. Quantification of **g** relative number of Tuj1-positive neurons, **h** neurite length, **i** number of branch points in axons, **j** number of non-trunk branches (not connected to an axon or dendrite), and **k** Gfap-positive (glial marker) cells in *Ybx1*-KO differentiating cells treated with DMSO or 500 nM GSK126 (*n* = 9 DMSO and 7 GSK126 images with cell numbers indicated). RT-qPCR with TaqMan assays of **l** neuronal genes and **m** glial genes in *Ybx1*-KO differentiating cells treated with DMSO or 500 nM GSK126. *n* = 6, except *n* = 3 ctrl *Neurod1* and *Neurog1*. Data are presented as mean ± SEM for **c**, **d**, and **g**–**m**. *P* values by two-tailed unpaired *t* test are indicated. n.s. indicates not significant. *, **, ***, and *** indicate *P* < 0.05, 0.01, 0.001, and 0.0001, respectively, by one-way ANOVA or unpaired *t* test with Holm–Sidak correction. Source data are provided in Source Data file. *P* values: 7**c**(Fgf8)-0.002, (Hes5)-<0.0001, (Six3)-<0.0001, (Emx2)-0.02, (Bcl11b)-0.02, (Dkk1)-0.0002 (NeuroD2)-0.0009; 7**d**(secondary #)-<0.0001, 7**d**(primary area)-<0.0001, 7**d**(secondary area)-0.03; 7**g**-0.02; 7**h**-<0.0001; 7**i**-0.001; 7**l**(Satb2)-0.01, (NeuroD2)-0.03, (Eno2)-0.02, (Map2)-0.02, (Dcx)-0.02; 7**m**(Gfap)-0.004, (Cd44)-0.02.

of PRC2 in neural development[46–48]. We found that *Cdkn1a* and *Reln* expression was significantly downregulated in *Ybx1*-KO compared with sibling control NPCs (Supplementary Fig. S8e), consistent with the loss of Ybx1-mediated inhibition of PRC2. GSK126 treatment led to the upregulation of both *Cdkn1a* and *Reln* in wild-type and *Ybx1*-KO NPCs, as expected due to inhibition of PRC2-mediated gene suppression (Supplementary Fig. S8f).

We examined whether GSK126 affects self-renewal and differentiation of *Ybx1*-KO NPCs. Compared with DMSO treatment, GSK126 significantly reduced the number and growth of *Ybx1*-KO neurospheres through serial passages (Fig. 7d). These findings suggest that PRC2 inhibition reduces the self-renewal of *Ybx1*-KO NPCs. For the in vitro differentiation assay, we treated *Ybx1*-KO NPCs with GSK126 before and during differentiation (Fig. 7e). After 14 days of culture in differentiation media, GSK126-treated *Ybx1*-KO cells formed more Tuj1-positive neurons that had longer neurites and more neurite branching (Fig. 7f–j). GSK126 treatment did not significantly affect the number of Gfap-positive glial cells (Fig. 7k), suggesting that glial differentiation was unaffected. GSK126 treatment led to increased expression of some neuronal genes and less effect on glial genes in differentiating *Ybx1*-KO cells (Fig. 7l, m). GSK126 treatment of differentiating control cells largely had modest to little effect on the expression of most neuronal and glial genes (Supplementary Fig. S8g). *Cdkn1a* expression was little affected, and *Reln* was downregulated (Supplementary Fig. S8h). In a separate differentiation assay, YBX1 re-expression by lentiviral transduction was not sufficient to alter the epigenetic programs and restore neuronal differentiation in *Ybx1*-KO (Supplementary S9a). Our data suggest that PRC2 inhibition in *Ybx1*-KO NPCs partially reversed cellular defects and restored neuronal differentiation.

We examined whether depletion of Eed, a PRC2 subunit, affects gene expression or differentiation of *Ybx1*-KO NPCs (purified by Neurofluor™ CDr3 FACS). We used *Nestin* promoter-driven Cre recombinase[49] to excise floxed *Eed*[50] in vivo (Fig. 8a). WB analysis confirmed conditional depletion of Eed and reduction of H3K27me3 levels in *Ybx1*-KO, *Eed*-conditional KO NPCs (referred to as *Ybx1*-KO; *Eed*-cKO) (Fig. 8a). Compared with *Ybx1*-KO NPCs, *Ybx1*-KO; *Eed*-cKO NPCs had significantly higher expression of forebrain lineage genes and lower expression of most midbrain and hindbrain lineage genes (Fig. 8b and Supplementary Fig. S9b, c). These results suggest that, similar to GSK126 treatment, Eed depletion in *Ybx1*-KO NPCs restores brain regionalization.

During in vitro differentiation, *Ybx1*-KO;*Eed*-cKO cells yielded significantly more neurons with longer neurites than those in *Ybx1*-KO cells (Fig. 8c–e). Eed depletion did not significantly affect neurite branching or the number of Gfap-positive glial cells (Fig. 8f–h), suggesting that neuron maturation and glial differentiation was less affected. Neither *Cdkn1a* or *Reln* expression was markedly affected by Eed depletion in *Ybx1*-KO differentiation (Supplementary Fig. S9d). Compared with *Ybx1*-KO, *Ybx1*-KO;*Eed*-cKO cells had significantly higher expression of some neuronal genes and decreased expression of some glial genes (Fig. 8i and Supplementary Fig. S9e). Altogether, our results suggest that in NPCs, Ybx1 fine-tunes PRC2 activities to mediate brain regionalization, modulate self-renewal, and promote neuronal differentiation.

## Discussion
The multifunctional role of Ybx1 in different cellular processes to promote cancer progression is well established[22,23]. However, Ybx1 has been mostly studied in cancer cell lines, and the physiological role of Ybx1 in brain development is relatively less understood. Our study advances the understanding of PRC2 regulation in brain development by uncovering that Ybx1 binds to PRC2 to promote PRC2 binding and proper genomic distribution, but inhibits H3K27me3 levels in NPCs. Via PRC2, Ybx1 likely influences gene

expression for NPC self-renewal modulation and neuronal differentiation (Fig. 8j). Ybx1 depletion in the brain leads to reduced forebrain specification concomitant with expanded midbrain and hindbrain regions; these phenotypes are the opposite of those in brains with Ezh2 depletion[51]. We propose the model that Ybx1 regulates PRC2 to mediate brain regionalization and the decision between self-renewal and neuronal differentiation. This model is strongly supported by the finding that chemical and genetic inhibition of PRC2 partially rescues the defective phenotypes of *Ybx1*-KO NPCs. Taken together, these findings point to a physical and functional interaction between Ybx1 and PRC2, such that PRC2 is locked and loaded at co-bound genes in NPCs. An aberrant decrease in PRC2 activity leads to forebrain expansion, whereas an aberrant increase in PRC2 activity in the absence of Ybx1 leads to forebrain reduction. Overall, our data suggest that Ybx1 optimizes PRC2 activity and promotes balanced brain development in vivo.

Because PRC2 and Ybx1 both have cell and development context-dependent influences, their interactions likely differ in forebrain versus mid/hindbrain. The crucial requirement of PRC2 and Ybx1 for spatiotemporal developmental programs, despite their ubiquitous expression, suggests that additional factors within this network provide developmental specificity in neural stem and progenitor cells.

We used in vitro assays for their multiple advantages of validating in vivo studies, extending the investigation of *Ybx1*-KO beyond the stage of embryonic lethality, showing that PRC2–Ybx1 participates in a cell-intrinsic mechanism to regulate NPCs. Although Ybx1 promotes the proliferation of medulloblastoma or glioblastoma cancer cells[25,26], it suppresses the proliferation and self-renewal of NPCs. This difference could be due to (i) the dosage effect of Ybx1 from complete genetic depletion in our study versus partial depletion by small interfering RNAs in other studies; (ii) varying activities of Ybx1 in different cell types; and (iii) developmental or pathological contexts of NPCs vs. cancer cells. Indeed, we found that Ybx1 suppresses the proliferation of NPCs but promotes the proliferation of MEFs. Future studies on cell type-specific regulation of Ybx1 are needed to fully understand its diverse mechanisms. The PRC2 variant complexes also have cell type-specific molecular activities and influences[52] that require further study, including whether Ybx1 regulates PRC2 in other stem and progenitor cells, or in cancer.

The activities of Ybx1 detailed herein uncover a fundamental mechanism that is crucial for appropriate chromatin dynamics and gene activation during development. Furthermore, Ybx1 likely reaches beyond H3K27 methylation, as it has been implicated in influencing P300/CBP and H3K27 acetylation[53]. We hypothesize that Ybx1 reduces H3K27me3 in vivo by inhibiting PRC2 methyltransferase activity, perhaps by reducing complex integrity or stability, and/or binding to RNAs. Alternative mechanistic scenarios include Ybx1 influencing gene activators or chromatin remodelers in order to reduce H3K27me3 at different loci. Future identification of other Ybx1 binding factors, as well as extensive biochemical, genetic, and genomic characterization will be required to pinpoint how Ybx1 affects PRC2 and potentially the activities of other chromatin modifiers.

Patterning and regionalization of the embryonic brain occur in a short developmental time window of 3 days[54]. Polycomb group proteins have emerged as key epigenetic factors that integrate cues from signaling pathways and the transcriptional circuitry to affect gene expression programs underlying brain patterning and regionalization[2–4]. Our findings suggest that Ybx1 reduces the level of H3K27me3 deposited by PRC2 in NPCs. As ChIP-seq is a population-based assay, H3K27me3 reduction potentially occurs in a subset of NPCs. Nevertheless, this reduction appears to be genome-wide and coincides with the Ybx1-dependent expression of PRC2 target genes involved in forebrain lineage specification

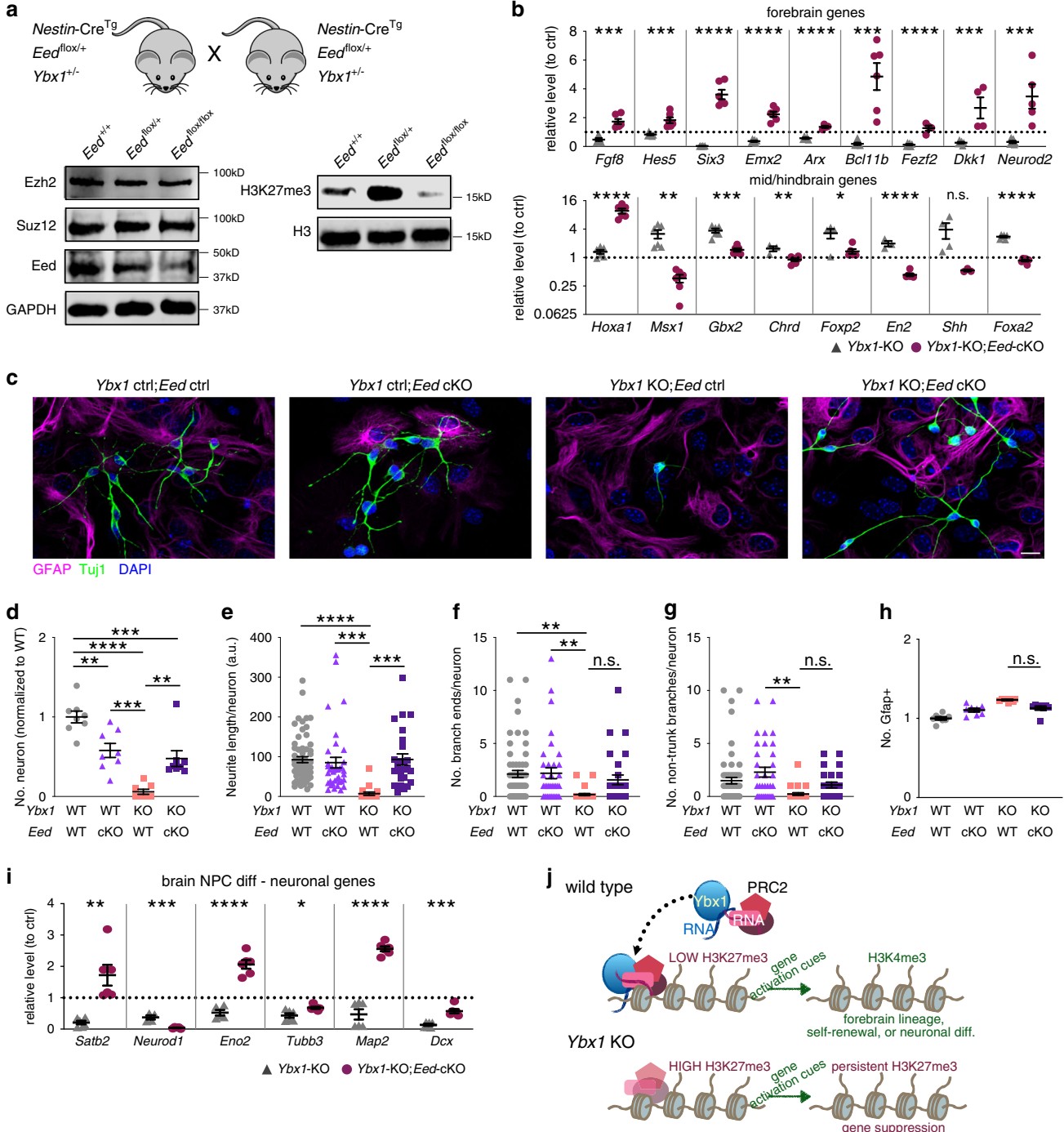

**Fig. 8 Depletion of PRC2 subunit Eed partially restores the appropriate gene expression and neuron differentiation in *Ybx1*-KO NPCs. a** Genetic cross scheme to obtain embryos at E13.5 with *Ybx1*-KO or *Ybx1-Eed*-cKO by *Nestin*:Cre. WB analysis of *Ybx1*-KO NPCs with indicated *Eed* alleles (all embryos are *Nestin:Cre^Tg*). **b** RT-qPCR with TaqMan assays of *Ybx1*-KO and *Ybx1-Eed*-cKO. Relative levels were displayed as their standardization to control NPCs. Dotted line indicates the level of gene expression in control cells. *n* = 6 except *n* ≥ 3 for *Chrd*, *En2*, and *Shh*. **c** IF analysis of cells with indicated genotypes at day 14 of differentiation. Quantification of **d** relative number of Tuj1-positive neurons, **e** neurite length, **f** number of branch points in axons, **g** number of non-trunk branches (not connected to an axon or dendrite), and **h** Gfap-positive (glial marker) cells in differentiating cells (*n* = 8 images). **i** RT-qPCR with TaqMan assays of neuronal genes in cells after 14 days of differentiation (*n* = 6). Dotted line indicates the level of gene expression in control cells. **j** Model of Ybx1 activities and its influence over PRC2 to ensure spatiotemporal precision in gene activation. Data are presented as mean ± SEM for **b**, **d**–**i**. *P* values by two-tailed unpaired *t* test are indicated for **b** and **i** and by ordinary one-way ANOVA followed by Tukey's post hoc test in **d**–**h**. n.s indicates not significant. *, ***, and **** indicate *P* < 0.05, 0.001, and 0.0001, respectively. Source data are provided in Source Data file. *P* values: 8**b**(Fgf8)-0.0001, (Hes5)-0.0007, (Six3)-<0.0001, (Emx2)-<0.0001, (Arx)-<0.0001, (Bcl11b)-0.0006, (Fezf2)-<0.0001, (Dkk1)-0.003, (NeuroD2)-0.003, (Hoxa1)-<0.0001, (Msx1)-0.002, (Gbx2)-0.0002, (Chrd)-0.005, (Foxp2)-0.02, (En2)-<0.0001, (Foxa2)-<0.0001; 8**d**(Ybx1 WT-Eed cKO)-0.004, (Ybx1 WT-Ybx1 KO)-<0.0001, (Ybx1WT-dKO)-0.0003, (Eed cKO-Ybx1 KO)-0.0003, (Ybx1 KO-dKO)-0.004; 8**e**(Ybx1 WT-Ybx1 KO)-<0.0001, (Eed cKO-Ybx1 KO)-0.0003, (Ybx1 KO-dKO)-0.0002; 8**f**(Ybx1 WT-Ybx1 KO)-0.0042, (Eed cKO-Ybx1 KO)-0.008; ;8**g**(Eed cKO-Ybx1 KO)-0.0013; 8**i**(Satb2)-0.001, (NeuroD1)-0.0009, (Eno2)-<0.0001, (Tubb3)-0.01, (Map2)-<0.0001, (Dcx)-0.0002.

and neuronal differentiation. Overall, our findings suggest that modest to low levels of H3K27me3, regulated by Ybx1, enable but also restrain the activation of PRC2 target genes during this short developmental time window.

## Methods

**Buffers.** PBS: 137 mM NaCl, 2.7 mM KCl, 10 mM $Na_2HPO_4$, 1.8 mM $KH_2PO_4$ (pH 7.4). PBS-T: PBS with 0.1% Triton X-100. HEPM: 25 mM HEPES (pH 6.9), 10 mM EGTA, 60 mM PIPES, 2 mM $MgCl_2$. IF blocking solution: 1/3 Blocker Casein (ThermoFisher 37528), 2/3 HEPM with 0.05% TX-100. Buffer A: 10 mM HEPES (pH 7.9), 10 mM KCl, 1.5 mM $MgCl_2$, 0.34 M sucrose, 10% glycerol. Buffer B: 3 mM EDTA, 0.2 mM EGTA. Buffer D: 400 mM KCl, 20 mM HEPES, 0.2 mM EDTA, 20% glycerol. FACS antibody staining solution: PBS containing 0.5% Tween 20, 1% bovine serum albumin, and 0.5 μg/μL RNase. CUT&RUN Binding buffer: 20 mM HEPES-KOH (pH 7.9), 10 mM KCl, 1 mM $CaCl_2$, and 1 mM $MnCl_2$. CUT&RUN Wash buffer: 20 mM HEPES pH 7.5, 150 mM NaCl, 0.5 mM spermidine, protease inhibitor (PI). CUT&RUN Digitonin block buffer: Wash buffer with 2 mM EDTA, 0.05% digitonin. CUT&RUN Stop Buffer: Into 1 mL of Digitonin block buffer, add 5 μL of 10 mg/mL RNase A and 133 μL of 15 mg/mL GlycoBlue. High-stringency wash buffer: 15 mM Tris-HCl (pH 7.5), 5 mM EDTA (pH 8.0), 2.5 mM EGTA, 1% Triton X-100, 1% sodium deoxycholate, 0.1% SDS, 120 mM NaCl, 25 mM KCl.

**Tissue culture cells.** H9/WA09 (WiCell) cells were grown on Matrigel with reduced growth factors (ThermoFisher 354230) in mTeSR1 medium (STEMCELL Technologies 85850) at 37 °C and 5% $CO_2$. NE4-C (ATCC CRL-2925; mouse neural stem cell line) cells were grown on Poly-L-lysine (10 μg/mL; Sigma P9155) in Eagle's Minimum Essential Medium (ATCC 30-2003) supplemented with 2 mM L-Glutamine (Invitrogen 25030-081) and 10% fetal bovine serum (Fisher Scientific 16000044) at 37 °C and 5% $CO_2$.

**Antibodies.** Supplementary Table S1 lists all antibodies and conditions used in this study.

**Co-immunoprecipitation.** For co-immunoprecipation, subcellular fractionation was performed as above, but Buffer D + PI + DTT was substituted for Buffer B and the nuclear fraction was diluted with an equal volume $H_2O$ + PI + DTT. Approximately 1.5 mg of the nuclear extract was incubated with primary antibody overnight at 4 °C. Extract and antibody were added to protein A and protein G Dynabeads™ (ThermoFisher 10002D and 10004D) for 5 h at 4 °C, washed with PBS-T, and eluted with 0.1 M glycine (pH 2.3). Eluates were neutralized with 1/10 volume of 1.5 M Tris buffer (pH 8.8). For benzonase, nuclear extracts were treated with 125 U/mL benzonase (Sigma) for 30 min at RT before addition of antibody and subsequent IP. High-stringency wash buffer was used in replacement for PBS-T washes in indicated figures.

**Subcellular fractionation.** Human and mouse NPCs were incubated in a 2× volume of Buffer A + PI + DTT for 5 min on ice to obtain the cytoplasmic fraction. After centrifugation at 1750 g for 2 min at 4 °C, the nuclei pellet was washed in 1× volume Buffer A and subsequently incubated for ~45 min in Buffer B + PI + DTT at 4 °C with rotation to obtain the nuclear fraction. Histones were acid extracted on pelleted chromatin with 0.1 N HCl at 4 °C O/N. Acid was neutralized with 1/10 volume of 1.5 M Tris buffer (pH 8.8). For whole-cell extracts, cell pellets were incubated directly in Buffer D + PI + DTT.

**Mass spectrometry.** For each of Supplementary Fig. S1d or S7b, we analyzed three biological replicate samples of JARID2 and IgG IP from the nuclear extract of H9 or NE-4C, respectively. Samples were run on a short gel. Sample slices were digested overnight at 37 °C, acidified and the peptides were extracted in Acetonitrile. Extracts were dried down in a speed vacuum and reconstituted in 5% Formic acid. Digested peptides were loaded on a nanoscale capillary reverse phase C18 column (75 id, 10 cm) by a HPLC system (Thermo Ultimate 3000 or EASY-nLC 1000). Buffer A was 0.2% Formic acid and Buffer B was 70% Acetonitrile; 0.2% Formic acid. The peptides were eluted by increasing organic from 12–70% over a 90-min liquid chromatography gradient. The eluted peptides were ionized by electrospray ionization, and detected by an inline mass spectrometer (Thermo LTQ Orbitrap Elite). The mass spectrometer was operated in data-dependent mode with a survey scan in Orbitrap (240,000 resolution, $1 \times 10^6$ AGC target and 200 ms maximal ion time) and 20 low resolution MS/MS scans in the ion trap (CID, 2 $m/z$ isolation width, Normalized collision energy 35, AGC target $7 \times 10^4$, and 250 ms maximal ion time) for each cycle. The raw data were searched against the UniProt mouse database concatenated with a reversed decoy database for evaluating false discovery rate. Database searches were performed using the Sequest[55] v.28 (rev. 12) search engine. Searches were performed using a 25 ppm mass tolerance for precursor and 0.5 Da for product ions, fully tryptic restriction with two maximal missed cleavages, three maximal modification sites, and the assignment of b, and y ions. Carbamidomethylation of Cysteine (+57.02146 Da) was used for static

modifications and Met oxidation (+15.99492 Da) was considered as a dynamic modification. All matched MS/MS spectra were filtered by mass accuracy and matching scores to reduce protein false discovery rate to <1%. Finally, all proteins identified in one gel lane were combined together. The total number of spectra, namely spectral counts (SC), matching to individual proteins may reflect their relative abundance in one sample after the protein size is normalized[56]. The abundance values (spectral count ×50 kD/protein size kD) of proteins were used to compared between the replicate datasets by T-test.

**Western blotting.** Equal amounts of nuclear extracts were separated by SDS-PAGE and transferred onto a nitrocellulose membrane (Bio-Rad). Membranes were blocked with 2% BSA in HEPM, incubated in primary antibodies (HEPM containing 1% BSA and 0.1% Triton X-100) overnight at 4 °C, washed in PBS-T, incubated in IRDye®-conjugated secondary antibodies (LI-COR) or Clean-blot IP detection reagent (Life Technologies 21230), followed by SuperSignal™ West Pico PLUS Chemiluminescent Substrate (ThermoFisher 34577), and imaged on an Odyssey® Fc imaging system (LI-COR). Signals were quantitated with the Image Studio™ software (version 1.0.14; LI-COR). Student's $t$ test was used for statistical analyses.

**Animals.** All mice were group-housed and maintained in the Animal Resource Center at St. Jude Children's Research Hospital under protocols approved by the Institutional Animal Care and Use Committee. We complied with all relevant ethical regulations for animal testing and research. Mice of both sexes were used for all experiments, and they were analyzed at multiple ages (E12.5–E14.5), as described in the text and legends for each figure. All samples were analyzed relative to littermates. For *Eed*-flox experiments, all animals were crossed to *Nestin*-Cre and genotypes are indicated in figures. Details of mouse strains are as follows:

*Nestin*-Cre: B6.Cg-Tg(Nes-Cre)1Kln/J, described in Tronche et al.[49].
*Eed*-flox: B6;129S1-*Eed*tm1Sho/J, described in Yu et al.[50].
*Ybx1*: *Ybx1*tm1Ley/*Ybx1*tm1Ley, described in Lu et al.[27].
*Sox2*-eGFP: B6;129S1-*Sox2*tm1Hoch/J, described in Arnold et al.[57].
For genotyping, primers are following. Ybx1 WT: F-AGGAACGGATACGGT TTCATCA; R-AGCGGGTCACATTCTTACATAG (54 °C, 30 s). Ybx1 null: F- AG GAACGGATACGGTTTCATCA; R- TGAGACGTGCTACTTCCATTT (54 °C, 30 s). Eed flox: F-GGCCCACATAGGCTCATAGA; R- CTACGGGCAGGAGGA AGAG (55 °C, 30 s). Nestin-Cre: F-ATGCCCAAGAAGAAGAGGAAGGT; R-GA AATCAGTGCGTTCGAACGCTAGA (56 °C, 30 s). Actb: F- ATGTCACGCACG ATTTCCCT, R- TCCCGGGTAACCCTTCTCTT (any AT, 45 s). Sox2-eGFP: F- C GTAAACGGCCACAAGTTCA, R-CTCAGGTAGTGGTTGTCGGG (56 °C, 40 s). Reaction products: 354 bp (*Ybx1*WT), 477 bp (*Ybx1*null); 200 bp (*Eed*WT), 400 bp (*Eed*flox); 447 bp (Nestin-CreTg); 719 bp (Actb); 543 bp (*Sox2*eGFP).

**Isolation, culture, and differentiation of Mouse Neural Progenitor Cells (mNPCs).** Tissue culture plates for adherent culture were prepared by coating them with 10 μg/mL poly-D-lysine (MilliporeSigma P7280) for either 2 h at room temperature (RT) or at 4 °C O/N (overnight). Plates were rinsed twice with 1× PBS and replaced with 10 μg/mL laminin (MilliporeSigma L2020) for 2 h at RT. As laminin is highly sensitive to drying, it was removed well-by-well and immediately replaced with sufficient culture medium to cover the surface of the well.

Cultures of mouse neural progenitor cells (mNPCs) were derived from E12.5–E13.5 mice as follows. Neural tubes were dissected from embryos in ice-cold Advanced DMEM culture media (ThermoFisher 2491015) and dissociated with 1/3 volume of Type II collagenase (10 mg/mL, Worthington LS004176) for 10 min at 37 °C. The tissue was washed once in 1× PBS and incubated in 500 μL 0.25% Trypsin-EDTA (ThermoFisher 25200056) for 5 min at 37 °C. Trypsinization was halted using 500 μL DMEM + 10% fetal bovine serum, and cells were mechanically dissociated by trituration and then pelleted by centrifugation at 1000 g for 3 min. Cells were resuspended in 1 mL of PBS and filtered through a 40 μm filter (ThermoFisher 22363547) to obtain single cells. NPCs were isolated by fluorescence-activated cell sorting (FACS) using Sox2-eGFP or NeuroFluor CDr3 (STEMCELL Technologies, #01800), resuspended in complete culture medium, and grown on ultra-low attachment six-well plates. After sphere formation, spheres were dissociated with Accutase (STEMCELL Technologies 07920) for 5 min in the incubator and single cells plated on coated plates.

Cells were grown in NeuroCult™Proliferation Media (STEMCELL Technologies 05702) supplemented with 10 μg/mL rhEGF (STEMCELL Technologies 78006) to a final concentration of 20 ng/mL. For routine passaging, cells were grown to 75–80% confluency and then dissociated using Accutase for 5 min in the incubator. Cells were then diluted with an equal volume of growth medium and dissociated by trituration. Cells were pelleted as described above and resuspended in fresh culture medium to plate at recommended densities. For differentiation, cells were plated at recommended densities on poly-D-lysine/laminin–coated six-well plates or Matrigel (ThermoFisher 354240)–coated eight-well chamber slides (EMD Millipore PEZGS0816) and left O/N. Cells were washed once in Neurocult™Proliferation Media without EGF and incubated in Neurocult™. Differentiation media (STEMCELL Technologies 05704) for indicated times are detailed in figure legends.

**BrdU administration.** Mice were administered 5-bromo-2′-deoxyuridine (BrdU, MilliporeSigma) reconstituted in sterile PBS by intraperitoneal injection at a dose of 50 mg/kg. For cell culture, cells were treated with 10 μM BrdU for indicated times at 37 °C.

**Immunofluorescence.** Cells and tissues were blocked with IF blocking solution for 2–3 h at RT and primary antibodies (diluted in blocking buffer) added and incubated O/N at 4 °C. After 3X wash in PBS-T, fluorescent dye-conjugated secondary antibodies (1:500, Alexa Fluor antibodies from ThermoFisher) were added and incubated for 3 h at RT. Secondary was washed with PBS-T three times and tissues incubated in DAPI (1:1000, ThermoFisher) for 5 min and coverslips mounted with Prolong Gold Antifade Mountant (ThermoFisher P36930). Cells were washed and coverslips mounted with Prolong Glass Mounting Reagent (Thermo Fisher P36981) which contains DAPI.

For BrdU detection, antigen retrieval was performed using HCl. Slides were thawed and rinsed in PBS for 5 min at RT. Samples were incubated in fresh 2 N HCl for 1 hr or 30 min at RT. Samples were rinsed through multiple PBS washes to remove all traces of HCl. For combination IF with SOX2 or YBX1, antibodies were tested for affinity post-HCl in a separate experiment and combined with BrdU for experiments. Antibodies were used as follows: **Sox2**: diluted 1:200. **Ybx1**: rabbit/mouse diluted 1:100. **BrdU**: diluted 1:100. **Foxg1**: diluted 1:50. **Gbx2**: diluted 1:100. **GFAP**: diluted rabbit-1:200, mouse-1:250. **Tuj1**: diluted 1:100. Images were acquired with Zeiss LSM780 or Keyence BZ-X and image modifications performed with Fiji or Adobe Photoshop.

**CellProfiler analysis.** For BrdU quantification, CellProfiler was used to directly quantify the number of positive cells in each image. The DAPI-positive cells were enumerated using *IdentifyPrimaryObjects*, and BrdU- positive cells were identified in relation to those cells by using *IdentifyPrimaryObjects* and *RelateObjects*. For IHC sections, the BrdU cell number was divided by the SOX2-positive cell number to account for differences in total cell numbers and mNPC numbers in each section. For Tuj1 quantification and measurements, CellProfiler was used to quantify neuron number and characteristics of neurons. First, color images were transformed to grayscale and neurites enhanced by through two runs of *EnhanceorSuppressFeatures*: (1) Enhance, Speckles, Feature Size 12, Fast/Hexagonal and (2) Enhance, Neurites, Tubness, Smoothing Scale 1.8. Next, *Morph* erode, Once, Disk 3.0 was used to erode excess pixels. *IdentifyPrimaryObjects* with automatic thresholding was used followed by *IdentifySecondaryObjects* with Adaptive Otsu thresholding (two classes, weighted variance, no smoothing, regularization factor of 0.05). Finally, *ApplyThreshold* (Global MCT threshold, Automatic smoothing, 0.3 Threshold correction factor), *Morph* (skel, once), and *MeasureNeurons* (Fill small holes, max hole size 50) were used on skeletonized images to obtain neuron measurements and characteristics.

**Flow cytometry.** For BrdU, pS10-H3, and Sox2 FACS analysis, mNPCs were washed with cold PBS and fixed in 70% ethanol at 4 °C overnight. Cells were pelleted and treated with 2 N HCl/Triton X-100 at room temperature for 30 min to denature DNA. After neutralization, cells were incubated with primary antibody for 30 min in FACS buffer and washed in PBS. Antibodies anti-BrdU APC (diluted 1:500), anti-pS10-H3-Alexa488 (diluted 1:1000), and anti-Sox2 PE (diluted 1:500) were used. After three washes in cold PBS, cells were resuspended in PBS and analyzed by BD FACSAria™ Fusion and FlowJo software.

**Neurosphere assay.** After filtration of cells derived from embryonic neural tubes, cells were incubated in 0.5 μM Neurofluor CDr3™ + DNase I (NEB) for 1 h in a 37 °C incubator. CDr3-positive mNPCs were sorted on a BD FACSAria™ Fusion flow cytometer and plated in ultra-low attachment six-well plates at a density of 10 cells/μL. Neurospheres were allowed to grow for 6 days, with media addition on day 3 and, and samples were imaged on day 6 with a Zeiss AxioObserver D1 at 5× magnification. Neurospheres were collected and treated with Accutase for 5 min at 37 °C. Cells were counted using trypan blue to assess viability and 10 cells/μl replated in triplicated to derive secondary neurospheres. This process was repeated again after 6 days to derive tertiary neurospheres. Eight images/well were taken for a total of 24 views and analyzed. Image stacks were then created for each cell line or drug treatment set of images. Stacks were created using the FIJI "Images to Stack" menu item (found under the menu "Images → Stacks → Images to Stack"). A binary images stack was created from the original image stack by using the FIJI plugin "Trainable Weka Segmentation," (found under the FIJI menu "Plugins → Segmentation → Trainable Weka Segmentation"). Trainable Weka segmentation uses a fast-random forest classification method to create a binary image with neurospheres colored black in the foreground with a blank white background. Binary images stacks were cleaned up using the pencil tool in FIJI, as Trainable Weka Segmentation did not perfectly segment all neurospheres and some needed to be segmented by hand. Clean binary image stacks of neurospheres were then quantified using the "Analyze Particles…" menu item (found under the menu "Analyze → Analyze Particles…"). The number of neurospheres and area of each neurosphere were recorded from data generated by the "Analyze Particles" menu item for further data analysis. For the clonal analysis, primary neurospheres were dissociated with Accutase for 5 min at 37 °C, triturated to single cells, and

resuspended in fresh media. After assessing viability and live cell counts using trypan blue, cells were diluted to a concentration of 5 cells/100 μl media or 3 cells/100 μl media for plating in 96 well ultra-low attachment plates. Plates were transferred to incubator and left untouched for 1 week until counting the number of formed neurospheres. For each datapoint, neurospheres were quantified from 20 wells of 5 cells/well or from 30 wells of 3 cells/well.

**RNA isolation.** Total RNA was isolated from cultured cells and mouse neural tubes (after SOX2-positive FACS sorting from E13.5 embryos) using Ribozol (VWR) and purified using the Direct-zol RNA Purification Kit (Zymo Research). DNA digestion with DNase I was performed as suggested and RNA concentration and purity was measured by NanoDrop (ThermoFisher).

**RNA-Seq.** Paired-end 100-cycle sequencing was performed on HiSeq 2000 or HiSeq 4000 sequencers, per the manufacturer's directions (Illumina). RNA-seq was mapped and HTSeq (version 0.6.1p1)[58] was used to estimate fragments per kilobase of transcript per million mapped reads (FPKM) based based on GENCODE (v24)[59]. After normalization by trimmed mean of $M$ values (TMM) and filtering out genes not expressed in both groups (FPKM < 1), Voom was used to identify differentially expressed genes. Volcano plots were generated with ProteinPaint viewer (St. Jude) at cloud.stjude.org hosted by DNANexus. For GSEA analysis[60], gene sets were put together with MSigDB database[61](C2, v5.1) and analyzed by using prerank mode (version 3.0).

**Reverse transcription qPCR.** Total RNA (250–1000 ng) was reverse transcribed using the SuperScript IV VILO Master Mix (ThermoFisher). Real-time qPCR reactions were performed with iTaq SYBR® Green Supermix (Bio-Rad 1725124) or PowerUp SYBR® Green Master Mix (ThermoFisher A25778), using Applied Biosystems QuantStudio 3 with primers listed in Supplementary Table S2. For Taqman assays, qPCR reactions were performed with Taqman® Fast Advanced Master Mix (ThermoFisher 4444964) on the same machine with Taqman assays listed in Supp Table 1. qPCR results were quantified using the difference in threshold cycle values between the gene of interest and the endogenous control by the $2^{-\Delta\Delta Ct}$ method. All experimental samples were compared to a wild-type littermate or DMSO control.

**Wild-type YBX1 and YBX1 mutant construct generation.** Amino acids 183-205 in the full-length Ybx1 were deleted to obtain the delNLS mutation. Wild-type human YBX1 was amplified from complementary DNA (cDNA) reverse transcribed from total RNA template of H9 human embryonic stem cells using primers YBX1-F and YBX1-R (Supplementary Table S3) which contain *attB* sites for Gateway cloning. The PCR product was used in a BP reaction with pDONR221 and sequence verified by Sanger sequencing. An LR reaction transferred the wild-type YBX1 cDNA to pcDNA-DEST53. All YBX1 mutant constructs were generated by PCR amplification from pDONR221-YBX1 with primers listed in Supplementary Table S3. Fragments were either immediately used in a BP reaction with pDONR221 or for delF2 and delNLS (deletion of amino acids 183-205)—gel extracted, digested with appropriate restriction enzymes and ligated with Quick Ligase (NEB). After sequence verification, constructs were LR cloned into final destination vectors.

**Lentiviral construct generation and lentivirus production.** To generate lentiviral constructs, YBX1-FL and YBX1-delNLS were excised from pcDNA-DEST53 plasmids via PCR with primers (YBX1(BamHI)-F and YBX1(XhoI)-R). PCR products were double-digested with restriction enzymes in CUTSMART Buffer (NEB). MSCV-IRES-mCHERRY (a gift from Martine Roussel, SJCRH) was also double digested in the same buffer and the backbone fragment isolated by gel extraction. Cut PCR products and backbone were ligated with Quick Ligase and transformed into E.coli TOP10 cells. Plasmids were Sanger sequencing verified before production of lentivirus. To produce lentivirus, viral constructs were co-transfected with pMD Gag-pol and p-VSVg (gifts from Martine Roussel) into HEK 293T cells with Xfect Transfection reagent. Lentiviral particles were harvested into unsupplemented NeuroCult™ Proliferation Media, and titer quantified by Lenti-X qRT-PCR titration kit (ClonTech). For transduction, *Ybx1*-KO mNPCs were inoculated with lentiviral particles mixed with fresh proliferation media supplemented with 20 ng/mL rhEGF. Cells were allowed to recover for two days before FACS sorting for mCHERRY expression. YBX1 expression was confirmed by qRT-PCR and immunofluorescence after growth of sorted cells.

**Ultra-low input native chromatin immunoprecipitation.** Mouse NPCs from Sox2-GFP positive embryos were isolated and sorted. Using an equal number of cells as input and spike-in *Drosophila* S2 chromatin, nuclei were isolated using Nuclear Isolation Buffer (MilliporeSigma NUC101) and diluted MNase enzyme added to digest DNA. The MNase reaction was carried out for 7.5 min at 37 °C and stopped by addition of EDTA. A 1% Triton/1% deoxycholate solution was added to lyse nuclei and complete immunoprecipitation buffer added to allow for ~200 μl lysate/immunoprecipitation. Antibodies were bound to Protein A/G magnetic beads (ThermoFisher) and after pre-clearing on washed magnetic beads with no

antibody, antibody-bead complexes were added to chromatin O/N at 4 °C. After successive washes in low salt and high salt wash buffers, chromatin was eluted in ChIP elution buffer for 1.5 h at 65 °C. DNA was extracted using phenol-chloroform-isoamyl alcohol extraction and MaXtract phase-lock tubes (QIAGEN) to retain maximal DNA. DNA was further purified using Ampure XP beads (Agencourt) and libraries constructed through the following processes: (1) End repair with T4 PNK, Klenow DNA polymerase, and T4 DNA polymerase; (2) A-tailing with dATP and Klenow (3′-5′ exo-) polymerase; (3) Adapter ligation with annealed Illumina adapters and Quick DNA ligase; and (4) Library amplification with Illumina paired-end indexed primers and 2X Phusion HF Master Mix. Library amplification was performed for 12 cycles and DNA purified by Ampure XP beads and library yield and quality evaluated using a D1000 High Sensitivity screentape on an Agilent TapeStation.

**CUT&RUN**. We followed CUT&RUN described in Skene and Henikoff[42], with minor variations. Briefly, mNPCs from Sox2-eGFP-positive embryos were isolated and sorted as described above. NPCs and spike-in S2 cells were pelleted and resuspended in wash buffer. Bio-Mag Plus Concanavalin-A coated beads (Bangs Laboratories BP531) were added to cells (diluted in binding buffer) to bind nuclei to beads. Supernatant was removed and samples were blocked for 5 min at RT with digitonin block buffer. Antibody diluted in digitonin block buffer was added and samples were rotated for 3 h, 5 h, or O/N in the cold. Beads were collected and washed 3× with digitonin block buffer before adding pA-MNase to beads. After a 1-h incubation, beads were washed 3× and resuspended in wash buffer. After equilibration on ice, 100 mM CaCl₂ was added to tubes and samples were incubated for 25 min with agitation at the 15-min mark. The reaction was stopped by adding the stop buffer. Samples were incubated at 37 °C for 30 min to release chromatin. DNA was isolated by phenol-chloroform-isoamyl alcohol extraction and MaXtract phase-lock tubes (QIAGEN) to maximally retain chromatin. Chromatin was resuspended in low EDTA TE buffer and analyzed by TapeStation using the HS DNA Kit (Agilent). Libraries were made using the Accel-NGS® 1S Plus DNA Library Kit (Swift Biosciences) and submitted for sequencing. In all cases, ~15% input was removed and IgG was used as a negative control.

**Deep sequencing analysis of chromatin immunoprecipitation or CUT&RUN**. 50 bp single-end reads for ChIP-seq or 50 bp paired-end reads for CUT&RUN-seq were obtained and aligned to mouse genome assembly mm10 and fruit fly genome assembly dm6 by BWA (version 0.7.12, default parameter). Duplicated reads were then marked by Picard (version 1.65 [1160]). For ChIP-seq, uniquely mapped reads were retained by SAMtools (parameter "-q 1 -F 1024", version 1.4), and quality control was ensured by following ENCODE criteria[62]. We extended reads to fragment size estimated by SPP[63] and then normalized to dm6 sequencing reads number(fruit fly S2 cells). Peaks were inspected on IGV[64] and peaks for each replicate were called by MACS2 (version 2.0.9 20111102 option "nomodel" with "extsize" defined as fragment size estimated by SPP)[65] and SICER (redundancy threshold 1, window size 200 bp, effective genome fraction 0.86, gap size 600 bp, FDR 0.00001 with estimated fragment size defined by SPP)[66] and then merged by bedtools (version 2.17.0) after removing a SICER-called peak if it overlaps MACS2 peaks. For Cut&Run, properly paired uniquely mapped reads were retained by SAMtools (parameter "-q 1 -F 1804", version 1.4), sorted by name and converted to bedpe format by bedtools. Only fragments shorter than 2000 bp were kept to peak calling and generating of bigwig tracks. Similarly, MACS2 (bedpe mode) and SICER were used for peak calling. To find reproducible peaks, we first called peaks for each replicate twice with an FDR corrected p-value cutoff of 0.05 as a high-confidence peak and an FDR-corrected P value cutoff 0.5 as low-confidence peak set. We only considered high-confidence peaks that also overlap at least low-confidence peaks in other replicates as reproducible peaks. We did the same by employing SEACR[67], using the top 1 percentile as the high-confidence peak and the top 5 percentile as the low-confidence peak ("non stringent" mode) and further require the reproducible peaks overlap SEACR reproducible peaks. deepTools[68] was used to plot the heatmap. Peak overlap was tested by the hypergeometric test with assumed binomial distributions. Gene ontology analysis was generated by Enrichr[44]. For histone modifications, the quality control standard was set as before[69]. Voom[70] was used to test significant differences between control and Ybx1-KO after TMM normalization. For enhancer analysis, we used active and poised enhancer signature lists from the embryonic brain at E14.5 that were downloaded from the ENCODE portal (https://www.encodeproject.org/) with the identifiers ENCFF682UGG, ENCFF386QNM, and ENCFF998FUV.

**Drosophila S2 transfection and protein pull down**. Drosophila S2 cells were grown and passaged in Schneider's Drosophila medium (ThermoFisher Scientific) plus 10% heat-inactivated fetal bovine serum (FBS). For transfections, 1 × 10⁶ cells/well were seeded in 1 mL complete medium in six-well plates. Before transfection, medium was removed and replaced with FBS-free Schneider's Drosophila medium. pAMW plasmids were transfected with Cellfectin™ II Reagent (ThermoFisher) and Opti-MEM medium (ThermoFisher) according to manufacturer's instructions with 6 μg plasmid/well. After 72 h, cells were collected by scraping and lysed in Buffer D + PI + DTT. Full-length open reading frame of JARID2 was cloned into pFastBac, using the Bac-to-Bac N-His TOPO cloning kit (Life Technologies) for baculovirus

generation via Bac-to-Bac baculovirus expression from the St. Jude Protein Production Facility. Full-length recombinant protein was bound overnight at 4 °C to Pierce™ Protein A/G Magnetic Beads (ThermoFisher Scientific) with JARID2 antibody (Novus Biologicals). Protein-bead complexes (or beads with no antibody) were washed with PBST, added to recombinant protein fragment lysate for 4 h at 4 °C in HEPM, washed in PBST, and eluted with 0.1 M glycine (pH 2.3). Eluates were neutralized with 1.5 M Tris buffer, pH 8.8, and analyzed by Western blotting.

**Ezh2 inhibitor administration**. The Ezh2 inhibitor GSK126 (Cayman Chemical) was dissolved in DMSO to a stock concentration of ~9.5 mM. It was further diluted in unsupplemented Neurocult™ Basal Medium before final dilution (to 100 or 500 nM) in either Proliferation Media or Differentiation Media, as indicated in figure legends. For the H3K27me3 western blot, mNPCs were treated for 48 h with drugs at indicated concentrations before harvesting. For NPC qPCR analysis, cells were treated for 6 days with drug, and for differentiation experiments cells were pretreated for 6 days and treated throughout the differentiation course. Media with freshly diluted and the drug was added every 3 days.

**Statistics and reproducibility**. Statistical parameters including the definitions and exact value of n (e.g., total number of experiments, replications, axons, organelles, or neurons), deviations, P values, and the types of the statistical tests are reported in the figures and corresponding figure legends. Statistical analysis was carried out using Prism 8 (GraphPad Software)[27]. Statistical analysis was conducted on data from three or more biologically independent experimental replicates. Comparisons between groups were planned before statistical testing and target effect sizes were not predetermined. Error bars displayed on graphs represent the mean ± SEM of at least three independent experiments. All western immunoblots were repeated at least two times with different biological samples. Micrographs are representative of a minimum of four images taken from at least two biological replicates.

**Reporting summary**. Further information on research design is available in the Nature Research Life Sciences Reporting Summary linked to this article.

## Data availability

All sequencing data are deposited in NCBI GEO database under accession number GSE137853. Mass spectrometry data were deposited in ProteomXchange, with project accession: PXD015670. All other relevant data supporting the key findings of this study are available within the article and its Supplementary Information files or from the corresponding author upon reasonable request. The source data underlying Figs. 1c, d, f–h, k, 2g, 3c–h, 6a–f, h, and 7b–d, g–m, 8a, b, d–i and Supplementary Figs. S1a–c, S2a, d, e, g, I, j, l, S3c, d, S5i, l, S6e, f, S7a, c, e–g, S8a–h, and S9b–e are provided as a Source Data file. A reporting summary for this article is available as a Supplementary Information file. Source data are provided with this paper.

## Code availability

Code is deposited at https://doi.org/10.6084/m9.figshare.7411835. Source data are provided with this paper.

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

## Acknowledgements
The authors thank A. Andersen and M.F. Roussel for discussions, T.J. Ley for the *Ybx1*-KO mouse strain; A. Andersen and V. Shanker for editing the manuscript; J. Houston for FACS; S. Olsen and D. Roeber for sequencing samples; L. Ding, M. J. Robert, and M. Rusch for sequencing mapping; M. Sahnine, J. Klein, I. Lam, H. Chen, A. Dash, and K. Kleinrichert for experimental assistance. Images were acquired at the Cell & Tissue Imaging Center, which is supported by SJCRH and NCI P30 (CA021765). M.E. is funded by NIH (1F32HD093276). B.X. and Y.F. are supported by NCI P30 (CA21765). This research is funded by American Lebanese Syrian Associated Charities, American Cancer Society (132096-RSG-18-032-01-DDC), and NIH (1R01GM134358-01). The content is solely the responsibility of the authors and does not necessarily represent the official views of the National Institutes of Health.

## Author contributions
M.E.: most experiments and analyzed data. Y.M.: genetic crosses, NPC FACS, MEF studies, CUT&RUN and library prep, and histone WB. B.X.: sequencing analyses. C.W.: co-IP followed by mass spectrometry or WB. J.L.: genetic crosses and neurosphere assay. L.M.: cryosection and image quantification. Y.F.: supervision of B.X. V.P.: mass spectrometry analysis. J.C.P.: designed the project, analyzed data, and wrote the manuscript with inputs from all authors.

## Competing interests
The authors declare no competing interests.
