## [Peer Review File · Nature Communications]

Reviewers' comments:

Reviewer #1 (Remarks to the Author):

In this paper the authors show that Ybx1 interacts with Jarid2-containing PRC2 in mouse and human neuronal precursor cells. In neuronal precursor cells, Ybx1 is shown to control self-renewal and neuronal differentiation. Mechanistically, Ybx1 binds to PRC2 target genes and reduces the amount of PRC2-mediated H3K27me3 and promotes expression of genes related to forebrain specification. In Ybx1 knock-out NPCs, phenotypes are partially rescued by genetic or enzymatic inhibition of PRC2. The authors present a large body of interesting work, which implicates Ybx1 in regulating neuronal differentiation and the authors connect this function to regulation of PRC2 activity. This connection between Ybx1 and PRC2 and regulation of neuronal differentiation, however, is not very strong and needs to be strengthened by additional experiments prior to publication. It also remains unclear how, mechanistically, Ybx1 negatively affects PRC2 activity.

Major issues

- Figure 1 is an important Figure, which links Ybx1 to PRC2 using interaction proteomics. The experimental approaches taken, however, are very flawed. The authors make use of a Jarid2 antibody to immunoprecipitate PRC2 complexes and in these IPs, peptides for Ybx1 are detected by mass spec. The authors, however, did not perform quantitative mass spec, but just report peptide counts in a table. The authors should perform triplicate control/specific pull-downs and visualize their data, for example using volcano plots. Furthermore, the authors should confirm their findings using an independent antibody to immunoprecipitate Jarid2/PRC2 and confirm co-immunoprecipitation of Ybx1 by mass spec and western blotting. Tagging strategies can also be pursued as independent validations. Ybx1-PRC2 interactions have not been previously detected in literature in ESCs or NPCs, so this experiment should be substantially and rigorously validated prior to publication. Furthermore, from the experimental procedures section, the stringency conditions for the IPs is not clear. Were these pull-downs performed with high stringency wash buffers and with lysates treated with benzonase or some such, to exclude indirect, DNA mediated interactions?

- In Figure 4 the authors show ChIP-seq data for H3K27me3, H3K4me3 and Ybx1 in WT versus Ybx1 KO cells. The H3K27me3 tracks do not look good, very weak in fact. Furthermore, to draw firm conclusions regarding an H3K27me3 increase in Ybx1 KO cells, the authors should make use of a spike-in method in which a certain amount of chromatin, for example drosophila chromatin, is added to WT and YBX1 KO chromatin which can be used for signal normalisation afterwards. This way one can draw firm conclusions about an increase or decrease of H3K27me3 signals in WT versus Ybx1 KO cells.

- The authors should perform Ybx1 expression rescue experiments, for example for the experiments shown in Figure 5.

Reviewer #2 (Remarks to the Author):

The authors describe a mechanistic role for the Y-box binding protein Ybx1 in negative regulation of PRC2 to control the transition between self-renewal and neuronal differentiation. While PRC2 controls H3K27me to suppress transcription by RNAPII, how it controls specific gene targets underlying brain development and regional neuronal specification is still not well understood. Here the authors used Jarid2 (a PRC2 subunit) as bait to pulldown Ybx1 in human ESCs as a novel PRC2-binding protein. The authors further show that Ybx1 is required to repress PRC2 to regulate self-renewal and neuronal differentiation. In general, the experiments are well performed with appropriate controls. However, several areas can be strengthened to widen the impact.

1. Fig. 2: Ybx1 is required for self-renewal: The authors use BrdU labeling, PH3 staining, and neurosphere assays to support the conclusion that Ybx1 is required for self-renewal of neural progenitors. However, this data is not compelling. Clonal analysis should be used to directly demonstrate deficits in self-renewal due to lack of Ybx1.
2. Fig. 3: Ybx1 is required for differential gene expression in specific brain regions: The authors claim Ybx1 controls gene expression differently in different neural regions (e.g., brain vs. spinal cord or forebrain vs. midbrain/hindbrain). However, the authors mainly use neural progenitor cells derived from mouse neural tubes. To bolster significance, the authors should analyze proliferation, self-renewal and differentiation of progenitors derived from different neural regions to identify new roles of Ybx1-PRC2 during neural development.
3. Fig. 4: Ybx1 suppresses H3K27me3 levels through binding of PRC2: To reinforce the causal role of Ybx1 in negative regulation of PRC2 targets, the authors should consider using CRISPR (or some other approach) to target Ybx1 to genomic loci that lacks PRC2 and test whether this is sufficient to recruit PRC2 and repress H3K27me3.

Reviewer #3 (Remarks to the Author):

Comments to the authors

The manuscript "Ybx1 negatively regulates PRC2 to affect gene expression programs for neural lineage specification and neural progenitor self-renewal and differentiation" reports a role for the cold-shock protein Ybx1 in neural lineage specification by maintaining the activation of neural-lineage genes via suppression of PRC2 activity. The findings by the authors are interesting, the methods- including gene targeting and small-scale ChIP-seq/CUT&RUN analyses- are also elegant. This paper extends the present understanding on neural development by uncovering a complex regulation mediated by activators (e.g. Ybx1) and repressors (e.g. PRC2), and therefore, should be worth publishing.

I have the following comments for the authors.

1. This paper was not easy to read. The authors should re-structure the manuscript to make it more accessible to the readers. Below, I mention several areas that could be improved.

1-a) Make the title simpler, such as "Ybx1 regulates neural lineage specification and neural progenitor self-renewal and differentiation by suppressing PRC2 activity".

1-b) Rewrite the running title as "Ybx1 inhibits PRC2 during neurogenesis".

1-c) The storyline is based on PRC2, while the findings are based on Ybx1 knockout. I suggest that the authors re-structure the manuscript by focusing on Ybx1, which will be consistent with the experimental evidence.

1-d) To assist the reviewer, please add page numbers and line numbers.

1-e) Very little has been mentioned on the biological functions of Ybx1 in the introduction. Ybx1 has widespread roles, that are not restricted to transcription factor activity, but also involved in mRNA binding, oncogenesis and so on. The authors should give a brief outline to summarize these roles.

1-f) To streamline the story, it might be better to start at the sub-chapter "Ybx1 modulates self-renewal of NPCs", rather than the mass-spec data of JARDI2 (PRC2). Also, the role of the Ybx1

NLS for regulation of NPC should be written as a separate sub-chapter (starting at "To verify the nuclear role of Ybx1 in self-renewal.....", and ending at ".....and nuclear Ybx1 is essential for modulating the self-renewal of NPCs"). This should be followed by the sub-chapters "Ybx1 is required for neurodevelopmental gene expression", and "Ybx1 is required for the differentiation of NPCs to neurons", respectively. Finally, the authors should move into PRC2, via the sub-chapters "Ybx1 suppresses H3K27me3 by PRC2 at select Ybx1-bound genes in NPCs", "PRC2 complex binds YBX1 in human stem cells and mouse neural tubes (could be re-written as "YBX1 interacts with PRC2 components")", and "Ybx1 exerts its regulation of NPCs in part through PRC2 inhibition".

2. I also have some comments on the text.

2-a) In the abstract, move the section "Ybx1 interacts with PRC2 in human and mouse neural progenitor cells (NPCs)", after "Mechanistically, Ybx1 binds PRC2 gene targets, reduces the levels of PRC2-mediated H3K27me3, and promotes the expression of genes in forebrain specification, cell proliferation, or neuronal differentiation".

2-b) In the abstract, I have some reservation about the statement that "Ybx1 binds PRC2 gene targets", because Ybx1- to be more precise- binds to "Ybx1 targets", and likely "suppresses ectopic increase of PRC2-mediated H3K27me3 at these sites".

2-c) The section "Quantification revealed significantly more BrdU-positive Ybx1-KO NPCs than those in control embryos", could be simply re-written as "These quantifications showed that there was a significant increase of NPCs in Ybx1-KO embryos". Also, as the authors have already defined NPCs as "BrdU-positive", it might not be necessary to always mention "NPCs" as "BrdU-positive NPCs".

2-d) As it does not directly deal with the neural phenotype, is the following section necessary "For comparison, we quantified the proliferation of mouse embryonic fibroblasts (MEFs) and showed that Ybx1-KO MEFs had decreased proliferation compared to controls (Supplemental Fig. 2e-g). Our findings are consistent with the reduction in proliferation of Ybx1-KO MEFs reported previously 25. These data suggest that Ybx1 has different cellular influences in different developmental contexts: promoting the proliferation of fibroblasts and suppressing the proliferation of NPCs."?

2-e) What is cell-autonomous self-renewal (as in "We next used the neurosphere assay to examine the effect of Ybx1 over cell autonomous self-renewal.")? Would simple "self-renewal" suffice?

2-f) Please bring supplemental figure 2i in the main figure, as it is important in showing that the NLS plays a role in Ybx1-mediated neurogenesis. Also, in materials and methods, please clearly mention which amino acids were deleted to obtain the NLS-mutant Ybx1.

2-g) In the sub-chapter "Ybx1 is required for neurodevelopmental gene expression", I would suggest that the reviewers focus much more on the genes that are "downregulated", as these should be the target of the increased PRC2 activity. The description on "upregulated genes" should be minimum, and moved below the section on the "downregulated genes".

2-h) The authors mention "Our RNA-seq analysis showed that several forebrain lineage genes were downregulated in Ybx1-KO NPCs." What are these genes, and how do they function in neurogenesis? This is an important part of the story and the authors should take care to explain the fact that Ybx1 is maintaining the activated state of these lineage specific genes, and that in the absence of Ybx1 the same group of genes tend to be silenced by ectopic H3K27me3/PRC2.

2-i) The description on H3K4me3, as it is not related to the main story (Ybx1 versus PRC2), could be much reduced. Please downsize (or remove altogether) the section "H3K4me3 ChIP-seq signals genome-wide were higher in Ybx1-KO than in control NPCs (Fig. 4f). Unlike H3K27me3 level increases genome-wide, H3K4me3 level increases occurred in specific regions. We separated Ybx1-

bound regions into 5 categories by intensity and location of H3K27me3 and H3K4me3 level increases, as shown by different average profiles at the top of the heat map in Fig. 4a. For example, H3K4me3 levels increased only at regions in category 4. Further, the numbers of H3K4me3-occupied regions at promoters 3' of transcription start site, exons, introns, termination sites, distal regions (within 50 kb of a coding gene), and intergenic regions all decreased in Ybx1-KO NPCs (Supplemental Fig. 4j, k). Although genes (regardless of Ybx1 binding status) with higher H3K4me3 levels were enriched in upregulated genes in Ybx1-KO NPCs (Supplemental Fig. 4l), we did not observe Ybx1 protein binding with H3K4 methyltransferases, which are represented by the core subunit Rbbp5 (Fig. 1d). These data suggest that Ybx1 does not directly influence H3K4me3. WB analysis of neural tube cells showed genome-wide increases of H3K27me3 and H3K4me3 in Ybx1-KO vs. WT (Supplemental Fig. 4m). Altogether, our data suggest that Ybx1 binding leads to reduction of H3K27me3 and promotion of gene activation in NPCs".

3. The model proposing an antagonistic relationship between Ybx1 and PRC2 should be clarified more. To this end, the authors should make extensive use of the NGS data that they have generated. In particular, they need to address the following questions.

3-a) Figure 4a shows that TSS binding profile of Ybx1, and supplementary figure 4b shows the distribution profile of the Ybx1 peaks over various genomic compartments such as TSS or TES. Supplementary figure 4b should be brought in to the main text, as it is an important piece of information. Also, this reviewer wonders if the TSS binding sites of Ybx1 overlap with CGIs (CpG islands)? It is well known that CGIs are major targets of polycomb complexes. It would make sense if Ybx1 binds and protects some CGIs (short in length?) from being targeted by PRC2. In the absence of Ybx1, the same CGIs could be bound by PRC2, via the intrinsic affinity of polycomb complexes for CGIs.

3-b) The authors should compare the group of Ybx1-bound TSS that are (i) upregulated in Ybx1-KO, (ii) downregulated in Ybx1-KO, and (iii) the rest. What are the levels of H3K27me3 and H3K4me3 in groups (i), (ii) and (iii); before or after of Ybx1 deletion? What are the gene expression levels of (i), (ii) and (iii); in WT and Ybx1-KO?

3-c) The phenotype of Ybx1-KO is partially rescued in a double KO of Ybx1 and Eed, or upon administration of a chemical compound that inhibits PRC2 activity. Although this shows that Ybx1 inhibits ectopic activity of PRC2, I wonder if the opposite is also true? What is the level of Ybx1 binding in Eed-KO cells or tissues? Does it increase?

3-d) It is not clear how, mechanistically, Ybx1 inhibits PRC2 activity. The authors showed that the NLS is required for normal biological roles of Ybx1 during neurogenesis, which is expected as without the NLS Ybx1 show reduced localization to the nuclei. Have the authors checked which Ybx1 domains are involved in the interaction with PRC2, for example by co-IP? This could be done in any relevant cell lines such as hNPC, or even in easier systems such as HEK293 or HeLa.

Reviewer #4 (Remarks to the Author):

Ybx1 negatively regulates PRC2 to affect gene expression programs for neural lineage specification and neural progenitor self-renewal and differentiation

This work seems well written besides few typos. In the paper the authors demonstrate quite clearly the ability of Ybx1 to counteract/offset PRC2 activity in order to specifically favour neural lineage specification, by directly orchestrating the regulation of a set of crucial genes. The title however is quite elaborated, and I think a bit cumbersome. I would suggest developing a less descriptive and pointier one. In the abstract and along the text the authors stress the importance of distinguishing cell-intrinsic and non-cell autonomous mechanisms, they describe how Ybx1 have

different cellular influences in different developmental contexts, and they identify forebrain specific dysregulations: I would put more emphasis to at least one of these aspects.

PRC2 complex binds YBX1 in human stem cells and mouse neural tube

I find the strategy to track PRC2 bindings very compelling, but it does not seem justified enough, and I could not find in the methods and supplementary material which antibody was used. Moreover, this piece of the paper supports the idea that Ybx1 binds PRC2 but I don't find in the introduction a part that problematize the various forms of PRC2 that have been proposed, and I expected to find here or in the discussion a brief argumentation on the possibility that in the forebrain Ybx1 is part of PRC2 and not just its "negative counteractor".

Ybx1 modulates self-renewal of NPCs

On a general note, I find that the problem of reproducibility is well confronted here. With a further quantification step (exencephaly in Ybx1 mice) with respect to previous literature that fortify the observations. I appreciated the coupling of BrdU assays in vivo and in spheroids. I wonder whether a marker different from Sox2 could be used to identify NPCs since NESTIN, for instance, has been used elsewhere in the same work and Sox2 is expressed at even earlier stages of development. Besides graphically improvable Fig.2E-G is very clear and upholds an extensive amount of work. The authors findings on Ybx1 KO effects in different developmental contexts is clearly an important observation that might merit future investigations.

Ybx1 is required for neurodevelopmental gene expression

Here I found the Methods for RNA-seq analysis quite poor. The authors apply Voom, using TMM normalization which is supposed to use a stringent algorithm and to provide stringent results, but the number of genes found differentially expressed is notable, and I found no indication of how the genes were filtered. How did the author define a gene as expressed? I would rather see in the methods a precise definition of how the genes were filtered, and in the supplementary a proof that the number of genes differentially expressed does not change dramatically after a certain level of filtering. In the supplementary materials, or at least via correspondence through the editor, a brief benchmark of the effect of selecting Salmon or Kallisto instead of HT-seq. Moreover, while the functional characterization is very useful to defend the main points of the paper, I find that a better characterization might produce a better dissection and serve some clues on the gene regulatory networks underlined by the interference with Ybx1. On a personal note, while showing relevant observations, I find GSEA a very basic level of RNA-seq analysis. Could we conclude from the authors data that "Ybx1 buffers self-renewal"? The combined down-regulation of forebrain lineage genes and up-regulation of mid- and hindbrain lineages genes in KOs seems very relevant. It should be further investigated: for instance, are there motifs of specific transcription factors that are differentially enriched in the promoters or enhancers of those genes? Again, I think it would be easy to do some master regulator analysis or clustering here. The results of these analyses would highly help discuss the differences and insights that the authors obtained from the experiments performed around the questions that they try to answer in this part of the paper.

Ybx1 suppresses H3K27me3 by PRC2 at select Ybx1-bound genes in NPCs

I found crucial the use of CUT&RUN to identify Ybx1 binding sites, but it is not obvious to me why ChIPseq was preferred for H3K27me3. I suppose it is just a matter of when the two experiments were performed but I find it a bit inconsistent, since CUT&RUN is significantly less expensive and one could have performed it anyway, to then use H3K27me3 peaks found in ChIP-seq and CUT&RUN to benchmark their ability to produce comparable tracks. Moreover, since IgG were obtained from the lines, I highly recommend performing peak calling on CUT&RUN data using SEACR. Given the difference between MACS2 and SICER I would also recommend showing somewhere in the supplementary what is the merit of performing peak-calling with those two methods and merge the outputs. What parameters were used by the authors with SICER and MACS2? I think it must be stated more clearly in the methods. Finally, for what concerns the differential deposition of histone marks, I did not understand why TMM-normalization was used, what tools were used to quantify the ChIP-seq reads occupation over the genome, and on what

subset of regions the quantification was performed. I suppose the author used deepTools but I would love to have a better description in the methods. A correct normalization is necessary to call differential mark deposition. Moreover, I would have expected the authors to take advantage of available enhancer references for NPCs and neuronal lineages, to associate differentially marked regions with differentially expressed genes. This process, which I don't find described in the paper, could have led to more mechanistic and specific dissections of the observed dysregulations. It is clear from the authors analysis that Ybx1 binds preferentially promoters at very close distance from the TSS, but what about the rest of the genome? Considering the well-known relevance of PRC2 in the regulation of bivalent promoters, and their regulation relevance in developmental processes, what is the proportion of bivalent promoters bound by Ybx1? Having ChIP-ped both H3K4me3 and H3K27me3 seems like an immediate link to bivalency. The authors write: "There was a strong correlation between downregulated gene expression and increase in level of H3K27me3 (nearly passing a more stringent FDR criterion; Fig. 4d,e)", I have no particular issues with presenting data not passing a stringent FDR, but I would have expected some ChIP-qPCR to validate some of the genes/region reported in Fig 4d, since from the picture the anyway expected phenomenon of general down-regulation of "hyper-H3K27threemethylated" genes - and vice-versa - does not appear as evident for the highlighted genes.

Ybx1 is required for the differentiation of NPCs to neurons

The authors use a very fast differentiation assays which do not produce very pure homogeneous cultures, in a context in which differentiation is put to test. I could not identify how many differentiation replicates were performed to compensate for such variability. The number of neurons screened seems anyway compatible with the aim of the authors. I am not convinced that one-sided Student t test is the best tool to verify that the evaluate the differences in fig. 5e-g but I reckon it is a common practice. A non-parametric test could be appreciated in this part. However, the effect seems striking and I am convinced that increasing the number of observations or changing the statistical tests would not change the outcome of this inquiry.

Ybx1 exerts its regulation of NPCs in part through PRC2 inhibition.

I find this part of the paper relevant and potentially crucial for the definition of Ybx1 contribution to the modulation of gene expression exerted by PRC2. The rescue of forebrain lineage genes and decreased expression of "most" midbrain and hindbrain lineage genes sounds a bit as an overstatement. I would have expected the authors to state more plainly that the expression levels of selected (standard, trustworthy, et simila) markers of the three lineages had been rescued via the chemical treatment of Ybx1 KO lines with GSK126. It has been shown in literature that EZH2 inhibition alone reduce self-renewal and favours differentiation. I would have expected to see (together with DMSO-Ybx1KO and DMSO-Ybx1KO-GSK126) DMSO-GSK126 and DMSO- treated lines. While I am convinced that the observed effects are robust, I am afraid that what the authors are observing is not necessary a rescue phenotype. I would have expected the authors to check the expression levels of CDKN1A and RELN, two crucial targets/interactors of EZH2 in neurogenesis, in these lines.

The further conditional depletion of EED helps supporting the claim that the evidences collected in the previous part of this section are related to a partial rescue of the correct balance between Ybx1 and PRC2, but I wonder what mechanism is exactly bringing up the effects that the authors observe, in a context in which neither Ybx1 nor PRC2 are effectively working/present. Does Ybx1 "simply" binds the promoters to prevent PRC2 interactions with the regulatory regions bound by Ybx1? H3K27me3 is a good proxy of PRC2 activity and genomic targets, but a ChIP-seq or CUT&RUN of EZH2 would help answer this question. An alternative solution I envision is to ChIP-seq Jarid2, with the same antibody used in the first section of the paper, for the sake of consistency.

I find that the final sentence of the paper would fit as a better title ("Ybx1 inhibits PRC2 to mediate brain regionalization, modulate self-renewal and promote neuronal differentiation")

Review Conclusions

I consider the paper to be very close to being appropriate for publication in Nature Communications. However, I consider some correction necessary and some useful to support the claims of the paper.

Necessary:

- 1) To specify clearly what JARID2 antibody was utilised
- 2) To perform a deeper analysis of RNA-seq data, starting by performing GO enrichments using a background/universe (the total list of expressed genes vs the differentially expressed genes can be done trivially and with several tools), and eventually by, for instance, performing some basic clustering of differentially expressed genes by z-scores or logFC
- 3) To perform TF/motif enrichment analysis on Ybx1 ChIP-seq to understand if Ybx1 has potential partners that might have been pulled down with it in the initial Jarid2immunoprecipitation
- 4) In the CUT&RUN analysis: to apply SEACR and to verify that SICER and MACS2 are not including too many false positives per se and even more after merging their outputs.
- 5) To specify in the methods the filtering approach applied to RNA-seq prior differential expression analysis
- 6) To specify in the methods the parameters and conditions used for quantitative analysis of ChIP-seq and CUT&RUN.

Highly suggested:

- 1) To expand in the introduction the description of known PRC2 complexes
- 2) To perform CUT&RUN or ChIP-seq on EZH2 or other PRC2-specific subunits
- 3) To verify expression levels of CDKN1A and RELN along differentiation and upon PRC2 inhibition in Ybx1 KO lines
- 4) To add DMSO-wildtype and DMSO-wildtype-GSK126 lines in the PRC2 inhibition experiments
- 5) To discuss bivalent regions.
- 6) Supplemental Figure 4: Pie chart are deprecated for the depiction of genome-wide peaks distribution: I highly suggest substituting them with barplots.

We thank the reviewers for their thoughtful suggestions, which collectively have much improved all sections and the organization of this manuscript. We have obtained new data to strengthen the following conclusions. (1) Ybx1 physically and functionally interacts with PRC2. (2) Ybx1 regulates gene expression and self-renewal vs. differentiation of brain neural progenitor cells (NPCs). (3) Ybx1 suppresses H3K27me3 levels at crucial developmental genes to promote gene expression.

We have obtained new data to draw conclusions about the mechanistic activities of Ybx1 that influence PRC2. (a) The Ybx1 domain is required for its interaction with PRC2. (b) Ybx1 binding highly overlaps CpG islands, bivalent regions, and PRC2-bound genes. (c) Ybx1 has enriched binding at poised enhancers and affects active enhancers involved in neural development, dendrite morphogenesis, axonogenesis, axon guidance, and synapse formation or function. (d) Ybx1 is likely a PRC2 accessory subunit regulating PRC2 binding to chromatin and neurodevelopmental programs.

We propose the following model (see right on the right). In wild-type NPCs, Ybx1 promotes and restrains PRC2 binding to chromatin while inhibiting H3K27me3 levels. These activities of Ybx1 ensure chromatin dynamics are 'just right' in order to facilitate gene activation during neural development. In Ybx1-knockout (KO) NPCs, PRC2 binding to chromatin is reduced, and H3K27 methylation by the remaining PRC2 is increased. Upon Ybx1-KO NPCs receiving gene activation cues, increased or persistent H3K27me3 levels lead to gene suppression and failure in executing neurodevelopmental programs.

Reviewer #1 (Remarks to the Author):

In this paper the authors show that Ybx1 interacts with Jarid2-containing PRC2 in mouse and human neuronal precursor cells. In neuronal precursor cells, Ybx1 is shown to control self-renewal and neuronal differentiation. Mechanistically, Ybx1 binds to PRC2 target genes and reduces the amount of PRC2-mediated H3K27me3 and promotes expression of genes related to forebrain specification. In Ybx1 knock-out NPCs, phenotypes are partially rescued by genetic or enzymatic inhibition of PRC2. The authors present a large body of interesting work, which implicates Ybx1 in regulating neuronal differentiation and the authors connect this function to regulation of PRC2 activity. This connection between Ybx1 and PRC2 and regulation of neuronal differentiation, however, is not very strong and needs to be strengthened by additional experiments prior to publication. It also remains unclear how, mechanistically, Ybx1 negatively affects PRC2 activity.

We thank the Reviewer for these insights. We have obtained new data to strengthen the observations on the PRC2–Ybx1 interaction and its regulation of neuronal differentiation. A separate, independent study supports that PRC2 binds Ybx1 (Oliviero et al. 2016 *Mol. Cell Proteomics*). Additional IP-mass spectrometry datasets using 2 JARID2 antibodies in 2 different cell types as well as IP-WB and pull down strengthened the conclusion that PRC2 binds YBX1. Gene expression and chromatin profiling

results support that PRC2 and Ybx1 co-bind and regulate genes (and their enhancers) that mediate neuronal differentiation, axonogenesis, axon guidance, and synapse formation and function.

We show that Ybx1 promotes PRC2 binding to chromatin while modulating the appropriate levels of H3K27me3 genome-wide. Although we have not been able to uncover the exact molecular mechanism by which Ybx1 regulates PRC2 activity, we hope the Reviewer agrees that the in-depth structural biological examination of the PRC2–Ybx1 interaction is beyond the scope of this study.

Altogether, our data suggest that Ybx1 fine-tunes PRC2 activities and the appropriate H3K27me3–H3K4me3 balance at key neurodevelopmental genes involved in brain regionalization, NPC self-renewal, and neuronal differentiation.

Major issues

- Figure 1 is an important Figure, which links Ybx1 to PRC2 using interaction proteomics. The experimental approaches taken, however, are very flawed. The authors make use of a Jarid2 antibody to immunoprecipitate PRC2 complexes and in these IPs, peptides for Ybx1 are detected by mass spec. The authors, however, did not perform quantitative mass spec, but just report peptide counts in a table. The authors should perform triplicate control/specific pull-downs and visualize their data, for example using volcano plots.

We agree with the Reviewer that this is an important issue and that the original mass spectrometry data presentation was not quantitative. We have improved IP-mass spectrometry data by adding new data and better data presentation. A separate publication (Oliviero et al. 2016 *Mol. Cell Proteomics*) used EZH2 IP-mass spectrometry to show that YBX1 is a candidate PRC2 interactor. However, there was no validation of EZH2-YBX1 binding in the 2016 paper.

In the revised manuscript, we present the IP-mass spectrometry datasets in Supplemental Fig. 1 to explain the rationale of studying Ybx1. We have moved the IP-western blotting and protein pull-down data to Fig. 6 and Supplemental Fig. 7 to the suggestion of Reviewer 3 to reorganize the sections.

We have performed a third JARID2 immunoprecipitation (IP)–mass spectrometry by using the first JARID2 antibody (Novus Biological NB100-2214) and H9 human embryonic stem cells. This analysis recovered JARID2, SUZ12, RBBP4, and YBX1, but not EZH2, EED, and AEBP2. Data from the original 2 datasets were highly consistent, with a correlation coefficient of 0.753, R^2 of 0.7876, and p (via the F test) of $2.599e-105$. Therefore, we present a volcano plot from these 2 datasets in Supplemental Fig. 1d. \log_2 of mean abundance values and \log_{10} of the p -values were used to construct the volcano plot. Abundance values were calculated by the formula, spectral count \times 50kD / protein size kD (Zhou et al. 2010 *J Proteome Res*). The p values were calculated by using the G-test to compare abundance values in JARID2 and IgG IP replicates. The enrichment of YBX1 in JARID2 IP was higher than that of other PRC2 subunits EZH2, EED, or AEBP2 (Supplemental Fig. 1a).

To further strengthen IP-mass spectrometry data, we used a separate, validated JARID2 antibody (Supplemental Fig. 7a; R&D Systems AF6090) and mouse neural stem cells NE4C (ATCC CRL-2925) to generate a new triplicate dataset. We chose NE4C cells for the second JARID2 IP-mass spectrometry dataset to strengthen the conclusion that Jarid2 and Ybx1 interact in mouse neural stem and progenitor cells. The new IP-mass spectrometry datasets are presented in Supplementary Fig. 7b. This new JARID2 IP-mass spectrometry also showed a high enrichment in YBX1, with a correlation coefficient of 0.548, R^2 of 0.748, and p (via the F test) of $6.68e-51$.

S1d**S7a****S7b****S1e**

Proteins	peptide #, IgG			spectral count, IgG			peptide #, JARID2			spectral count, JARID2			p value		
	IP1	IP2	IP3	IP1	IP2	IP3	IP1	IP2	IP3	IP1	IP2	IP3	IP1	IP2	IP3
JARID2	0	0	0	0	0	0	14	27	15	47	63	75	1e-15	2e-20	9e-6
SUZ12	0	0	0	0	0	0	6	24	11	23	53	55	2e-8	2e-17	2e-4
EZH2	0	0	0	0	0	0	6	16	0	13	22	0	3e-5	6e-8	n.a.
EED	0	0	0	0	0	0	5	6	0	13	19	0	4e-5	5e-7	n.a.
AEBP2	0	0	0	0	0	0	9	3	0	6	15	0	6e-3	9e-6	n.a.
RBBP4	0	0	0	0	0	0	6	8	2	13	12	10	3e-5	8e-5	1e-1
YBX1	0	0	0	0	0	0	7	4	15	35	15	75	6e-6	8e-6	9e-6

Legend: (S1a) Volcano plot from duplicate JARID2 IP-mass spectrometry compares the enrichment of identified protein in JARID2 vs. IgG IP. Abundance was calculated with the formula of spectral count \times 50kD / protein size kD. *P* value was calculated by the G-test comparing abundance values in JARID2 vs. IgG IP. (S5d) Peptide and spectral counts from triplicate IgG and JARID2 IP.

Furthermore, the authors should confirm their findings using an independent antibody to immunoprecipitate Jarid2/PRC2 and confirm co-immunoprecipitation of Ybx1 by mass spec and western blotting. Tagging strategies can also be pursued as independent validations. Ybx1-PRC2 interactions have not been previously detected in literature in ESCs or NPCs, so this experiment should be substantially and rigorously validated prior to publication. Furthermore, from the experimental procedures section, the stringency conditions for the IPs is not clear. Were these pull-downs performed with high stringency wash buffers and with lysates treated with benzonase or some such, to exclude indirect, DNA mediated interactions?

Using the hESC nuclear extract, we performed JARID2 and YBX1 co-IP-western blotting with the benzonase treatment (125 units/ml for 30 minutes; Sigma Millipore E1014) (Supplemental Fig. 7f) and high-stringency wash buffer (15 mM Tris-HCl pH 7.5, 5 mM EDTA, 2.5 mM EGTA, 1% Triton X-100, 1% sodium deoxycholate, 0.1% SDS, 120 mM NaCl, 25 mM KCl) (Fig. 6a). We used 2 additional JARID2 antibodies and high-stringency washes in IP-western blotting to confirm that JARID2 co-immunoprecipitated with YBX1: Supplemental Fig. 7g shows results using a JARID2 antibody from R&D Systems AF6090, and Supplemental Fig. 7h shows results using a JARID2 antibody from EMD Millipore ABE425.

We added to text, lines 350-356: "We confirmed the JARID2–YBX1 binding with new IP-mass spectrometry datasets using a different JARID2 antibody (R&D Systems AF6090) in mouse neural stem cells NE4C (Supplemental Fig. 7a, b). We validated the specificity of the antibody targeting human

YBX1/mouse Ybx1 (Supplemental Fig. 2a, 7c, and 7d). Using the hESC nuclear extract, we then performed additional co-IP experiments using benzonase treatment (to eliminate RNAs and DNAs as factors mediating the JARID2–Ybx1 interaction), high-stringency washes, and 2 other JARID2 antibodies (Fig. 6a and Supplemental Fig. 7e-g).”

We used His-tagged JARID2 and GST-tagged YBX1 for IP–western blotting to confirm the JARID2–YBX1 interaction (Fig. 6e). In response to Reviewer 2 (please see the section on pages 14-15 of this document; Fig. 6f), we then narrowed down the protein domain within YBX1 that binds to JARID2.

Legend: Western blotting of IP using (S7f) benzonase treatment, (6a) high-stringency wash buffer, (S7g) an alternative JARID2 antibody from R&D Systems, and (S7h) an alternative JARID2 antibody from EMD Millipore.

- In Figure 4 the authors show ChIP-seq data for H3K27me3, H3K4me3 and Ybx1 in WT versus Ybx1 KO cells. The H3K27me3 tracks do not look good, very weak in fact. Furthermore, to draw firm conclusions regarding an H3K27me3 increase in Ybx1 KO cells, the authors should make use of a spike-in method in which a certain amount of chromatin, for example drosophila chromatin, is added to WT and YBX1 KO chromatin which can be used for signal normalization afterwards. This way one can draw firm conclusions about an increase or decrease of H3K27me3 signals in WT versus Ybx1 KO cells.

Data from the original submission did use a spike-in method. Please note that in the original submission, we had included “then normalized to dm6 sequencing reads number(fruit fly S2 cells)” for sequencing analysis in the methods section (now lines 1021-1022). We regret that we had not sufficiently explained that we used spike-in *Drosophila* S2 chromatin and the ChIP’d *Drosophila* read counts for normalization between WT and *Ybx1*-KO datasets.

We added in lines 194-196: “ChIP-seq and CUT&RUN-seq were performed with spike-in method: *Drosophila* S2 chromatin for histone profiling and H2Av CUT&RUN in S2 cells for Ybx1 CUT&RUN-seq.” We have added the following text in the Methods section: “and spike-in *Drosophila* S2 chromatin” in line 977 for ChIP-seq and “NPCs and spike-in S2 cells” in line 998 for CUT&RUN-seq. Also relevant is the information that in response to Reviewer 4’s comments, we have used spiked-in method to perform H3K27me3 CUT&RUN-seq (please see the relevant section on page 20) to support conclusions from the original H3K27me3 ChIP-seq.

We assume that the Reviewer might have reacted to the low peak values of the average H3K27me3 profiles in Ybx1-bound regions (Fig. 4a). The heat map in Fig. 4a only included H3K27me3 profiles specifically within Ybx1-bound peaks. To address Reviewer 4's comment about the SEACR software, we examined only H3K27me3 peaks (see figure on the right). As shown by the average profiles of 2 replicate datasets, the H3K27me3 peaks were quite robust.

- The authors should perform Ybx1 expression rescue experiments, for example for the experiments shown in Figure 5.

We used lentiviruses carrying wild-type YBX1, a mutant YBX1 excluded from the nucleus (delNLS), and empty vector as negative control to transduce *Ybx1*-KO NPCs. We then performed FACS by the mCherry marker in the lentiviruses, and examined gene expression and NPC differentiation. In the original submission, we showed that wild-type YBX1 rescued (reduced) the proliferation of *Ybx1*-KO NPCs (Fig. 1j, k).

To address this comment, we showed that wild-type YBX1 restored the expression of some forebrain lineage genes (Supplemental Fig. 2l). This suggests that YBX1 partially rescues gene expression in *Ybx1*-KO NPCs.

In the *in vitro* differentiation assay, however, wild-type YB1 did not rescue neuronal differentiation in *Ybx1*-KO NPCs (Supplemental Fig 9a). In order for *Ybx1*-KO NPCs to differentiate to neurons, the cells would need to alter their epigenetic programs, which appeared to become "fixed" (we infer that these included PRC2 activities and increased H3K27me3 levels). In contrast to our chemical and genetic means to inhibit PRC2, re-expression of YBX1 was not sufficient to influence PRC2 activities and reduce H3K27me3 levels to abrogate the "fixed" epigenetic program.

S2l

S9a

Legend: (Supp Fig 2l) RT-qPCR analysis of gene expression in *Ybx1*-KO NPCs transduced with empty control, wild-type *Ybx1*-, or delNLS-*Ybx1*. (Supp Fig 9a) IF of differentiation of *Ybx1*-KO NPCs transduced with lentiviral negative control, wild-type *Ybx1*, and delNLS-*Ybx1*. Bar, 100 µm.

Reviewer #2 (Remarks to the Author):

The authors describe a mechanistic role for the Y-box binding protein Ybx1 in negative regulation of PRC2 to control the transition between self-renewal and neuronal differentiation. While PRC2 controls H3K27me to suppress transcription by RNAPII, how it controls specific gene targets underlying brain development and regional neuronal specification is not well understood. Here the authors used Jarid2 (a PRC2 subunit) as bait to pulldown Ybx1 in human ESCs as a novel PRC-binding protein. The authors further show that Ybx1 is required to repress PRC2 to regulate self-renewal and neuronal differentiation. In general, the experiments are well performed with appropriate controls. However, several areas can be strengthened to widen the impact.

We thank the Reviewer for recognizing the strengths of our study and recommending experiments to improve the sections of this manuscript.

1. Fig. 2: Ybx1 is required for self-renewal: The authors use BrdU labeling, PH3 staining, and neurosphere assays to support the conclusion that Ybx1 is required for self-renewal of neural progenitors. However, this data is not compelling. Clonal analysis should be used to directly demonstrate deficits in self-renewal due to lack of Ybx1.

We respectfully want to clarify that the original Fig. 2 suggests that Ybx1 modulates the self-renewal of NPCs. Ybx1 is not required for the self-renewal of NPCs.

We performed a new clonal assay to analyze 2 sibling control and 2 *Ybx1*-KO NPC groups. These results revealed that significantly more neurospheres were formed by the *Ybx1*-KO groups than the control groups ($p=8.9e-5$ by Student's paired *t* test, assuming unequal variance; Supplemental Fig. 2i, j). Results from this clonal assay confirmed that *Ybx1*-KO NPCs had increased self-renewal.

We adapted the protocol of this clonal assay from Gan et al. 2011 J Neurochem and Ramasamy et al. 2014 Stem Cells. We dissociated primary neurospheres, subjected the cell suspension through 40um cell strainer, and then diluted the resultant single-cell suspension to 30 cells/ml or 50 cells/ml density. We pipetted 100 μ l into each well of a 96-well plate so (i) 3 or (j) 5 cells were in each well. For each datapoint in the graphs, neurospheres were quantified from 20 wells of 5 cells/well or from 30 wells of 3 cells/well. The Neurosphere Formation Unit (NFU) was calculated as the number of neurospheres formed per 100 cells for 5 cells/well or 90 cells for 3 cells/well.

Legend: Number of neurospheres formed by NPCs from primary neurospheres in 96-well plates. Neurospheres were formed by (S2i) 3 or (S2j) 5 NPCs per 96-well. ****, $p<0.0001$ by one-sided *t* test.

2. Fig. 3: Ybx1 is required for differential gene expression in specific brain regions: The authors claim Ybx1 controls gene expression differently in different neural regions (e.g., brain vs. spinal cord or forebrain vs. midbrain/hindbrain). However, the authors mainly use neural progenitor cells derived from mouse neural tubes. To bolster significance, the authors should analyze proliferation, self-renewal and differentiation of progenitors derived from different neural regions to identify new roles of Ybx1-PRC2 during neural development.

We greatly appreciate the point made by the Reviewer. *Ybx1* is likely required for forebrain lineage specification: the *Ybx1*-KO forebrain failed to form by embryonic day E11, and the presumptive midbrain (positive for *Gbx2*) invaded the forebrain region of *Ybx1*-KO. This developmental defect precludes the possibility of isolating NPCs specifically from the forebrain. Our experimental data also led us to specifically focus on the PRC2–*Ybx1* interaction in the brain. For all these reasons, we compared gene expression in brain NPCs vs. spinal cord NPCs for the original submission.

To address this comment, we formed neurospheres from spinal cord NPCs of sibling control and *Ybx1*-KO. Quantification showed that neurospheres from control and *Ybx1*-KO spinal cords did not significantly differ. This result suggests that although *Ybx1* modulates the self-renewal of brain NPCs, it does not seemingly affect the self-renewal of NPCs from the spinal cord.

We then attempted to perform a differentiation assay in NPCs from the spinal cord. In multiple experiments that used different conditions, spinal cord NPCs became quiescent when cultured *in vitro*. The control and *Ybx1*-KO NPC groups did not differ in their response to the differentiation media.

Our old and new data led us to conclude that *Ybx1* has a significant influence over brain NPCs and our study is better suited to focus on the PRC2–*Ybx1* interaction in the brain.

Legend: Number of neurospheres formed by NPCs from spinal cord.

3. Fig. 4: *Ybx1* suppresses H3K27me3 levels through binding of PRC2: To reinforce the causal role of *Ybx1* in negative regulation of PRC2 targets, the authors should consider using CRISPR (or some other approach) to target *Ybx1* to genomic loci that lacks PRC2 and test whether this is sufficient to recruit PRC2 and repress H3K27me3.

We agree that to test the sufficiency of *Ybx1* to recruit PRC2, we would have to use CRISPR to tether the *Ybx1*–dCas9 fusion transgene to ectopic sites and determine the status of PRC2 and H3K27me3. This experiment would be expensive and take a lot of time for assay optimization. Other groups and our group have tested multiple genes and regions and observed that the use of CRISPR for robust recruitment of dCas9 fusion genes is especially challenging for heterochromatin silencing; 1 example is CRISPR interference with use of dCas9-KRAB (Mandegar et al. 2016 *Cell Stem Cell*). A negative result could be due to the inefficient recruitment of *Ybx1*–dCas9 by CRISPR. On the other hand, we would not be able to preclude the possibility that a positive result might be a false positive caused by transgene overexpression. For these reasons, we unfortunately will not be able to address this comment with sufficient data rigor.

We have obtained new data to indicate that in NPCs, *Ybx1* is a potential accessory subunit of PRC2 regulating PRC2 binding to chromatin and distribution (pages 26-28 of this document; Fig. 5). We have narrowed down the protein domain in *Ybx1* required for binding to *Jarid2* (pages 14-15 of this document; Fig. 6). We hope the Reviewer agrees that these new results contribute significantly to the mechanistic understanding of the influence of *Ybx1* over PRC2 complex in the neurodevelopmental context.

Reviewer #3 (Remarks to the Author): Comments to the authors

The manuscript “Ybx1 negatively regulates PRC2 to affect gene expression programs for neural lineage specification and neural progenitor self-renewal and differentiation” reports a role for the cold-shock protein Ybx1 in neural lineage specification by maintaining the activation of neural-lineage genes via suppression of PRC2 activity. The findings by the authors are interesting, the methods- including gene targeting and small-scale ChIP-seq/CUT&RUN analyses- are also elegant. This paper extends the present understanding on neural development by uncovering a complex regulation mediated by activators (e.g. Ybx1) and repressors (e.g. PRC2), and therefore, should be worth publishing.

I have the following comments for the authors.

We thank the Reviewer for appreciating the significance of this study and supporting its publication.

1. This paper was not easy to read. The authors should re-structure the manuscript to make it more accessible to the readers. Below, I mention several areas that could be improved.

We also thank the Reviewer for suggesting to improve the presentation and writing of this manuscript. We have considered each suggestion very carefully and followed the guidance as best as we could. We hope that this revised manuscript is now much improved.

1-a) Make the title simpler, such as “Ybx1 regulates neural lineage specification and neural progenitor self-renewal and differentiation by suppressing PRC2 activity”.

Thank you for this valuable suggestion. We have changed the title to: “Ybx1 fine-tunes PRC2 activities to mediate brain regionalization, modulate self-renewal, and promote neuronal differentiation.”

1-b) Rewrite the running title as “Ybx1 inhibits PRC2 during neurogenesis”.

We have changed the running title to “Ybx1 regulates PRC2 during neurogenesis”. We chose ‘regulate’ over ‘inhibit’ because Ybx1 promotes but also restrains PRC2 binding to chromatin.

1-c) The storyline is based on PRC2, while the findings are based on Ybx1 knockout. I suggest that the authors re- structure the manuscript by focusing on Ybx1, which will be consistent with the experimental evidence.

We have restructured the manuscript as recommended which has much improved the flow of the manuscript. In the introduction section and first results section (Supplemental Figure 1), we explained that our rationale for studying Ybx1 began with JARID2 IP-mass spectrometry. We hope the Reviewer agrees with this decision.

1-d) To assist the reviewer, please add page numbers and line numbers.

Page numbers and line numbers indeed help us identify changes made in the revision. We have made the requested changes.

1-e) Very little has been mentioned on the biological functions of Ybx1 in the introduction. Ybx1 has widespread roles, that are not restricted to transcription factor activity, but also involved in mRNA binding, oncogenesis and so on. The authors should give a brief outline to summarize these roles.

We have now added more information about Ybx1 in lines 72-81 of the introduction section: “Ybx1 is a nucleic acid-binding protein known to affect transcriptional activation, DNA repair and replication, RNA processing and stability, and protein translation^{28,29}. Its multiple molecular roles enable Ybx1 to have wide cellular influences over proliferation, apoptosis, cell differentiation, and cell stress response. For example, Ybx1 binds the consensus sequence 5'-CTGATTGG-3' to mediate transcriptional activation of genes involved in epithelial-to-mesenchymal transition and drug resistance, likely thereby promoting the growth and progression of myriad cancer types³⁰. Its overexpression drives the proliferation of glioblastoma and medulloblastoma cancer cells^{31,32}. Ybx1 depletion results in hematopoiesis failure and exencephaly in the mouse embryo³³⁻³⁵, suggesting a crucial role of Ybx1 in the developing blood and brain.”

1-f) To streamline the story, it might be better to start at the sub-chapter “Ybx1 modulates self-renewal of NPCs”, rather than the mass-spec data of JARDI2 (PRC2). Also, the role of the Ybx1 NLS for regulation of NPC should be written as a separate sub-chapter (starting at “To verify the nuclear role of Ybx1 in self-renewal.....”, and ending at “.....and nuclear Ybx1 is essential for modulating the self-renewal of NPCs”). This should be followed by the sub-chapters “Ybx1 is required for neurodevelopmental gene expression”, and “Ybx1 is required for the differentiation of NPCs to neurons”, respectively. Finally, the authors should move into PRC2, via the sub-chapters “Ybx1 suppresses H3K27me3 by PRC2 at select Ybx1-bound genes in NPCs”, “PRC2 complex binds YBX1 in human stem cells and mouse neural tubes (could be re-written as “YBX1 interacts with PRC2 components”)”, and “Ybx1 exerts its regulation of NPCs in part through PRC2 inhibition”.

We appreciate the Reviewer’s suggestion for section rearrangement. We have incorporated the above suggestions, which has improved the flow of this manuscript. After rearranging sections, we have tracked text changes by the blue font color.

2. I also have some comments on the text.

2-a) In the abstract, move the section “Ybx1 interacts with PRC2 in human and mouse neural progenitor cells (NPCs)”, after “Mechanistically, Ybx1 binds PRC2 gene targets, reduces the levels of PRC2-mediated H3K27me3, and promotes the expression of genes in forebrain specification, cell proliferation, or neuronal differentiation”.

We have rearranged the abstract as suggested, and it now aligns with the organization of the rearranged sections.

2-b) In the abstract, I have some reservation about the statement that “Ybx1 binds PRC2 gene targets”, because Ybx1- to be more precise- binds to “Ybx1 targets”, and likely “suppresses ectopic increase of PRC2-mediated H3K27me3 at these sites”.

We agree with the Reviewer and have changed the statement to “Mechanistically, Ybx1 highly overlaps PRC2 binding genome-wide, controls PRC2 distribution, and inhibits H3K27me3 levels. These functions are consistent with Ybx1-mediated promotion of genes involved in forebrain specification, cell proliferation, or neuronal differentiation.”

2-c) The section “Quantification revealed significantly more BrdU-positive Ybx1-KO NPCs than those in control embryos”, could be simply re-written as “These quantifications showed that there was a significant increase of NPCs in Ybx1-KO embryos”. Also, as the authors have already defined NPCs as “BrdU- positive”, it might not be necessary to always mention “NPCs” as “BrdU-positive NPCs”.

We have simplified the text in lines 114-115. There were 2 mentions of “BrdU-positive” to describe a subset of NPCs. Therefore, we respectfully chose to remain these mentions.

2-d) As it does not directly deal with the neural phenotype, is the following section necessary “For comparison, we quantified the proliferation of mouse embryonic fibroblasts (MEFs) and showed that Ybx1-KO MEFs had decreased proliferation compared to controls (Supplemental Fig. 2e-g). Our findings are consistent with the reduction in proliferation of Ybx1-KO MEFs reported previously 25. These data suggest that Ybx1 has different cellular influences in different developmental contexts: promoting the proliferation of fibroblasts and suppressing the proliferation of NPCs.”?

Respectfully, we believe that this section needs to be included in the manuscript, because it highlights the opposite influences of Ybx1 over the proliferation of MEFs and NPCs as well as brain NPCs and brain cancer cells.

2-e) What is cell-autonomous self-renewal (as in “We next used the neurosphere assay to examine the effect of Ybx1 over cell autonomous self-renewal.”)? Would simple “self-renewal” suffice?

We have deleted the above phrase.

2-f) Please bring supplemental figure 2i in the main figure, as it is important in showing that the NLS plays a role in Ybx1-mediated neurogenesis. Also, in materials and methods, please clearly mention which amino acids were deleted to obtain the NLS-mutant Ybx1.

We have now added the original Supplemental Fig. 2i to now Fig. 1i. We have made minor improvements to the figure panel so it can better fit within the space of Fig. 1.

We apologize for not clearly stating the generation of the delNLS mutation. It is now detailed in the Methods section and includes the information that amino acids 183-205 were deleted. The mutagenesis description is in lines 945-956:

Wild-type YBX1 and YBX1 mutant construct generation

Wild-type human YBX1 was amplified from complementary DNA (cDNA) reverse transcribed from total RNA template of H9 human embryonic stem cells using primers YBX1-F and YBX1-R (Supplemental table 3) which contain *attB* sites for Gateway cloning. The PCR product was used in a BP reaction with pDONR221 and sequence verified by Sanger sequencing. An LR reaction transferred the wild-type YBX1 cDNA to pcDNA-DEST53. All YBX1 mutant constructs were generated by PCR amplification from pDONR221-YBX1 with primers listed in Supplemental Table 3.

Fragments were either immediately used in a BP reaction with pDONR221 or for delF2 and delNLS (deletion of amino acids 183-205) – gel extracted, digested with appropriate restriction enzymes and ligated with Quick Ligase (NEB). After sequence verification, constructs were LR cloned into final destination vectors.

2-g) In the sub-chapter “Ybx1 is required for neurodevelopmental gene expression”, I would suggest that the reviewers focus much more on the genes that are “downregulated”, as these should be the target of the increased PRC2 activity. The description on “upregulated genes” should be minimum, and moved below the section on the “downregulated genes”.

We have rearranged the writing and de-emphasized the discussion of upregulated genes by deleting the sentence “These upregulated genes enriched in functions...”

This section and the next sections are inter-related, so see below for expanded discussion of downregulated genes.

2-h) The authors mention “Our RNA-seq analysis showed that several forebrain lineage genes were downregulated in Ybx1-KO NPCs.” What are these genes, and how do they function in neurogenesis? This is an important part of the story and the authors should take care to explain the fact that Ybx1 is maintaining the activated state of these lineage specific genes, and that in the absence of Ybx1 the same group of genes tend to be silenced by ectopic H3K27me3/PRC2.

We have expanded the discussion on the important downregulated genes in lines 161-165: “Our RNA-seq analysis showed that several crucial forebrain lineage genes were downregulated in Ybx1-KO NPCs. These genes included *Fgf8*³⁹, *Six3*^{40,41}, *Emx2*⁴², *Arx*⁴³, and *Dkk1*⁴⁴, which are required for the patterning and formation of the forebrain. Other downregulated genes included *Hes5*, which is required for NPC proliferation and differentiation to neurons^{45,46}, and *Fezf2*, which is required for neuronal differentiation in the forebrain⁴⁷.”

We have added to lines 170-173: “We infer that reduced forebrain specification in Ybx1-KO embryos led to the overgrowth of the midbrain and hindbrain. We concluded that Ybx1 regulates the expression of genes that are crucial for forebrain patterning and formation, NPC proliferation, and neuronal differentiation.”

We have added to lines 245-249: “Relating our data to the observed developmental defects in Ybx1-KO embryos, we concluded that Ybx1 directly attenuates the deposition of H3K27me3 to promote the expression of genes involved in forebrain specification and neuronal differentiation and that Ybx1 loss leads to H3K27me3 increases and suppression of the same genes.”

2-i) The description on H3K4me3, as it is not related to the main story (Ybx1 versus PRC2), could be much reduced. Please downsize (or remove altogether) the section “H3K4me3 ChIP-seq signals genome-wide were higher in Ybx1-KO than in control NPCs (Fig. 4f). Unlike H3K27me3 level increases genome-wide, H3K4me3 level increases occurred in specific regions. We separated Ybx1-bound regions into 5 categories by intensity and location of H3K27me3 and H3K4me3 level increases, as shown by different average profiles at the top of the heat map in Fig. 4a. For example, H3K4me3 levels increased only at regions in category 4. Further, the numbers of H3K4me3-occupied regions at promoters 3' of transcription start site, exons, introns, termination sites, distal regions (within 50 kb of a coding gene), and intergenic regions all decreased in Ybx1-KO NPCs (Supplemental Fig. 4j, k). Although genes (regardless of Ybx1 binding status) with higher H3K4me3 levels were enriched in upregulated genes in Ybx1-KO NPCs (Supplemental Fig. 4l), we did not observe Ybx1 protein binding with H3K4 methyltransferases, which are represented by the core subunit Rbbp5 (Fig. 1d). These data suggest that Ybx1 does not directly influence H3K4me3. WB analysis of neural tube cells showed genome-wide increases of H3K27me3 and H3K4me3 in Ybx1-KO vs. WT (Supplemental Fig. 4m).

Altogether, our data suggest that Ybx1 binding leads to reduction of H3K27me3 and promotion of gene activation in NPCs”.

We appreciate the Reviewer’s suggestion and have downsized the initial section. However, to address Reviewer 4’s suggestion about analyzing bivalent genes, which are associated with both H3K27me3 and H3K4me3, we have added this analysis after the initial section. Please see pages 30 of this document about this addition.

3. The model proposing an antagonistic relationship between Ybx1 and PRC2 should be clarified more. To this end, the authors should make extensive use of the NGS data that they have generated. In particular, they need to address the following questions.

We appreciate the Reviewer’s detailed suggestions below to expand sequencing data analyses. These suggestions have considerably improved this section.

3-a) Figure 4a shows that TSS binding profile of Ybx1, and supplementary figure 4b shows the distribution profile of the Ybx1 peaks over various genomic compartments such as TSS or TES. Supplementary figure 4b should be brought in to the main text, as it is an important piece of information. Also, this reviewer wonders if the TSS binding sites of Ybx1 overlap with CGIs (CpG islands)? It is well known that CGIs are major targets of polycomb complexes. It would make sense if Ybx1 binds and protects some CGIs (short in length?) from being targeted by PRC2. In the absence of Ybx1, the same CGIs could be bound by PRC2, via the intrinsic affinity of polycomb complexes for CGIs.

We have now presented a clear explanation in the Results and figure legends that Fig. 4a shows all binding profiles of Ybx1, not just TSSs. Lines 196-197 state “The heat map in Figure 4a shows Ybx1 distribution (including transcription start sites) in *Ybx1*-KO and control NPCs.”

We have expanded the discussion of Supplemental Fig. 4c in the main text. In response to Reviewer 4’s suggestion, we have changed the pie chart in Supplemental Fig. 4c to a bar plot. Lines 201-206 include “...58.8% within promoters (within 2kb of transcription start sites; Supplemental Fig. 4c). Of other Ybx1-bound peaks, 12.1% were located in exons, 13.2% in introns, 0.7% in transcription termination sites, 9.4% in 5’ distal (2-50kb) regions, 2.7% in 3’ distal regions, and 3.2% in intergenic (beyond 50kb from a protein-coding gene) regions (Supplemental Fig. 4c). In total, only 15.3% of Ybx1-bound peaks were located distal to a gene.”

We analyzed the overlap of Ybx1-bound regions and then TSSs with a list of mouse CpG islands that is published by Andy Feinberg’s group (Wu et al. 2010 *Biostatistics*).

In the original submission, we separated all Ybx1-bound regions into 5 categories based on H3K27me3 and H3K4me3 distribution (Fig. 4a). The graphical summary of categories 1-5 is in Supplemental Fig. 5d. H3K27me3 levels increased by about 2-fold in all 5 categories. The distribution of H3K4me3 differed in all 5 categories. H3K4me3 increased by about 2 fold in category 4. Category 5 did not have H3K4me3 occupancy. The enrichment of CpG islands in all categories of Ybx1-bound regions and TSSs were significant, with $p < 2.6e-16$.

We have included these results in lines 274-279: “We analyzed CpG islands because PRC2 preferentially binds CpG islands. Nearly all Ybx1-bound regions (99.6%) and TSSs (99.8%) in categories 1-4 overlapped CpG islands (Fig. 5a, Supplemental Fig. 6a). However, 44.9% of Ybx1-

bound regions and 30.5% of Ybx1-bound TSSs in category 5 overlapped CpG islands (Fig. 5a, Supplemental Fig. 6a). This finding suggests that Ybx1 binds CpG islands.”

About the comment “In the absence of Ybx1, the same CGIs could be bound by PRC2, via the intrinsic affinity of polycomb complexes for CGIs.”

Ezh2/1 CUT&RUN-seq in control and *Ybx1*-KO NPCs revealed that *Ybx1*-KO resulted in reduced PRC2 binding to chromatin genome-wide. PRC2 also lost binding to CpG islands. Of note, PRC2 retained binding to Hox clusters (e.g. Hoxa cluster; Supplemental Fig. 6g) in *Ybx1*-KO NPCs.

3-b) The authors should compare the group of Ybx1-bound TSS that are (i) upregulated in Ybx1-KO, (ii) downregulated in Ybx1-KO, and (iii) the rest. What are the levels of H3K27me3 and H3K4me3 in groups (i), (ii) and (iii); before or after of Ybx1 deletion? What are the gene expression levels of (i), (ii) and (iii); in WT and Ybx1-KO?

We agree that this set of comparisons would better reveal the correlation between changes in gene expression, H3K27me3, and H3K4me3. We first generated box plots to validate the expression of Ybx1-bound genes that were unchanged, downregulated, or upregulated in RNA-seq datasets of *Ybx1*-KO and control NPCs (Supplemental Fig. 3b).

In lines 263-268, we added the following: “We next generated the distributions of H3K27me3 and H3K4me3 distribution at transcription start sites of Ybx1-bound genes that were unchanged, downregulated, or upregulated in *Ybx1*-KO NPCs (Supplemental Fig. 5i). As an additional comparison, we generated the same profiles at genes that were not bound by Ybx1; they had little change in *Ybx1*-KO NPCs (Supplemental Fig. 5j). These comparisons showed that Ybx1 KO led to upregulation and increased H3K4me3 primarily at Ybx1-bound genes. H3K27me3 levels increased by approximately 2 fold at all Ybx1-bound TSSs.”

We noticed that the H3K27me3 peak center of downregulated Ybx1-bound genes had lower H3K27me3 levels compared with that in the surrounding nucleosomes in control and Ybx1-KO NPCs. Other profiles did not have such a H3K27me3 distribution. We concluded that Ybx1 potentially influences H3K27me3 distribution at TSSs of genes it promotes the expression of.

Legend: (S3b) Boxplot of FPKM values of genes that were unchanged, downregulated, or upregulated in Ybx1-KO and control NPC RNA-seq datasets. (S5h, i) Average profiles of Ybx1, H3K27me3, and H3K4me3 distribution at Ybx1-bound and other TSSs. Profiles were separated by their gene expression changes in Ybx1-KO NPCs.

3-c) The phenotype of Ybx1-KO is partially rescued in a double KO of Ybx1 and Eed, or upon administration of a chemical compound that inhibits PRC2 activity. Although this shows that Ybx1 inhibits ectopic activity of PRC2, I wonder if the opposite is also true? What is the level of Ybx1 binding in Eed-KO cells or tissues? Does it increase?

We agree that this is an intriguing question, but in our opinion it is tangential to our study's central message. Published studies have shown that Eed and Ezh2 depletion result in markedly different effects on neural development (Zemke et al. 2015 *BMC Biology*; Salmani et al. 2018 *Development*). Rigorously addressing the question of "how PRC2 affects Ybx1 activities" would require extensive characterization of the different subunits, Eed, Ezh2, and Jarid2 in the mouse embryo. Unfortunately, these characterizations will be very time consuming and expensive therefore not feasible.

3-d) It is not clear how, mechanistically, Ybx1 inhibits PRC2 activity. The authors showed that the NLS is required for normal biological roles of Ybx1 during neurogenesis, which is expected as without the NLS Ybx1 show reduced localization to the nuclei. Have the authors checked which Ybx1 domains are involved in the interaction with PRC2, for example by co-IP? This could be done in any relevant cell lines such as hNPC, or even in easier systems such as HEK293 or HeLa.

We performed the protein pull-down assay to confirm that JARID2 and YBX1 proteins bind to each other (Fig. 6e). We then performed JARID2 pull down of different YBX1 deletions to determine that amino acids 1-104 are required for YBX1 binding to JARID2; however, this domain was not sufficient to bind to JARID2 (Fig. 6f). We reasoned that the Jarid2–Ybx1 binding interaction likely requires proper folding of the Ybx1 protein domain. We also noted that other studies of Ybx1 protein interactions encountered similar issues. In Results, lines 362-365 added: "We performed pull-down analyses to

show that JARID2 protein binds to YBX1 protein (Fig. 6e). Pull-down of JARID2 with various myc-tagged fragments of YBX1 uncovered that amino acids 1-104 were required for binding to JARID2 (Fig. 6f). We concluded that YBX1 binds PRC2 in various cell types from mice and humans.”

The exact molecular mechanism by which Ybx1 inhibits H3K27me3 by PRC2 is not clear. We performed the histone methyltransferase assay and found that Ybx1 did not directly affect H3K27 methylation by PRC2 *in vitro*. As this result of the *in vitro* assay were negative, we could not clearly determine how Ybx1 suppresses H3K27 methylation by PRC2. We used Ezh2/1 CUT&RUN-seq in control and *Ybx1*-KO NPCs to show that Ybx1 KO leads to lower levels of Ezh2/1 binding to chromatin genome-wide (see pages 26-28 of this document). These findings suggest that Ybx1 promotes and restrains PRC2 binding to chromatin and inhibits H3K27me3 levels. These activities likely promote the expression of neurodevelopmental genes.

Legend: Western blotting analysis of protein pull-down of His-JARID2 with (6e) GST-YBX1 proteins and (6f) myc-tagged YBX1 deletion mutants. **Methods:** His-JARID2 was expressed and purified from Sf9 cells infected with baculovirus. GST-YBX1 was expressed and purified from the BL21(DE3) bacterial extract. Myc-tagged YBX1 fragments were expressed in S2 cells. Bait protein or control extract (not expressing the bait protein) was pre-bound to beads overnight and mixed with target protein for 3h at room temperature. After high-stringency washes, eluates were obtained and subjected to western blotting analysis.

Reviewer #4 (Remarks to the Author):

Ybx1 negatively regulates PRC2 to affect gene expression programs for neural lineage specification and neural progenitor self-renewal and differentiation

This work seems well written besides few typos. In the paper the authors demonstrate quite clearly the ability of Ybx1 to counteract/offset PRC2 activity in order to specifically favour neural lineage specification, by directly orchestrating the regulation of a set of crucial genes. The Title however is quite elaborated, and I think a bit cumbersome. I would suggest developing a less descriptive and pointier one. In the abstract and along the text the authors stress the importance of distinguishing cell-intrinsic and non-cell autonomous mechanisms, they describe how Ybx1 have different cellular influences in different developmental contexts, and they identify forebrain specific dysregulations: I would put more emphasis to at least one of these aspects.

We thank the Reviewer for appreciating this work and suggesting to emphasize at least 1 the 3 aspects of Ybx1 function: cell-intrinsic mechanism, different influences in different cell/developmental contexts, and forebrain specification.

We have accordingly added the following to the Discussion: “Our study advances the understanding of PRC2 regulation in brain development by uncovering the role of Ybx1. We provide evidence that Ybx1 physically interacts with PRC2 to promote PRC2 binding and proper genomic distribution, but inhibits H3K27me3 levels genome-wide in NPCs. Via PRC2, Ybx1 likely influences gene expression programs for NPC self-renewal modulation and proper neuronal differentiation (Supplemental Fig. 9d)... Taken together, these findings point to a physical and functional interaction between Ybx1 and PRC2, such that PRC2 is “locked and loaded” at co-bound genes in NPCs. An aberrant decrease in PRC2 activity leads to forebrain expansion, whereas an aberrant increase in PRC2 activity in the absence of Ybx1 leads to forebrain reduction. Ybx1 suppresses PRC2 and H3K27me3 levels at forebrain lineage genes in order to promote their proper expression during forebrain specification... Because PRC2 and Ybx1 both have cell and development contexts-dependent influences, their interactions likely differ in forebrain vs. mid/hindbrain. The crucial requirement of PRC2 and Ybx1 for spatiotemporal developmental programs, despite their ubiquitous expression in the neural tube, suggests that additional factors within this network provide developmental specificity in neural stem and progenitor cells.

We used *in vitro* assays for their multiple advantages of validating *in vivo* studies, extending the investigation of *Ybx1*-KO beyond the stage of embryonic lethality, showing that PRC2–Ybx1 participates in a cell-intrinsic mechanism to regulate NPCs, and revealing the different influences of Ybx1 in different cellular or developmental contexts. Although Ybx1 promotes the proliferation of medulloblastoma or glioblastoma cancer cells^{31,32}, it suppresses the proliferation and self-renewal of NPCs.”

PRC2 complex binds YBX1 in human stem cells and mouse neural tube

I find the strategy to track PRC2 bindings very compelling, but it does not seem justified enough, and I could not find in the methods and supplementary material which antibody was used. Moreover, this piece of the paper supports the idea that Ybx1 binds PRC2 but I don't find in the introduction a part that problematize the various forms of PRC2 that have been proposed, and I expected to find here or in the discussion a brief argumentation on the possibility that in the forebrain Ybx1 is part of PRC2 and not just its “negative counteractor”.

We apologize for omitting the antibody information It has now been added: Novus Biological NB100-2214. In the revised manuscript, we have added Supplemental Table 1, which lists all the antibodies used and their dilutions.

About PRC2 variant complexes – detailed in the specific section on page 26.

About Ybx1 being part of PRC2 in the forebrain

We have generated compelling protein-protein interaction data and chromatin profiling data to suggest that Ybx1 is an accessory subunit of PRC2. Further, Ybx1 likely promotes but restrains PRC2 binding to chromatin while modulating H3K27me3 levels in brain NPCs. We did not specifically mention the possibility that Ybx1 is a part of PRC2 in the forebrain, but stated that due to cell type–dependent influences of Ybx1, its interaction with PRC2 likely differs in different regions, i.e., forebrain vs. mid/hindbrain. We stated in lines 452-454 of the Discussion section: “Our protein binding and chromatin profiling results suggest that Ybx1 is an accessory subunit of PRC2. Overall, our data suggest that Ybx1 optimizes PRC2 activity and promotes balanced brain development *in vivo*.”

Ybx1 modulates self-renewal of NPCs

On a general note, I find that the problem of reproducibility is well confronted here. With a further quantification step (exencephaly in Ybx1 mice) with respect to previous literature that fortify the observations. I appreciated the coupling of BrdU assays *in vivo* and in spheroids. I wonder whether a marker different from Sox2 could be used to identify NPCs since NESTIN, for instance, has been used elsewhere in the same work and Sox2 is expressed at even earlier stages of development. Besides graphically improvable Fig.2E-G is very clear and upholds an extensive amount of work. The authors findings on Ybx1 KO effects in different developmental contexts is clearly an important observation that might merit future investigations.

Other than Sox2, we have used an alternative marker, FABP7, to identify NPCs for subsequent assays. We used Neurofluor CDr3 (Yun et al. 2012 *PNAS*; Stem Cell Technologies catalog 01800) to stain FABP7, purify NPCs by FACS, and use these NPCs for Figures 1E-J, 2G, 5, 6, and 7 of the original submission. In the original submission, we described the use of Neurofluor CDr3 for FACS in the Methods section, text for the original Figures 1E-J, and the original Figure 5a that illustrates the differentiation assay. We apologize for not clearly stating the FACS markers that were used for assays in the original manuscript. In the revised manuscript, we have improved our explanation by adding “purified by Neurofluor CDr3 FACS,” for NPCs in lines 157, 178-9, and 412.

Graphs in the now Figure 1F-G (2F-G in the original submission) have been changed to violin plots. We hope that this change has graphically improved data presentation.

Legend: Quantification of the number and area of (f) primary, (g) secondary, and (h) tertiary neurospheres formed by NPCs from Ybx1-KO and sibling control embryos.

Ybx1 is required for neurodevelopmental gene expression

Here I found the Methods for RNA-seq analysis quite poor. The authors apply Voom, using TMM normalization which is supposed to use a stringent algorithm and to provide stringent results, but the number of genes found differentially expressed is notable, and I found no indication of how the genes were filtered. How did the author define a gene as expressed? I would rather see in the methods a precise definition of how the genes were filtered, and in the supplementary a proof that the number of genes differentially expressed does not change dramatically after a certain level of filtering. In the supplementary materials, or at least via correspondence through the editor, a brief benchmark of the effect of selecting Salmon or Kallisto instead of HT-seq. Moreover, while the functional characterization is very useful to defend the main points of the paper, I find that a better characterization might produce a better dissection and serve some clues on the gene regulatory networks underlined by the interference with Ybx1. On a personal note, while showing relevant observations, I find GSEA a very basic level of RNA-seq analysis. Could we conclude from the authors data that “Ybx1 buffers self-renewal”? The combined down-regulation of forebrain lineage genes and up-regulation of mid- and hindbrain lineages genes in KOs seems very relevant. It should be further investigated: for instance, are there motifs of specific transcription factors that are differentially enriched in the promoters or enhancers of those genes? Again, I think it would be easy to do some master regulator analysis or clustering here. The results of these analyses would highly help discuss the differences and insights that the authors obtained from the experiments performed around the questions that they try to answer in this part of the paper.

About RNA-seq

We thank the Reviewer for pointing out the need to improve our explanation of methodology. The Methods section in lines 925-929 states: “RNA-seq was mapped as previously described and HTSeq (version 0.6.1p1)⁵² was used to estimate fragments per kilobase of transcript per million mapped reads (FPKM) based on GENCODE(v24)⁵³. After normalization by trimmed mean of M-values (TMM) and filtering out genes not expressed in both groups (FPKM<1), Voom was used to identify differentially expressed genes.” Results in lines 142-143: “We analyzed ‘expressed’ genes as those having FPKM values>1 in either control or *Ybx1*-KO NPCs.”

We had performed RNA-seq of expressed genes by filtering out genes whose FPKM values were <1 in wild-type and *Ybx1*-KO. We chose this criterion because FPKM=1 corresponds to ~1 mRNA molecule per cell (Mortazavi et al. 2008 *Nature Methods*). To test the robustness of this FPKM=1 cutoff, we had tried other cutoffs of FPKM=0.5 and FPKM=2. Although the numbers of expressed genes became reduced with increasing cutoffs (~14,000 genes with FPKM > 0.5, ~12,000 genes with FPKM > 1, and ~10,000 genes with FPKM > 2), differentially expressed genes did not change significantly. Please see the left graph below, with R2=0.9995 between datasets using FPKM>1 vs. FPKM>2).

We tried Kallisto for RNA-seq analysis. As shown in the right graph below, results from Kallisto and Voom with TMM did not differ significantly. The second graph compared Kallisto analysis and Voom with TMM, yielding R2=0.8049. We detected a few genes specific to Kallisto analysis; these genes are indicated in the right graph below. These few genes did not appear to have biological relevance to our study. A previous publication showed that Kallisto did not separate paralogous genes whose sequences were highly similar (Ballouz et al. 2018 *Nucleic Acids Research*). Due to these reasons, we chose not to present our results using Kallisto.

Legend: The graph on the left compares RNA-seq using Cutoff with RPKM <1 with cutoff with RPKM <2. The graph on the right compares RNA-seq using Kallisto with Voom with TMM normalization.

About the comment, “Could we conclude from the authors data that “Ybx1 buffers self-renewal”? The combined down-regulation of forebrain lineage genes and up-regulation of mid- and hindbrain lineage genes in KOs seems very relevant.”

We hesitate to make the conclusion that Ybx1 “buffers” the self-renewal of NPCs. We agree that when considering other published findings about the effect of Ybx1 on the proliferation of MEFs and cancer cell lines, Ybx1 appears to buffer cell proliferation. Because our study focuses on Ybx1 in NPCs, we respectfully want to stick with the conclusion that Ybx1 modulates the self-renewal of NPCs.

We agree that downregulation of forebrain lineage genes coupled with upregulation of midbrain and hindbrain lineage genes in *Ybx1*-KO is an important finding. Throughout the manuscript, we have discussed this finding in terms of Ybx1 suppressing H3K27me3 to promote the expression of forebrain lineage genes. In the *Ybx1*-KO brain, the upregulation of midbrain and hindbrain lineage genes is likely a consequence of the expansion of midbrain and hindbrain.

About the motif search – please see the specific section below on page 24.

Ybx1 suppresses H3K27me3 by PRC2 at select Ybx1-bound genes in NPCs.

I found crucial the use of CUT&RUN to identify Ybx1 binding sites, but it is not obvious to me why ChIPseq was preferred for H3K27me3. I suppose it is just a matter of when the two experiments were performed but I find it a bit inconsistent, since CUT&RUN is significantly less expensive and one could have performed it anyway, to then use H3K27me3 peaks found in ChIP-seq and CUT&RUN to benchmark their ability to produce comparable tracks. Moreover, since IgG were obtained from the lines, I highly recommend performing peak calling on CUT&RUN data using SEACR. Given the difference between MACS2 and SICER I would also recommend showing somewhere in the supplementary what is the merit of performing peak-calling with those two methods and merge the outputs. What parameters were used by the authors with SICER and MACS2? I think it must be stated more clearly in the methods. Finally, for what concerns the differential deposition of histone marks, I did

not understand why TMM-normalization was used, what tools were used to quantify the ChIP-seq reads occupation over the genome, and on what subset of regions the quantification was performed. I suppose the author used deepTools but I would love to have a better description in the methods. A correct normalization is necessary to call differential mark deposition. Moreover, I would have expected the authors to take advantage of available enhancer references for NPCs and neuronal lineages, to associate differentially marked regions with differentially expressed genes. This process, which I don't find described in the paper, could have led to more mechanistic and specific dissections of the observed dysregulations. It is clear from the authors analysis that Ybx1 binds preferentially promoters at very close distance from the TSS, but what about the rest of the genome? Considering the well-known relevance of PRC2 in the regulation of bivalent promoters, and their regulation relevance in developmental processes, what is the proportion of bivalent promoters bound by Ybx1? Having ChIP-ped both H3K4me3 and H3K27me3 seems like an immediate link to bivalency. The authors write: "There was a strong correlation between downregulated gene expression and increase in level of H3K27me3 (nearly passing a more stringent FDR criterion; Fig. 4d,e)", I have no particular issues with presenting data not passing a stringent FDR, but I would have expected some ChIP-qPCR to validate some of the genes/region reported in Fig 4d, since from the picture the anyway expected phenomenon of general down-regulation of "hyper-H3K27threemethylated" genes - and vice-versa - does not appear as evident for the highlighted genes.

We did perform ChIP prior to CUT&RUN. To address comments here, we used H3K27me3 CUT&RUN-seq to benchmark H3K27me3 ChIP-seq. Supplemental Fig. 5a (also right) shows all H3K27me3-occupied regions by CUT&RUN-seq and ChIP-seq. We examined all H3K27me3-occupied regions, not just peaks (done for SEACR analysis); therefore, the y-axis values are not high. Regions in H3K27me3 ChIP-seq and CUT&RUN-seq highly overlapped, and there were similarly genome-wide H3K27me3 increases in *Ybx1*-KO NPC. We used the new H3K27me3 CUT&RUN-seq to generate an MA plot (Supplemental Fig. 5d) that validate regions with H3K27me3 increases in Fig. 4d.

Legend: (S5A) Heatmap of H3K27me3 peak region distribution by CUT&RUN (CR in graph)-seq and ChIP-seq in control and *Ybx1*-KO NPCs. (S5d) With replicate datasets, we were able to identify significant changes, red dots, with the criteria of FDR-adjusted $p < 0.05$ and fold change > 2 .

About SEACR – please see the specific section below on pages 24-25.

About parameters used for SICER and MACS and rationale for TMM normalization

Parameters are stated in the specific section below.

Our rationale to use TMM normalization was that for assessing genome-wide changes in H3K27me3, we asked whether a subset of regions increased more or increased less when compared with the entire genome. TMM-normalization can likely help us better answer that question, because it only uses the less variable regions for

S5a

S5d

normalization. Previous studies show that it is a common practice for ChIP-Seq analysis (Lun et al. 2016 *Nucleic Acids Research*; Pflueger et al. 2018 *Genome Research*).

About neural enhancers

We used active and poised enhancer signature lists from the embryonic brain at E14.5 that were downloaded from the ENCODE portal (<https://www.encodeproject.org/>) with the identifiers ENCFF682UGG, ENCFF386QNM, and ENCFF998FUV. Active enhancers were defined by their co-occupation by H3K4me1 and H3K27ac. Poised enhancers were defined by their co-occupation by H3K4me1 and H3K27me3. Co-occupation or binding was determined by the criterion of 1-nt overlap between regions. Of note, most of the Ybx1-bound regions with enhancer signatures located within 2kb of TSSs. For example, 3717 Ybx1-bound active enhancers were located 2kb-proximal to TSSs, but only 378 Ybx1-bound active enhancers were located 2kb-distal to TSSs. We then analyzed the association of Ybx1 to active and poised enhancers. We found that 18.4% (4235/23070) and 14.5% (3352/23070) of Ybx1-bound regions overlapped active and poised enhancers, respectively. Further, 12.1% (4095/33934) of active enhancers were bound by Ybx1. In contrast, 29.1% (3782/9229) of poised enhancers were bound by Ybx1. Enrichment of Ybx1 binding to poised enhancers suggests a potential functional association of Ybx1 with poised enhancers.

We next determined how Ybx1-bound enhancers associated with differentially expressed genes in *Ybx1*-KO NPCs. We used the distance of 50 kb between enhancers and genes as the criterion for their association. Of Ybx1 target genes that were upregulated in *Ybx1*-KO NPCs, 425 had active enhancers and 339 had poised enhancers in the wild-type brain. Of Ybx1 target genes that were downregulated in *Ybx1*-KO NPCs, 271 had active enhancers and 115 had poised enhancers in the wild-type brain. Our data suggest that Ybx1 suppresses H3K27me3 to promote gene expression. Therefore, we focused on Ybx1-bound active enhancers that became downregulated in *Ybx1*-KO NPCs. These genes were enriched in functions related to nervous system development, synapses, axonogenesis, axon guidance, and dendrite morphogenesis (Supplemental Fig. 6c). Gene ontology analysis of other Ybx1-bound

enhancer types are presented here for the Reviewer. Ybx1 is potentially involved in the promotion of active enhancers for expression of genes involved in neural development, dendrite morphogenesis, axonogenesis, axon guidance, and synapse formation or function.

S6c

Legend: Gene ontology analysis identifying ontology terms of Ybx1-bound enhancers of differentially expressed genes in *Ybx1*-KO vs. control NPCs. Terms were ranked by *p*-value significance, with the number of enriched genes indicated.

About bivalent promoters – please see the specific section below on page 30.

Ybx1 is required for the differentiation of NPCs to neurons

The authors use a very fast differentiation assays which do not produce very pure homogeneous cultures, in a context in which differentiation is put to test. I could not identify how many differentiation replicates were performed to compensate for such variability. The number of neurons screened seems anyway compatible with the aim of the authors. I am not convinced that one-sided Student t test is the best tool to verify that the evaluate the differences in fig. 5e-g but I reckon it is a common practice. A non-parametric test could be appreciated in this part. However, the effect seems striking and I am convinced that increasing the number of observations or changing the statistical tests would not change the outcome of this inquiry.

We appreciate the Reviewer's point. Our goal was to generate a heterogeneous cell population to better assess the effect of Ybx1 on NPC differentiation to neurons or glia in an unbiased manner.

We performed the differentiation tests twice, each time with 2 control and 2 *Ybx1*-KO lines for Fig. 5 in the original submission. Chemical rescues for Fig. 6 were repeated twice, each time with 2 control and 2 *Ybx1*-KO lines. Genetic rescues for Fig. 7 were performed with embryos from 2 genetic crosses. Due to data variance between the differentiation assays, we chose to present data from 1 set of differentiation.

Using the Prism software, we used a one-way ANOVA and unpaired t test with the Holm-Sidak correction to analyze data in the original Fig. 5e-g. We corrected this error in the figure legends of now Fig. 3, 7, and 8: "one-way ANOVA or unpaired t test with Holm-Sidak correction."

Ybx1 exerts its regulation of NPCs in part through PRC2 inhibition.

I find this part of the paper relevant and potentially crucial for the definition of Ybx1 contribution to the modulation of gene expression exerted by PRC2. The rescue of forebrain lineage genes and decreased expression of "most" midbrain and hindbrain lineage genes sounds a bit as an overstatement. I would have expected the authors to state more plainly that the expression levels of selected (standard, trustworthy, et similia) markers of the three lineages had been rescued via the chemical treatment of *Ybx1* KO lines with GSK126. It has been shown in literature that EZH2 inhibition alone reduce self-renewal and favours differentiation. I would have expected to see (together with DMSO- *Ybx1*KO and DMSO-*Ybx1*KO-GSK126) DMSO-GSK126 and DMSO- treated lines. While I am convinced that the observed effects are robust, I am afraid that what the authors are observing is not necessary a rescue phenotype. I would have expected the authors to check the expression levels of CDKN1A and RELN, two crucial targets/interactors of EZH2 in neurogenesis, in these lines.

The further conditional depletion of EED helps supporting the claim that the evidences collected in the previous part of this section are related to a partial rescue of the correct balance between Ybx1 and PRC2, but I wonder what mechanism is exactly bringing up the effects that the authors observe, in a context in which neither Ybx1 nor PRC2 are effectively working/present. Does Ybx1 "simply" binds the promoters to prevent PRC2 interactions with the regulatory regions bound by Ybx1? H3K27me3 is a good proxy of PRC2 activity and genomic targets, but a ChIP-seq or CUT&RUN of EZH2 would help answer this question. An alternative solution I envision is to ChIP-seq Jarid2, with the same antibody used in the first section of the paper, for the sake of consistency.

We agree with the Reviewer that we need to be more careful with making conclusive statements. We have accordingly changed lines 379-382: "Compared with dimethyl sulfoxide (DMSO) treatment, GSK126 treatment of *Ybx1*-KO NPCs significantly increased the expression of assayed forebrain

lineage genes (Fig. 7c and Supplemental Fig. 8a) and decreased the expression of most of the assayed midbrain and hindbrain lineage genes (Supplemental Fig. 8b).”

About DMSO and GSK126 treatment of control NPCs and differentiation – please see the specific section below on pages 29-30.

About CDKN1A and RELN – please see the specific section below on pages 28-29.

About CUT&RUN-seq of a PRC2 subunit in control and Ybx1-KO NPCs – please see the specific section below on pages 26-28.

I find that the final sentence of the paper would fit as a better title (“Ybx1 inhibits PRC2 to mediate brain regionalization, modulate self-renewal and promote neuronal differentiation”)

We thank the Reviewer for point out this more suitable title. We have modified this title to better describe findings from the Ezh2/1 CUT&RUN-seq.

Review Conclusions

I consider the paper to be very close to being appropriate for publication in Nature Communications. However, I consider some correction necessary and some useful to support the claims of the paper. Necessary:

As the Reviewers’ detailed comments and summary below do not completely overlap, we have responded to comments in the sections above and below.

1) To specify clearly what JARID2 antibody was utilised

Information of the antibody used, Novus Biological NB100-2214, is now in Supplemental Table 1.

2) To perform a deeper analysis of RNA-seq data, starting by performing GO enrichments using a background/universe (the total list of expressed genes vs the differentially expressed genes can be done trivially and with several tools), and eventually by, for instance, performing some basic clustering of differentially expressed genes by z-scores or logFC.

In the original submission, we had performed gene ontology (GO) enrichment by comparing differentially expressed genes (now Fig. 2d, 2e) and Ybx1-bound genes (now Supplemental Fig. 4d, e) against all expressed genes.

In the revised manuscript, we have performed basic clustering of differentially expressed genes by Log2(fold change). We have added this information in lines 218-221: “We performed unsupervised clustering of differentially expressed genes in these functional categories [GO of differentially expressed Ybx1-bound

Legend: Unsupervised clustering of log2(fold change) of differentially expressed Ybx1-bound genes in control and Ybx1-KO NPCs.

genes] to effectively separate control and *Ybx1*-KO NPCs (Supplemental Fig. 4f), further supporting that *Ybx1* directly regulates these neurodevelopmentally important genes.”

3) To perform TF/motif enrichment analysis on *Ybx1* ChIP-seq to understand if *Ybx1* has potential partners that might have been pulled down with it in the initial Jarid2immunoprecipitation.

We used Homer2 (v4.9.1) to search for motifs in genes that were differentially expressed by the following: (a) upregulated genes in *Ybx1*-KO, (b) downregulated genes in *Ybx1*-KO, and (c) upregulated genes vs. downregulated genes. The criteria used for differential analyses were FDR-adjusted $p < 0.05$ and fold change > 2 . We also performed a separate analysis with the more relaxed criterion of fold change > 2 to obtain the same results.

We next compared the identified motifs of proteins with JARID2 IP-mass spectrometry datasets. These analyses revealed that the only protein whose motif was enriched was the YBX1's DNA-binding motif with $p < 1e-6$. This motif was enriched in promoters of upregulated genes in *Ybx1*-KO vs. control.

TGCCCAACATCC (Supplemental Figure 6d)

We next used the ChEA (genes bound by specific proteins by ChIP-seq or ChIP-chip) function of Enrichr to analyze genes that contained this consensus motif. We found that 132 Suz12-bound genes were enriched in these genes. Although we could not identify any master regulator, our results present an additional support that *Ybx1* preferentially binds the promoters of PRC2-bound genes and regulate their expression.

We have added lines 304-312 to the Results: “To identify other potential factors involved in *Ybx1*-mediated suppression of H3K27me3, we searched for motifs in genes that were differentially expressed in *Ybx1*-KO vs. control NPCs. This approach identified the enrichment of a consensus DNA motif for *Ybx1* with $p < 1e-6$ (Supplemental Fig. 6d) in promoters of genes that were upregulated in *Ybx1*-KO NPCs, consistent with our observation that known *Ybx1* target genes were upregulated in *Ybx1*-KO NPCs (Fig. 2e). On the other hand, of 382 *Ybx1*-bound genes that were upregulated in *Ybx1*-KO NPCs, 132 are known to be bound by PRC2 ($p = 2.3e-18$ in Suz12 ChIP-seq datasets by using ChEA of Enrichr⁵⁰). These results additionally support that *Ybx1* binds promoters of PRC2-bound genes and regulates their expression.”

4) In the CUT&RUN analysis: to apply SEACR and to verify that SICER and MACS2 are not including too many false positives per se and even more after merging their outputs.

We thank the Reviewer for pointing out the possibility of using SEACR. We are aware that SEACR is a newly optimized method for CUT&RUN peak calling. We have performed H3K27me3 CUT&RUN-seq and tested SEACR, but did not find it suitable for our needs. For example, H3K27me3 peaks called by SEACR were 10 times more and overlapped almost every peak called by MACS2 or SICER. Peaks called by MACS and SICER were more consistent than by SEACR. H3K27me3 distribution tends to be broad and can confound the peak-calling software more suitable for sharp binding sites of transcription factors. The heat map on the right shows H3K27me3 peaks commonly called by the 3 softwares and the ones specific to SEACR.

Further analysis of SEACR-specific H3K27me3 peaks showed that they had low H3K27me3 signals and were much weaker than the common peaks. Although SEACR was disadvantageous when analyzing H3K27me3 CUT&RUN-seq, we think SEACR serves as a tool to analyze CUT&RUN-seq datasets. We have updated our results by further filtering by SEACR to generate more reproducible

peaks. We have also changed the Methods section accordingly. These changes have not affected the biological conclusions.

Legend: Average profiles and heatmaps of H3K27me3 peaks called by SICER, MACS2, and SEACR.

5) To specify in the methods the filtering approach applied to RNA-seq prior differential expression analysis

Changes to Methods were in lines 925-929: “RNA-seq was mapped as previously described and HTSeq (version 0.6.1p1)⁶² was used to estimate fragments per kilobase of transcript per million mapped reads (FPKM) based on GENCODE(v24)⁶³. After normalization by trimmed mean of M-values (TMM) and filtering out genes not expressed in both groups (FPKM<1), Voom was used to identify differentially expressed genes.”

6) To specify in the methods the parameters and conditions used for quantitative analysis of ChIP-seq and CUT&RUN.

We improved this Methods method:

Deep Sequencing Analysis of Chromatin Immunoprecipitation or CUT&RUN

50bp single-end reads for ChIP-seq or 50bp paired-end reads for CUT&RUN-seq were obtained and aligned to mouse genome assembly mm10 and fruit fly genome assembly dm6 by BWA (version 0.7.12, default parameter). Duplicated reads were then marked by Picard (version 1.65 [1160]). For ChIP-seq, uniquely mapped reads were retained by SAMtools (parameter “-q 1 -F 1024”, version 1.4), and quality control was ensured by following ENCODE criteria⁶⁸. We extended reads to fragment size estimated by SPP⁶⁹ and then normalized to dm6 sequencing reads number (fruit fly S2 cells). Peaks were inspected on IGV⁷⁰ and peaks for each replicate were called by MACS2 (version 2.0.9 20111102 option “nomodel” with “extsize” defined as fragment size estimated by SPP)⁷¹ and SICER (redundancy threshold 1, window size 200bp, effective genome fraction 0.86, gap size 600bp, FDR 0.00001 with estimated fragment size defined by SPP)⁷² and then merged merged by bedtools (version 2.17.0), remove a SICER peak if it overlaps MACS2 peaks. For Cut&Run, properly paired uniquely mapped reads were retained by SAMtools (parameter “-q 1 -F 1804”, version 1.4), sorted by name and converted to bedpe format by bedtools. Only fragments shorter than 2000 bp were kept to peak calling and generating of bigwig tracks. Similarly, MACS2 (bedpe mode) and SICER were used for peak calling. To find reproducible peaks, we first called peaks for each replicate twice with an FDR corrected p-value cutoff of 0.05 as a high-confidence peak and an FDR-corrected p-value cutoff 0.5 as low-confidence peak set. We only considered high-confidence peaks that also overlap at least low-confidence peaks in other replicates as reproducible peaks. We did the same by employing SEACR⁷³, using the top 1 percentile as the high-confidence peak and the top 5 percentile as the low-confidence peak (“non stringent” mode) and further require the reproducible peaks overlap SEACR reproducible peaks. deepTools⁷⁴ was used to plot the heatmap. Peak overlap was tested by the hypergeometric test with assumed binomial distributions. Gene ontology analysis was generated by Enrichr⁵⁰. For histone modifications, the quality control standard was set as previously described⁷⁵. Voom⁷⁶ was used to test significant differences between control and Ybx1-KO after TMM⁷⁷ normalization. For enhancer analysis, we used active and poised enhancer signature lists from the embryonic brain at E14.5 that were downloaded from the ENCODE portal (<https://www.encodeproject.org/>) with the identifiers ENCFF682UGG, ENCFF386QNM, and ENCFF998FUV.

Highly suggested:

1) To expand in the introduction the description of known PRC2 complexes

In the introduction, we added in lines 61-66: “PRC2 exists as 2 variant complexes that are defined by their accessory subunits: PRC2.1 associates with subunit EPOP or PALI1¹⁸⁻²⁰, whereas PRC2.2 associates with JARID2 and AEBP2²¹. These accessory subunits and others likely have different and nonoverlapping influences over the PRC2 complex’s binding to chromatin, methylating H3K27, and/or suppressing genes. Different influences of these accessory subunits over the PRC2 variant complexes result in distinct roles in developmental gene regulation.”

2) To perform CUT&RUN or ChIP-seq on EZH2 or other PRC2-specific subunits

We highly appreciate the Reviewer’s suggestion. Data generated in response to this comment have greatly contributed to the mechanistic understanding of Ybx1 over the PRC2 complex. As there is no available antibody that specifically recognizes Ezh2, we used an antibody that recognizes Ezh2 and Ezh1 (Active motif 39875) to profile PRC2 binding on chromatin.

We have added lines 313-347 of Results: “We next examined how Ybx1 affects the expression and chromatin binding of PRC2. The expression of PRC2 subunits did not differ between control and *Ybx1*-KO NPCs (Supplemental Fig. 6e, f), suggesting that Ybx1 does not affect the levels of PRC2 subunits. To examine the genomic distribution of PRC2, we performed CUT&RUN-seq of the enzymatic subunits of PRC2, Ezh2/1, in sibling control and *Ybx1*-KO NPCs (by Sox2-GFP FACS). Ezh2/1 distribution at the *Hox* a cluster paralleled H3K27me3 distribution and remained unchanged in *Ybx1*-KO NPCs (Supplemental Fig. 6g). Moreover, Ezh2/1 occupancy was significantly lower at the forebrain lineage genes *Foxg1*, *Neurod2*, and *Satb2* (Fig. 5c) and cell proliferation genes *Lin28a* and *Hmga2* (Supplemental Fig. 6h), whereas H3K27me3 levels were increased. Also, Ezh2/1 localized to an ectopic (only in *Ybx1*-KO but not in control NPCs) PRC2-bound region that showed increased H3K27me3 in *Ybx1*-KO NPCs (Supplemental Fig. 6i). We concluded that at these genes, PRC2 binding decreased but H3K27me3 levels increased in *Ybx1*-KO vs. control NPCs.

Extending our analysis genome-wide, and using criteria of FDR-corrected $p < 0.05$, we identified 43,469 Ezh2/1-bound regions in control NPCs and 10,902 Ezh2/1-bound regions in *Ybx1*-KO NPCs. Using this criterion, 15,036/43,469 (34.5%) Ezh2/1-bound regions were found to overlap with Ybx1-bound regions in control NPCs. With a more relaxed criterion of $p < 0.05$, 29,068/43,469 (66.9%) Ezh2/1-bound regions were found to overlap Ybx1-bound regions. We generated heat maps to compare Ybx1 and Ezh2/1 distribution at all Ybx1-bound regions (Fig. 5d) or all Ezh2/1-bound regions (Fig. 5e) in control and *Ybx1*-KO NPCs. In control NPCs, Ezh2/1 and Ybx1 binding patterns were highly similar and likely overlapped by more than 66.9%. These data suggest that Ybx1 is a potential accessory subunit of PRC2 at most PRC2-bound regions.

Furthermore, we found that Ezh2/1 binding at most regions genome-wide was reduced in *Ybx1*-KO NPCs compared to control NPCs (Fig. 5e). Ezh2/1 gained binding at ectopic regions (“specific to *Ybx1*-KO” in Fig. 5e). We could not attribute functional significance to ectopic Ezh2/1-bound regions as they were not associated with differentially expressed genes in *Ybx1*-KO NPCs. We compared H3K27me3 and H3K4me3 distribution to Ezh2/1 distribution in control and *Ybx1*-KO NPCs. H3K27me3 levels increased at all regions with reduced Ybx1 and Ezh2/1 binding in *Ybx1*-KO NPCs (H3K4me3 remained unchanged; Fig. 5e). At the ectopic Ezh2/1-bound regions in *Ybx1*-KO, H3K27me3 levels increased and H3K4me3 levels remained undetectable (Fig. 5e). Collectively, our data suggest that Ybx1 binding promotes PRC2 binding at many sites while restraining its activity including H3K27 methylation. This

balancing/fine-tuning of PRC2 binding and histone modification levels likely facilitates precise spatiotemporal gene regulation required for neural development.”

Supplemental Figure 6 – Ybx1-bound genes and Ezh2/1 CUT&RUN-seq in control and Ybx1-KO NPCs. (e) FPKM values of PRC2 subunits and Ybx1 in control and Ybx1-KO RNA-seq datasets. (f) WB analysis of control and Ybx1-KO NPCs. Ezh2/1 and Ybx1 CUT&RUN-seq tracks and H3K27me3 ChIP-seq tracks at (g) *Hox A* cluster, (h) *Lin28a* and *Hmga2* loci, and (i) an ectopic Ezh2/1-bound region in control and Ybx1-KO NPCs. n.s. and * indicate not significant and $p < 0.05$ by one-sided Student's *t* test.

Figure 5. (c) Ezh2/1 CUT&RUN-seq tracks H3K27me3 and H3K4me3 ChIP-seq and at *Foxg1*, *Neurod2*, and *Satb2* gene loci in Ybx1-KO and sibling control NPCs. (d) Heat maps of Ybx1 and Ezh2/1 distribution within 5kb of Ybx1 CUT&RUN-seq peaks. (e) Heat maps of Ezh2/1, Ybx1, H3K27me3, and H3K4me3 within 5kb of Ezh2/1 CUT&RUN-seq peaks.

e Ezh2/1-bound regions — profile in control
 — profile in Ybx1-KO only

3) To verify expression levels of *CDKN1A* and *RELN* along differentiation and upon PRC2 inhibition in *Ybx1* KO lines

Previous work has shown that PRC2 controls *Cdkn1a* (Fan et.al. 2001 *Mol Cancer Res*) and *Reln* (Zhao et.al. 2015 *Scientific Reports*) expression in the brain through deposition of H3K27me3. *Ybx1* KO is expected to result in downregulated expression of *Cdkn1a* and *Reln*. Consistent with this expectation, *Cdkn1a* and *Reln* expression was significantly downregulated in *Ybx1*-KO compared with sibling control NPCs (Supplemental Fig. 8e). During NPC differentiation, *Cdkn1a* levels did not change in *Ybx1*-KO differentiating cells (Supplemental Fig. 8e), likely because its level was already quite low in *Ybx1*-KO NPCs. In a differentiation assay that enables neuronal and glial differentiation, *Reln* expression was not readily detectable in these cells (CT values >45 in qPCR analyses; Supplemental Fig. 8e).

After GSK126 treatment of wild-type NPCs, *Cdkn1a* and *Reln* levels were upregulated and as expected. GSK126 treatment of *Ybx1*-KO NPCs led to robust upregulation of *Cdkn1a* and *Reln* (Supplemental Fig. 8f), which was as expected. During NPC differentiation, GSK126 treatment affected *Cdkn1a* expression (Supplemental Fig. 8h), and *Reln* was not detectable. Overall, results from GSK126 treatment of control and *Ybx1*-KO NPCs support the PRC2–*Ybx1* functional interaction.

Eed depletion is expected to result in the upregulation of *Cdkn1a* and *Reln* in *Ybx1*-KO NPCs. Although Eed depletion did lead to robust upregulation of *Reln*, *Cdkn1a* was downregulated in *Ybx1*-KO NPCs (Supplemental Fig. 8i).

4) To add DMSO-wildtype and DMSO-wildtype-GSK126 lines in the PRC2 inhibition experiments

We had treated control NPCs with DMSO and GSK126 in the experiments performed in the original submission. In the now Supplemental Fig. 8, we presented these data. We chose not to combine data from the 4 experimental groups, because of the number of assayed genes made the graphs very difficult to interpret. GSK126 treatment decreased the expression of some forebrain lineage genes and increased the expression of some mid- and hindbrain lineage genes in wild-type NPCs (Supplemental Fig. 8c, d). While these results are not consistent with the expectation that PRC2 inhibition by GSK126 would mimic the effect of Ezh2 depletion in the brain (Zemke et al. 2015 *BMC Biology*), it is well known that pharmacological inhibition and genetic deletion do not always have the same effects. GSK126 treatment largely had modest to little effect on the expression of most neuronal and glial genes in differentiating control cells (Supplemental Fig. 8g). These results revealed that GSK126 treatment had the opposite effects on control and Ybx1-KO NPCs. We wrote in lines 380-384 to “In contrast, GSK126 treatment of control NPCs led to decreased expression of some forebrain lineage genes and increased expression of some midbrain and hindbrain lineage genes (Supplemental Fig. 8c, d). These data suggest that PRC2 inhibition in Ybx1-KO NPCs leads to the reactivation of forebrain lineage genes and suppression of midbrain and hindbrain lineage genes.” We have changed lines 407-409 to “GSK126 treatment of differentiating control cells largely had modest to little effect on the expression of most neuronal and glial genes (Supplemental Fig. 8g).”

Legend: (S8c-d) RT-qPCR with TaqMan assays in control NPCs treated with DMSO or 500 μ M GSK126 after 4 days. RT-qPCR with TaqMan assays of (S8g) neuronal genes and glial genes in control NPCs treated with DMSO or 500 μ M GSK126 and differentiation media for 14 days. n.s., not significant. *, **, ***, and **** indicate $P < 0.05$, 0.01, 0.001, and 0.0001, respectively, by one-way ANOVA with Holm-Sidak correction.

5) To discuss bivalent regions.

We appreciate the Reviewer's comments. In lines 280-289 of the Results section, we added our findings. "We investigated whether Ybx1 binds to bivalent regions, which are co-occupied by H3K27me3 and H3K4me3. Using the 1-nt overlap criterion, we determined that Ybx1 bound to 4344/7953 (54.6%) of bivalent regions in NPCs (Fig. 5b). This strong enrichment prompted us to determine whether Ybx1 KO led to differential expression of bivalent genes. Compared with sibling control NPCs, 179 bivalent genes were downregulated and 212 bivalent genes were upregulated in *Ybx1*-KO NPCs (Supplemental Fig. 6b), suggesting Ybx1 affects the expression of bivalent genes. As transcription at bivalent genes is generally low and so was not expected to be downregulated, we examined RNA-seq profiles of some of the 179 bivalent genes. At most of these genes, RNA-seq reads were restricted to the transcription start sites in control NPCs and reduced further in *Ybx1*-KO NPCs."

Legend: (Fig 5b) 54.6% of bivalent (H3K27me3 and H3K4me3) regions were Ybx1-bound. (Fig S6b) Fisher's test showed enrichment of Ybx1 binding in downregulated bivalent genes in *Ybx1*-KO NPCs.

6) Supplemental Figure 4: Pie chart are deprecated for the depiction of genome-wide peaks distribution: I highly suggest substituting them with barplots. **S4b**

We have changed the data presentation to a bar plot.

Reviewer #1 (Remarks to the Author):

The authors have done a good job addressing my main concerns. However, there are still serious issues with the new quantitative mass spectrometry data that the authors provide in the revised manuscript. The volcano plots that are presented look strange and parameters shown in the plots (i.e. Corr, R2 value etc) make no sense at all. Furthermore, volcano plots require triplicates (triplicates for IgG and specific pull-down) to enable T test statistics, how did the authors generate these plots with duplicate pull-downs? The authors are encouraged to study relevant interaction proteomics papers and their methods on this topic and adjust their experimental setup and data visualisation accordingly. The S7b figure also has serious problems: how can an interacting protein for Jarid2 (Ybx1) be much more enriched than the bait protein itself, is this indicative of antibody cross-reactivity? How much of the input samples did the authors load on the western blot shown in Figure 6C? Assuming this is 10% and looking at the amount of Ybx1 precipitated from the input, at most 1% of cellular Ybx1 is in complex with Jarid2. A similar pattern can be seen in Figure 6B, where the large majority of Jarid2 appears not to be bound to Ybx1. Thus, it would be important to acknowledge that the interaction between PRC2 and Ybx1 is highly substoichiometric and I would not use the term 'accessory subunit' as is used for PCL proteins, for example. As a consequence, some of the observed effects on chromatin and the interplay between Ybx1 and PRC2 in vivo may be indirect.

Reviewer #2 (Remarks to the Author):

I appreciate the thoughtful discussion and the additional experiments that the authors performed to address my concerns since the initial submission. The authors addressed most of our questions plus many more from the other Reviewers. While they did not use CRISPR to target Ybx1, the authors acknowledge these would be necessary but challenging experiments beyond the scope of this study. Overall, I think this paper contributes to our knowledge of the role of Ybx1 in the regulation of PRC2 in brain development and neuronal differentiation.

Reviewer #3 (Remarks to the Author):

The revised version has been improved by additional experiments and data analyses. The paper also reads better than the previous version. The authors have sufficiently answered the concerns raised by this reviewer.

I have the following minor comments to the authors.

1. The title, as pointed out by other reviewers too, still looks cumbersome and somewhat difficult to understand. Given that the authors have such a nice story, and have performed quite a few elegant experiments, wouldn't it be better to make the title simpler and make their paper more accessible to the readers?
2. I would suggest bringing the model (now in supplementary figure 9E) to one of the main figures (last panel in Figure 8?), as it nicely sums up the main findings of the story.
3. The title of the sub-chapter (page 15, line 367) could be simplified as "Ybx1-mediated regulation of NPCs involves inhibition of H3K27me3".

The model that Ybx1 facilitates PRC2 binding but inhibits H3K27me3, although counter-intuitive, is very interesting. Usually, this would suggest that Ybx1 also interacts with other activator complexes- such as H3K4me3- to counterbalance H3K27me3 deposition. The authors, intriguingly, did not see any protein-protein interactions between Ybx1 and H3K4me3 (Fig 1d). How Ybx1 suppresses H3K27me3, therefore, remains an open question. One way of addressing this question

in future studies could be to purify, and carefully analyze, the Ybx1 complex. If not with H3K4me3 methyltransferases, Ybx1 could still interact with other activator complexes, or chromatin remodelers.

Reviewer #4 (Remarks to the Author):

I am pleased with the answers provided and with the corrections made to the manuscript. I think the authors made a great effort introducing clear and conclusive experiments.

We thank the reviewers for taking the time to comment on this manuscript. The reviewers' points are in black color, and our response in blue color.

Reviewer #1 (Remarks to the Author):

The authors have done a good job addressing my main concerns. However, there are still serious issues with the new quantitative mass spectrometry data that the authors provide in the revised manuscript. The volcano plots that are presented look strange and parameters shown in the plots (i.e. Corr, R2 value etc) make no sense at all. Furthermore, volcano plots require triplicates (triplicates for IgG and specific pull-down) to enable T test statistics, how did the authors generate these plots with duplicate pull-downs? The authors are encouraged to study relevant interaction proteomics papers and their methods on this topic and adjust their experimental setup and data visualisation accordingly. The S7b figure also has serious problems: how can an interacting protein for Jarid2 (Ybx1) be much more enriched than the bait protein itself, is this indicative of antibody cross-reactivity? How much of the input samples did the authors load on the western blot shown in Figure 6C? Assuming this is 10% and looking at the amount of Ybx1 precipitated from the input, at most 1% of cellular Ybx1 is in complex with Jarid2. A similar pattern can be seen in Figure 6B, where the large majority of Jarid2 appears not to be bound to Ybx1. Thus, it would be important to acknowledge that the interaction between PRC2 and Ybx1 is highly substoichiometric and I would not use the term 'accessory subunit' as is used for PCL proteins, for example. As a consequence, some of the observed effects on chromatin and the interplay between Ybx1 and PRC2 in vivo may be indirect.

We appreciate the Reviewer's guidance to further improve the manuscript.

About Volcano plots

Previous Supplementary Fig. 1d was generated with duplicate IP datasets and tested with the F test, which allowed comparison of duplicate datasets. In the previous response (page 2), we stated different usages of F test and G test. Previous supplementary Fig. 7b was generated with triplicate IP datasets. We had included correlation coefficients and R values of the replicate datasets in order to show data consistency. We have now eliminated the presentation of these values and generated new plots with triplicate datasets for both Supplementary Fig. 1d and 7b. In Supplemental Fig. 7b, Ybx1 and Suz12 were both more enriched than the bait Jarid2, and Aebp2 and Ezh2 were nearly as enriched as Jarid2. This suggests that Jarid2 was identified at a relatively lower efficiency than the other proteins. This relatively lower efficiency of Jarid2 identification was likely not due to antibody cross-reactivity. The more likely causes could be less efficient identification of Jarid2 peptides. Jarid2 protein is nearly 3 times the size of Ybx1. However, we identified fewer peptides of Jarid2 (5, 3, and 5) than Ybx1 (7, 3, and 7) across triplicate datasets for Supplementary Fig. 7b. In the experiments for Supplementary Fig. 1d, we identified 14, 27, and 15 JARID2 peptides and 7, 4, and 15 YBX1 peptides. We regret that the second run of experiments was not as ideal.

We recognize that we need to improve the description of mass spectrometry. Below is the improved method section, in lines 546-571 on page 24: "For each of Supplementary Fig. 1d or 7b, we analyzed 3 biological replicate samples of JARID2 and IgG IP from the nuclear extract of H9 or NE4C, respectively. Samples were run on a short gel as described in a previously

published protocol⁶¹. Samples in gel slices were digested overnight at 37°C, acidified and the peptides were extracted in Acetonitrile. Extracts were dried down in a speed vacuum and reconstituted in 5% Formic acid. Digested peptides were loaded on a nanoscale capillary reverse phase C18 column (75 id, 10cm) by a HPLC system (Thermo Ultimate 3000or EASY-nLC 1000). Buffer A was 0.2% Formic acid and Buffer B was 70% Acetonitrile; 0.2% Formic acid. The peptides were eluted by increasing organic from 12%-70% over a 90-min liquid chromatography gradient. The eluted peptides were ionized by electrospray ionization, and detected by an inline mass spectrometer (Thermo LTQ Orbitrap Elite). The mass spectrometer was operated in data-dependent mode with a survey scan in Orbitrap (240,000 resolution, 1 X 10⁶ AGC target and 200 ms maximal ion time) and 20 low resolution MS/MS scans in the ion trap (CID, 2m/z isolation width, Normalized collision energy 35, AGC target 7 X 10⁴, and 250 ms maximal ion time) for each cycle. The raw data were searched against the UniProt mouse database concatenated with a reversed decoy database for evaluating false discovery rate. Database searches were performed using the Sequest⁶² v.28 (rev. 12) search engine. Searches were performed using a 25ppm mass tolerance for precursor and 0.5Da for product ions, fully tryptic restriction with two maximal missed cleavages, three maximal modification sites, and the assignment of b, and y ions. Carbamidomethylation of Cysteine (+57.02146 Da) was used for static modifications and Met oxidation (+15.99492 Da) was considered as a dynamic modification. All matched MS/MS spectra were filtered by mass accuracy and matching scores to reduce protein false discovery rate to <1%. Finally, all proteins identified in one gel lane were combined together. The total number of spectra, namely spectral counts (SC), matching to individual proteins may reflect their relative abundance in one sample after the protein size is normalized⁶³. The abundance values (spectral count × 50kD / protein size kD) of proteins were used to compared between the replicate datasets by T-test.”

About Fig. 6B and 6C

We agree with the Reviewer that the PRC2–Ybx1 interaction was substoichiometric in the nuclear extract. On chromatin, however, we respectfully emphasize that the binding pattern of Ybx1 highly overlapped that of Ezh2/1 of PRC2. The proportion of Ybx1 co-immunoprecipitated by Jarid2 was markedly larger than the proportion of Jarid2 immunoprecipitated by Ybx1. (We loaded 5% input in WB analyses.) Although the proportion of Ybx1 binding to Jarid2–PRC2.2 was low, the proportion of Jarid2–PRC2.2 binding to Ybx1, especially on chromatin, could be high. As Ybx1 is multi-functional, this result was not unexpected.

On lines 358-360 of page 15, we added “However, the enrichment of JARID2 in YBX1 co-IP and YBX1 in SUZ12 co-IP were low (Fig. 6c-d), suggesting that Ybx1 binding to PRC2 in the nuclear extract was substoichiometric.”

On pages 14 and 19, we deleted the statements that Ybx1 may be an accessory subunit of PRC2.

Also relevant is that we agree with Reviewer 3 the mechanistic activities of Ybx1 are not yet clear. On lines 477-479 on page 20, we added the following: “Alternative mechanistic scenarios include Ybx1 influencing gene activators or chromatin remodelers in order to reduce H3K27me3 at different loci. Future identification of other Ybx1 binding factors, as well as extensive biochemical, genetic, and genomic characterization will be required to pinpoint how Ybx1

affects PRC2 and potentially the activities of other chromatin modifiers.”

S1d

S7b

Volcano plots of triplicate IgG and Jarid2 IP-mass spectrometry from (S1d) the hESC or (S7b) NE4C nuclear extract. *P* value was calculated by the G-test comparing abundance values of JARID2 vs. IgG IP. Abundance was calculated with the formula of spectral count \times 50kD / protein size kD.

Reviewer #2 (Remarks to the Author):

I appreciate the thoughtful discussion and the additional experiments that the authors performed to address my concerns since the initial submission. The authors addressed most of our questions plus many more from the other Reviewers. While they did not use CRISPR to target Ybx1, the authors acknowledge these would be necessary but challenging experiments beyond the scope of this study. Overall, I think this paper contributes to our knowledge of the role of Ybx1 in the regulation of PRC2 in brain development and neuronal differentiation.

We thank the Reviewer for their support.

Reviewer #3 (Remarks to the Author):

The revised version has been improved by additional experiments and data analyses. The paper also reads better than the previous version. The authors have sufficiently answered the concerns raised by this reviewer.

I have the following minor comments to the authors.

We thank the Reviewer for the encouraging response and guidance to further improve the manuscript.

1. The title, as pointed out by other reviewers too, still looks cumbersome and somewhat difficult to understand. Given that the authors have such a nice story, and have performed quite a few elegant experiments, wouldn't it be better to make the title simpler and make their paper more accessible to the readers?

We have simplified the title to: “Ybx1 fine-tunes PRC2 activities to control embryonic brain development.”

2. I would suggest bringing the model (now in supplementary figure 9E) to one of the main figures (last panel in Figure 8?), as it nicely sums up the main findings of the story.

We thank the Reviewer for appreciating the model figure; it is now in Fig. 9j.

3. The title of the sub-chapter (page 15, line 367) could be simplified as “Ybx1-mediated regulation of NPCs involves inhibition of H3K27me3”.

We have changed the sub-chapter title.

The model that Ybx1 facilitates PRC2 binding but inhibits H3K27me3, although counter-intuitive, is very interesting. Usually, this would suggest that Ybx1 also interacts with other activator complexes- such as H3K4me3- to counterbalance H3K27me3 deposition. The authors, intriguingly, did not see any protein-protein interactions between Ybx1 and H3K4me3 (Fig 1d). How Ybx1 suppresses H3K27me3, therefore, remains an open question. One way of addressing this question in future studies could be to purify, and carefully analyze, the Ybx1 complex. If not with H3K4me3 methyltransferases, Ybx1 could still interact with other activator complexes, or chromatin remodelers.

We agree with the Reviewer's insight. We did not discuss the potential involvement of other activator complexes with Ybx1 due to worries about overreaching statements. In the discussion (lines 477-479 on page 20), we added the following: “Alternative mechanistic scenarios include Ybx1 influencing gene activators or chromatin remodelers in order to reduce H3K27me3 at different loci. Future identification of other Ybx1 binding factors, as well as extensive biochemical, genetic, and genomic characterization will be required to pinpoint how Ybx1 affects PRC2 and potentially the activities of other chromatin modifiers.”

Reviewer #4 (Remarks to the Author):

I am pleased with the answers provided and with the corrections made to the manuscript. I think the authors made a great effort introducing clear and conclusive experiments.

We thank the Reviewer for their support.

REVIEWERS' COMMENTS:

Reviewer #1 (Remarks to the Author):

The authors have addressed my remaining technical concerns.